# Subcellular mRNA kinetic modeling reveals nuclear retention as rate-limiting

David Steinbrecht [1,2,5], Igor Minia[3,5], Miha Milek[4], Johannes Meisig[1,2], Nils Blüthgen [1,2✉] & Markus Landthaler [3✉]

## Abstract

Eukaryotic mRNAs are transcribed, processed, translated, and degraded in different subcellular compartments. Here, we measured mRNA flow rates between subcellular compartments in mouse embryonic stem cells. By combining metabolic RNA labeling, biochemical fractionation, mRNA sequencing, and mathematical modeling, we determined the half-lives of nuclear pre-, nuclear mature, cytosolic, and membrane-associated mRNAs from over 9000 genes. In addition, we estimated transcript elongation rates. Many matured mRNAs have long nuclear half-lives, indicating nuclear retention as the rate-limiting step in the flow of mRNAs. In contrast, mRNA transcripts coding for transcription factors show fast kinetic rates, and in particular short nuclear half-lives. Differentially localized mRNAs have distinct rate constant combinations, implying modular regulation. Membrane stability is high for membrane-localized mRNA and cytosolic stability is high for cytosol-localized mRNA. mRNAs encoding target signals for membranes have low cytosolic and high membrane half-lives with minor differences between signals. Transcripts of nuclear-encoded mitochondrial proteins have long nuclear retention and cytoplasmic kinetics that do not reflect co-translational targeting. Our data and analyses provide a useful resource to study spatiotemporal gene expression regulation.

**Keywords** RNA Dynamics; Kinetic Modeling; Subcellular Fractionation; Nuclear Retention; Metabolic Labeling
**Subject Categories** Chromatin, Transcription & Genomics; RNA Biology

## Introduction

The life cycle of mRNA is a complex process involving multiple steps in different subcellular compartments (Glisovic et al, 2008). The precise regulation of mRNA dynamics in each step is critical for cellular transcript homeostasis (Berry and Pelkmans, 2022). In a typical mammalian cell, about 10,000 protein-coding genes are expressed, producing thousands of mRNAs per minute (Schwanhäusser et al, 2011). Protein-coding transcripts undergo maturation in the nucleus through splicing and polyadenylation, followed by export to the cytosol. Translation either occurs in the cytosol or at membrane-bound organelles, predominantly the endoplasmic reticulum (ER), culminating in their turnover. The resulting flow of mRNAs affects cell function by shaping the dynamic amount of mRNAs available for translation in the cytoplasm (Eisen et al, 2020; Mor et al, 2010) and the ER (Das et al, 2021). The overall mRNA abundance scales with cell size (Padovan-Merhar et al, 2015; Kempe et al, 2015; Battich et al, 2015; Swaffer et al, 2023).

Notably, mRNA half-lives exhibit a remarkable variability exceeding 100-fold among different protein-coding transcripts (Herzog et al, 2017; Dölken et al, 2008; Friedel et al, 2009). To unravel the nuanced regulation of individual processes in the mRNA life cycle, a crucial need arises for time-resolved and subcellular quantification.

Early studies on the subcellular dynamics of mRNA considered either all poly(A) RNA collectively (Jelinek et al, 1973) or only a few individual genes (Mor et al, 2010; Grünwald et al, 2011; Tutucci et al, 2018). Recent technological advances have enabled global measurements of mRNA dynamics and subcellular distribution. Metabolic labeling of newly synthesized RNA with uridine analogs has provided experimental means to monitor RNA dynamics with a minimal perturbation to the cell (Dölken et al, 2008). Using 4-thiouridine for labeling, kinetic rates of mRNA have been quantified on a transcriptome-wide level in a number of cell types from different species (Chen and van Steensel, 2017; Herzog et al, 2017; Rabani et al, 2011; Rutkowski and Dölken, 2017; Schofield et al, 2018). The first global measurement of global nucleocytoplasmic mRNA dynamics was achieved by Chen and colleagues in Drosophila S2 cells (Chen and van Steensel, 2017). However, metabolic labeling studies on mammalian cells have mostly considered the total cellular mRNA to determine kinetic rates, overlooking subcellular aspects (Herzog et al, 2017; Rabani et al, 2011; Rutkowski and Dölken, 2017). A recent study based on single-cell in situ sequencing of 5-ethynyl uridine-labeled RNA measured transcription, translocation and degradation of individual transcript molecules, uncovering subcellular mRNA profiles across time and space at the single-cell level for a collection of almost 1000 genes (Ren et al, 2023). Furthermore, two independent

[1]Charité—Universitätsmedizin Berlin, Institute of Pathology, Berlin, Germany. [2]Humboldt-Universität zu Berlin, Institute of Biology, Berlin, Germany. [3]Max Delbrück Center for Molecular Medicine in the Helmholtz Association (MDC), Berlin Institute for Medical Systems Biology, Berlin, Germany. [4]Core Unit Bioinformatics, Berlin Institute of Health at Charité, Berlin, Germany. [5]These authors contributed equally: David Steinbrecht, Igor Minia. ✉E-mail: nils.bluethgen@charite.de; markus.landthaler@mdc-berlin.de

works combined 4-thiouridine, cellular fractionation and RNA sequencing to measure the rates at which RNAs are exported from the nucleus in mammalian cells (Müller et al, 2024; Ietswaart et al, 2024), with one study additionally providing evidence for mRNA degradation in the nucleus (Ietswaart et al, 2024).

Here, we combined metabolic RNA labeling with cellular fractionation of mouse embryonic stem cells in nuclear and cytosolic, also a membrane-bound fraction composed mostly of the ER and mitochondria to produce mRNA sequencing data with spatial and temporal resolution. We developed a mathematical framework that enabled us to infer the kinetic rate constants of nuclear pre-, nuclear mature, cytosolic and membrane-bound mRNAs transcriptome-wide from our metabolic labeling and fractionation time series sequencing data. In addition, we estimated a transcript elongation rate to account for an observed labeling incorporation bias. Our method provides subcellular half-life information of the relative contributions of each step of the mRNA life cycle to transcript steady-state levels. We provide evidence that nuclear retention of mRNAs is the rate-limiting step in the life cycle of protein-coding transcripts. In addition, it uncovers expected differences in rates of mRNAs coding for distinct protein families, such as transcription factors and proteins of the secretory pathway. Comparison of the kinetic rate constants for these steps across genes will provide novel insights into subcellular mechanisms of differential gene regulation.

## Results

### Spatiotemporal measurement of newly transcribed mRNA

To capture the nucleocytoplasmic kinetics of mRNA, we conducted a time-resolved SLAM-seq experiment in combination with subcellular fractionation in mouse embryonic stem cells (mESCs). Briefly, mESCs were exposed to 4-thiouridine (4sU) for various times to label newly synthesized RNA. Subsequently, we performed subcellular biochemical fractionation to generate cytosolic, membrane and nuclear fractions (Fig. 1A). 4sU concentrations were carefully chosen for sufficient labeling rates while minimizing potential toxicity elicited by 4sU, with higher concentrations of 4sU (500 µM) for labeling periods from 15 min to 1 h, and lower concentrations (100 µM) from 1 to 3 h. A differential gene expression analysis confirmed that 4sU labeling caused none or minimal perturbation of transcriptome (Fig. EV1A). The efficacy of the fractionation procedure was confirmed by Western blot analysis with antibodies against compartment-specific marker proteins. The nuclear proteins histone H3, lamin A/C (LMNA) and TATA-binding protein (TBP), the cytoplasmic proteins GAPDH and β-tubulin (TUBB), and the ER marker protein BCAP31 were only detectable or strongly enriched in their respective fraction. In contrast, RPS6, a component of the 40S ribosomal subunit, was found in all fractions (Fig. 1B).

To quantify the subcellular kinetics of mRNA globally, we next extracted RNA from each fraction and from whole cells, followed by iodoacetamide (IAA) alkylation and poly(A)-selected strand-specific mRNA library preparation. Briefly, sequencing reads were processed, aligned to the mouse genome and quantified for gene expression, T and T2C conversion counts in both exons and introns

(see "Methods" for details). Based on a binomial mixture model (Jürges et al, 2018), we determined the conversion rate per sequenced RNA sample. With the conversion rates, T and T2C conversion counts the share of new to total mRNA was calculated on intron and exon level, which was used later on as an input for the kinetic model (see Fig. 1A, bottom).

The experimental and computational complexity demanded an evaluation of the effects of biological and technical variability on mRNA quantification. Principal component analysis (PCA) revealed that the largest differences in mRNA quantification were a consequence of the assayed subcellular compartment rather than the difference in biological replicates (Fig. 1C), indicating that the data captured biological variability well with a low influence of technical effects.

To convert relative to absolute subcellular RNA expression ratios, we needed to quantify how the total amount of RNA is shared between the subcellular compartments. Under the assumption that the RNA expression in the whole cell can be reconstructed by summing the subcellular RNA expression with a corresponding factor for each, we fit the relative abundance of nuclear, cytosolic and membrane mRNA with the constraint that they sum up to 1 (the whole-cell expression). We estimated that 39%, 15%, and 45% of total mRNA is present in the nucleus, cytosol and membrane-bound, respectively (see Fig. 1D). These relative abundances are used to derive expression ratios mimicking absolute RNA levels that are related to kinetic parameter ratios and can therefore be used to constrain the parameter space (see Box 1).

Comparing gene expression in the cytosol and the membrane, we observed a bimodal distribution, with 17% and 78% of the transcripts being enriched at the membrane and in the cytosol, respectively (Fig. 1E). Based on this distribution, we classified mRNAs into cytosol- and membrane-localized, where membrane-localized means that the cytoplasmic expression of a mRNA is highly enriched in the membrane fraction compared to the majority of mRNAs, and a small group of undefined transcripts not belonging to these compartments. This classification determined the particular model used in the fitting process (see bottom left Fig. 1A).

In addition, the distribution of share of labeled mRNAs over all quantified genes ($n = 9809$) reflected the cellular mRNA maturation and showed expected trends for all subcellular compartments (Fig. 1F). Collectively, these results showed that the quantification of global mRNA kinetics in subcellular fractions of mouse ES cells was of high quality.

### Model of subcellular mRNA dynamics

To integrate the subcellular transcriptome and metabolic RNA labeling data and to estimate half-lives as a description of subcellular mRNA dynamics, we developed transcript-wise mathematical models similar to previous work (Chen and van Steensel, 2017). Each transcript was modeled by a linear, inhomogeneous system of ordinary differential equations (ODE) containing three or four steps for cytosol- or membrane-localized transcripts, respectively, describing the life cycle from mRNA transcription to degradation (see Box 1). After transcription in the nucleus, nuclear pre-mRNA is processed to mature mRNA (pre-mRNA processing rate $k_1$). From there, mature mRNA can be exported to the cytosol or already be degraded (nuclear export $k_2$ and nuclear degradation $\gamma_2$). By default, the nuclear

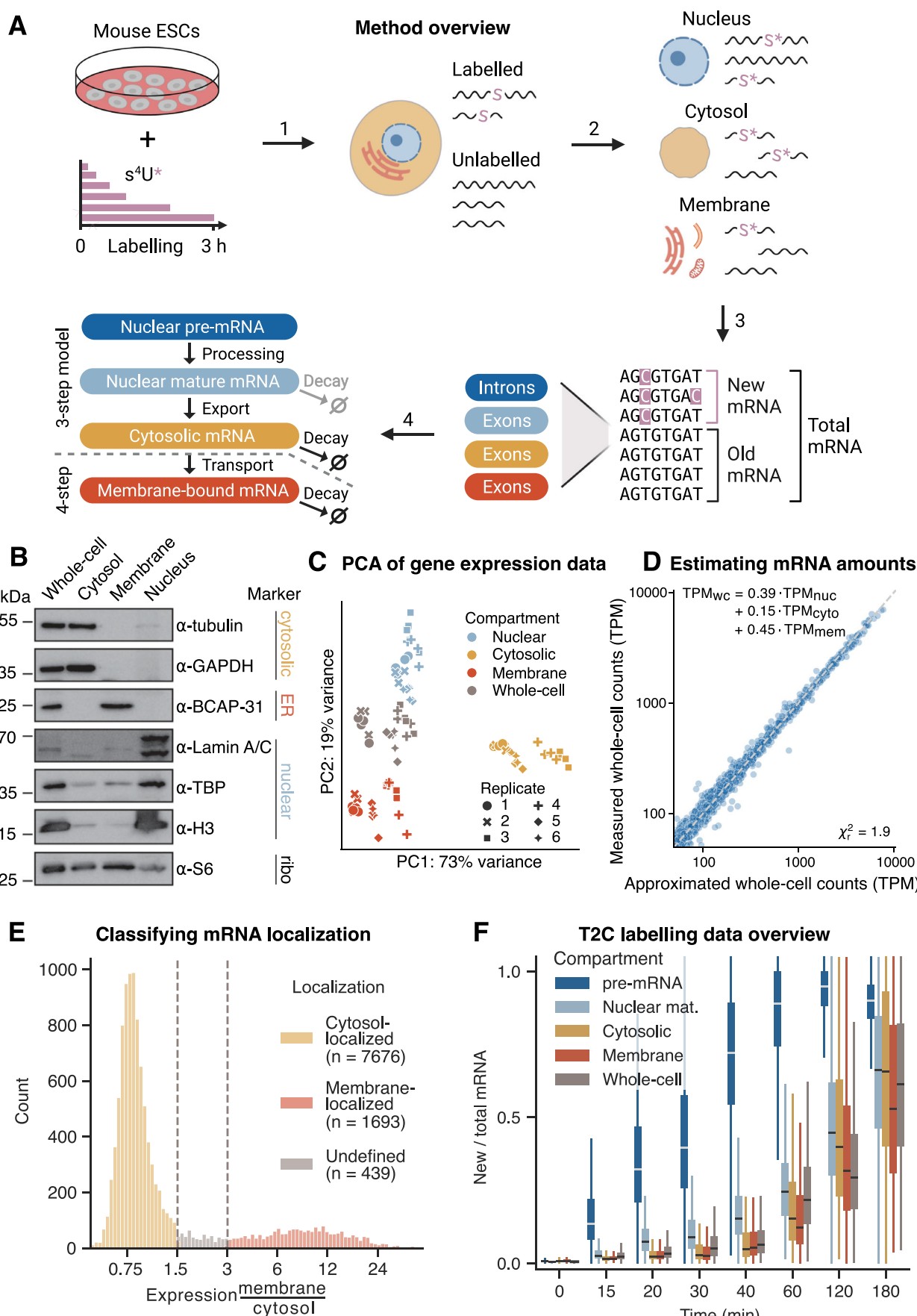

**Figure 1.   Newly transcribed mRNA is measured spatio-temporally using cell fractionation and metabolic labeling RNA sequencing.**

(A) Schematic overview of experimental and computational steps. Mouse embryonic stem cells (mESCs) are labeled with 4sU for 15, 20, 30, 40, 60, 120, and 180 min, leading to T2C conversions in newly transcribed RNA (1). Cells are fractionated into nuclear, cytosolic and membrane-bound compartments (2). Ratios between new and total mRNA are estimated in a Bayesian framework for exons and whole intronic regions (3) and given as input to a kinetic model to fit subcellular kinetic rate parameters (4). See Methods for details on each individual step. (B) Results from Western blot analysis measuring nuclear, cytosolic, endoplasmic reticulum, and ribosomal protein markers in the subcellular fractions and the whole-cell extract. (C) Principal component analysis of log-transformed bulk mRNA-seq data in transcripts per million (TPM) using 500 most variable genes. Samples cluster by compartment. (D) Estimates of relative mRNA abundance in subcellular compartments. Subcellular TPM values are fitted to whole-cell TPM values, resulting in estimates for the relative abundance of mRNA in each compartment, see equation in top left. The reduced chi-squared value is shown in the bottom right. (E) Classification of membrane and cytosolic mRNA localization in mESCs through subcellular mRNA expression. Histogram of the ratio between steady-state gene expression (TPM, mean over all time points) in membrane over cytosolic fraction, from here on membrane enrichment, for 11,711 most expressed genes. Vertical gray lines indicate chosen cutoffs to classify mRNA localization. For membrane enrichment <1.5, <3, and >3: cytosol-localized, undefined, and membrane-localized, respectively. Numbers of successfully fitted genes in each localization category are shown in the legend. (F) Time- and compartment-resolved box plot of T2C labeling data ($n = 9795$). Center lines of box plots depict the median values. Medians of share of new to total mRNA increases with labeling time. Lower and upper hinges of box plots correspond to the 25th and 75th percentiles, respectively. Lower and upper whiskers extend from the hinge to the smallest or largest value no further than the 1.5× interquartile range from the hinge, respectively. Source data are available online for this figure.

**Box 1.   Kinetic model of the mRNA life cycle**

The individual equations in Eq. (1) describe the change in absolute amount of new nuclear pre- ($x_1$, dark blue), nuclear mature ($x_2$, light blue), cytosolic ($x_3$, yellow) and membrane-bound ($x_4$, red) mRNA. The biological process of each parameter is illustrated in the sketch on the bottom left. We rescale the system by dividing each compartment with its steady-state solution, so that Eq. (2) describe the change in relative share of new mRNA ($\tilde{x}_i$). The analytical solutions to Eq. (2) are shown in Box 2 in "Methods". We use the solutions of the first three (3-step model) and all four equations (4-step model) to fit cytosol- and membrane-localized

mRNAs, respectively. The measured subcellular mRNA expression ratios, multiplied by their corresponding relative mRNA amount factor (see Fig. 1D), are used to constrain the parameter space via the steady-state ratios (see middle right), allowing to distinguish nuclear export from decay. By default, the nuclear decay $\gamma_2$ was set to 0 and was only allowed to vary if fitting was significantly improved by including it (see Fig. EV5). The production rate of pre-mRNA p is not present in the rescaled system and hence not fitted.

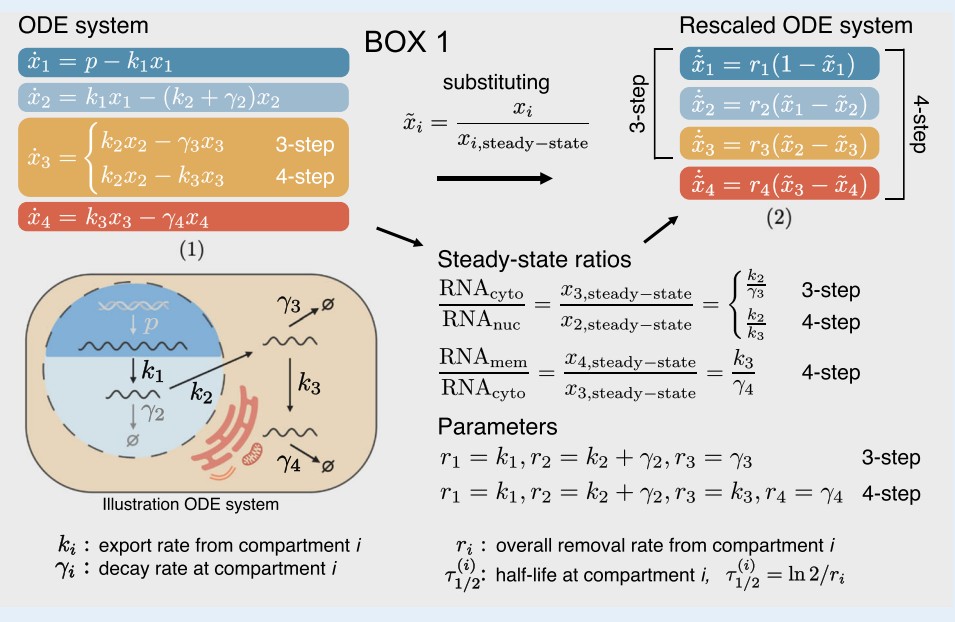

degradation parameter is set to 0 and is only included as a variable parameter in the model if fitting is significantly improved (see "Methods" and Fig. EV5). In the 3-step model, mRNA is translated and degraded in the cytosol (cytosolic decay $\gamma_3$), whereas in the 4-step model, mRNA is first localized to the ER membrane or other membrane-bound organelles (cytosolic transport $k_3$, no decay), where it is translated and subsequently degraded (membrane decay $\gamma_4$). We derived a system of ODEs describing the relative amount of new

mRNA. The share of new to total mRNA estimated from the SLAM-seq data is used as input in the solutions of this ODE system to fit the parameters mentioned above.

For ease of interpretation, we will mainly use half-lives instead of rates in both text and figures in the following (see Box 1), namely: pre-mRNA processing ($\frac{ln2}{k_1}$), nuclear retention ($\frac{ln2}{k_2+\gamma_2}$), cytosolic stability ($\frac{ln2}{\gamma_3}$ or $\frac{ln2}{k_3}$) and membrane stability ($\frac{ln2}{\gamma_4}$). Nuclear retention time is used as a general term to comprise the various

events after maturation of a single transcript molecule until its emergence into the cytoplasm, including chromatin dissociation, nuclear diffusion, and binding to and transport across the nuclear pore. The pre-mRNA processing rate describes the rate at which introns present in polyadenylated RNA are removed. As splicing already occurs co-transcriptionally, it is not the splicing rate that is quantified here, but rather a post-polyadenylation mRNA maturation rate. Figures that include both the elongation rate and subcellular kinetic parameters display the corresponding rates (in kb/min and min$^{-1}$, respectively).

## Modeling of transcriptional elongation rates

For mRNAs whose transcription is initiated prior to labeling with 4sU, but terminated after labeling start, we expected that only part of the transcript was labeled. More specifically, we expect that sequences at a distance of d to the 3' end are only labeled after $t = d/v$, where $v$ is the transcription elongation rate (see Fig. 2A). Therefore, particularly for longer transcripts and shorter time points, partial labeling affects our measurements and has to be taken into consideration.

Our transcriptome sequencing data covers polyadenylated RNA, and quantification of labeling includes sequences spanning the entire transcripts. When we grouped exonic sequences by distance to the 3' end, we observed that the median share of labeled mRNAs in the nuclear fraction decreased with increasing distance to the 3' end, particularly for shorter time periods (see Fig. 2A). To account for this bias, which would otherwise lead to an underestimation of the subcellular mRNA flow rates, we incorporate a transcript elongation rate in our model. Specifically, we modeled each transcript with one variable per exon, where the dynamics of each exon is modeled using the model described above but with a time delay according to $\tilde{t} = t - d/v - \delta$ with d being the distance of the exon to the 3' end (see Fig. 2B) and $\delta = 5$ min being an overall delay until any labeling is experimentally observed, possibly due the time required to add a poly(A) tail to an mRNA after transcription termination. All exon models of one transcript share the same kinetic parameters.

Our method of estimating the transcription elongation rate has its caveats, particularly when compared to methods that more directly measure those. Firstly, we noted that the T2C conversion rate increases over time, even at constant 4sU concentrations (see Fig. EV1E). For genes requiring more than 30 min to be fully transcribed, exons near the 5' end are labeled with lower efficiency than those near the 3' end, leading to an underestimation of elongation rates. Secondly, our time resolution of 15min does not allow estimation of elongation rates of shorter genes, which would require more frequent sampling (e.g., at 8 and 12 min). Interestingly, however, we observed that the average elongation rate estimates are in agreement with previous reports using more direct methods (see next section).

## Model parameterization unveils length-dependent elongation rate

To derive estimates of kinetic parameters and elongation rates, we fitted the models to the time-resolved SLAM-seq data. More specifically, we simultaneously optimized model parameters for each transcript such that they best fit (a) the share of labeled mRNA

at the level of each exon for the different time points and (b) the steady-state levels of the transcripts in the nucleus, cytoplasm and membrane compartment. Out of 8501 transcribed genes with sufficient labeling data in multiple exons, the elongation rate estimation converged for 6494 genes with a mean of 1.6 kb/min and a median relative error of 30% (see Fig. EV3A). For the remaining multiple-exon genes, estimates were at the set boundaries and did not converge. Boundary-limited estimates were deemed unreliable and hence excluded from all analyses regarding the elongation rate. Furthermore, we performed a sensitivity analysis and found that, for the typical transcript, variation of the elongation rate within the 95% confidence interval changes the other subcellular kinetic parameters by only 2–8% (see Fig. EV3D). For gene lengths shorter than 12 kb, between 12 kb and 30 kb and longer than 30 kb, the mean elongation rate is 1.0, 1.4, and 2.2 kb/min, respectively (see Fig. 2C). Increases in the elongation rate of RNA polymerase II for longer genes, as well as our quantitative range of values, coincide with results from previous studies focusing specifically on transcript elongation (Jonkers et al, 2014; Fuchs et al, 2014; Veloso et al, 2014; Shao et al, 2022). The main point of estimating the elongation rate here is to correct for the observed labeling bias. For state-of-the-art quantifications, results from the studies cited above should be considered.

## Nuclear retention time is the rate-limiting step in the life cycle of most mRNAs

For mRNAs derived from roughly 9800 genes, we observed three common kinetic profiles: (i) for transcripts with fast turnover labeled RNA dynamics are similar in all compartments (exemplified by *Myc*, Fig. 3A), with the exception of nuclear pre-mRNA; (ii) cytosol-stable transcripts show a slower accumulation of labeled RNA in the cytosolic than in the nuclear mature compartment (e.g., *Nf1*, Fig. 3A), with overall turnover varying from fast to slow; (iii) membrane-localized mRNAs, as defined earlier, accumulate labeled RNA equally fast in nuclear mature and cytosolic compartments, but significantly more slowly in the membrane than in the cytosolic compartment (e.g., *Tfrc*, Fig. 3A). This distinct pattern is due to the low cytosolic expression and the short cytosolic residence time of these transcripts.

All kinetic parameters for transcripts in different compartments were highly variable (see Fig. 3B). Nuclear pre-mRNA half-life showed the narrowest distribution, ranging from 10 to 46 min (10th and 90th percentile, resp.) with a median of 22 min. Cytosolic transcript half-lives displayed the widest distribution and the highest variability, ranging from 1 to 42 min (10th and 90th percentile, respectively) with a median of 10 min. Membrane mRNA half-lives range from 6 to 69 min with a median of 21 min. Interestingly, our data suggest that the nuclear retention of mature mRNAs is the rate-limiting step for most transcripts with a median half-life of 78 min and ranging from 25 min to 236 min. This observation agrees with findings by Müller and colleagues describing the nucleus-to-cytosol step to be rate-limiting (Müller et al, 2024).

Based on the Bayesian Information Criterion (BIC), we tested if subcellular mRNA dynamics are better described by a model including nuclear degradation of mature mRNA and found strong support (difference in BIC >10) of nuclear degradation for 579 transcripts (see Fig. EV5A and "Methods"). Nuclear decaying

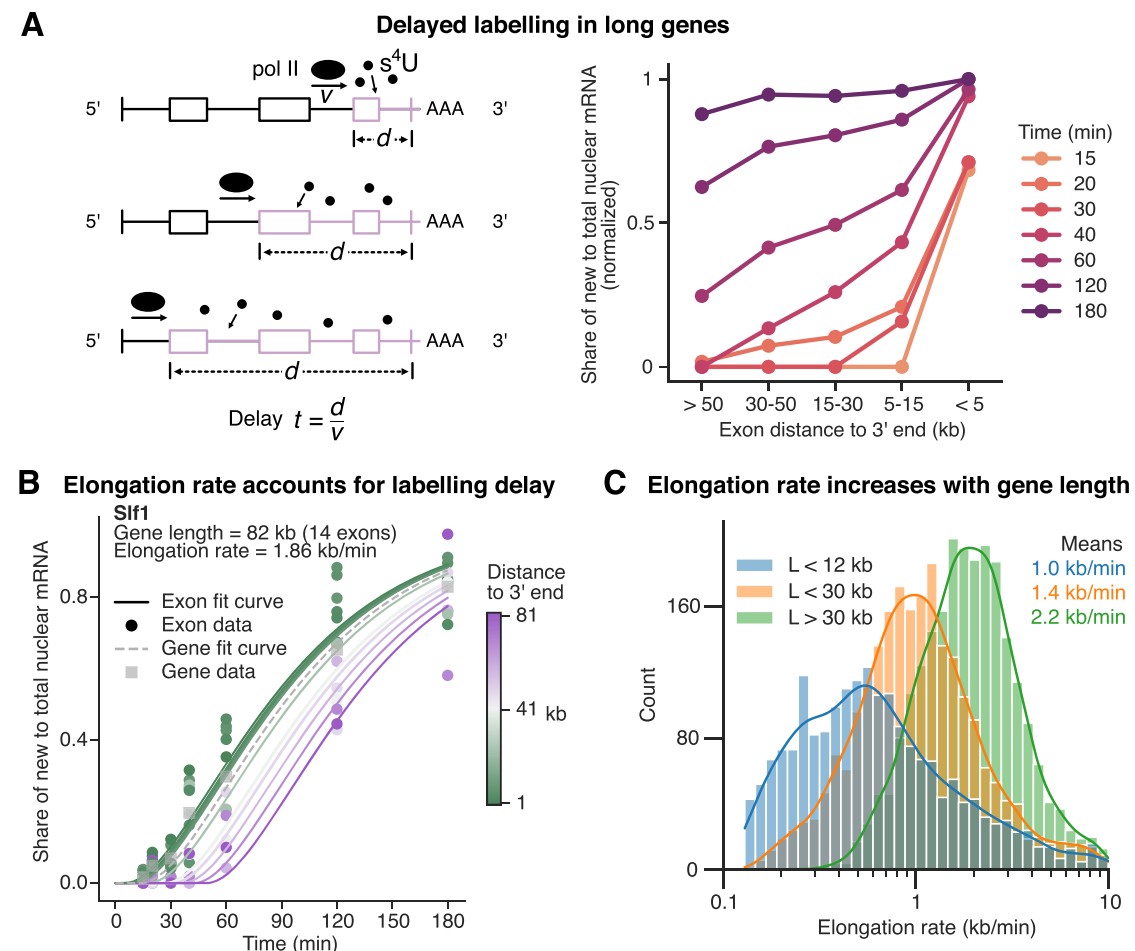

**Figure 2. Transcript elongation rate is estimated to account for labeling delay in long genes.**

(A) Left: Sketch to illustrate how exons (in longer genes) are labeled with a time delay that increases linearly with distance to 3′ end. As poly(A)+ sequencing was used, transcripts close to being fully transcribed will contain T2C conversions first (close to the 3′ end), while it takes longer to observe conversions in the whole transcript. Right: Pointplot of new over total exonic, nuclear mRNA binned by distance to 3′ end. Points show medians across exons per bin and time and are normalized to the 15 min and <5 kb point. Lines connect points of equal time. Equal labeling efficiency is seen after 3 h. (B) Fit result of Slf1 (only nuclear mRNA) is shown to illustrate the fitting procedure. Share of new to total mRNA per exon is plotted over the 4sU labeling time until cell harvesting. All exons of a gene share the same kinetic parameters, but each exonic curve is time-delayed by $\tilde{t} = t - d/v - \delta$, with $d$ being the mean exonic distance to 3′ end, $v$ the elongation rate and $\delta$ the overall delay of 5 min (straight lines). Typically, the closer an exon is to the 3′ end (indicated by color), the higher its ratio of new mRNA is (points). The time-delayed fit accounts for this bias. Gray squares show the gene-level T2C data. Gray, dashed line shows a fit curve delayed by mean 3′ end distance weighted by expression of exons. (C) Histogram with kernel-density estimation (KDE) of converged elongation rates ($n = 6494$) grouped by gene length (L). Mean rate per group is shown in the top right. Source data are available online for this figure.

transcripts show a median decay rate four times higher than the median retention rate in the nucleus, longer pre-mRNA processing, shorter nuclear half-lives and longer cytosolic and membrane stability than other transcripts (see Fig. EV5B). Performing an over-representation analysis we were able to associate roughly 100 genes with nuclear decaying mRNA to DNA processes (see Fig. EV5C).

In summary, our analysis suggests that most mRNAs spend more than half of their lifetime in the nucleus as mature spliced transcripts. Recent results using metabolic labeling RNA sequencing suggested that nuclear transcripts remain associated to DNA for a longer time, and once dissociated they are exported rather quickly (Ietswaart et al, 2024).

## Functionally related transcripts tend to have similar kinetic profiles

It has been previously shown that groups of mRNAs encoding proteins with specific molecular functions display distinct kinetics rates along the mRNA life cycle (Chen and van Steensel, 2017; Ren et al, 2023). To this end, we performed gene set enrichment analyses (GSEA) considering the pre-mRNA processing, nuclear retention, cytosolic and membrane stability, respectively, to identify groups of transcripts from the gene ontology (GO) that have particularly short or long residence time in each compartment. Each GSEA resulted in a large number of significantly enriched GO terms, with a subset shown in Fig. 3C. For the nuclear retention

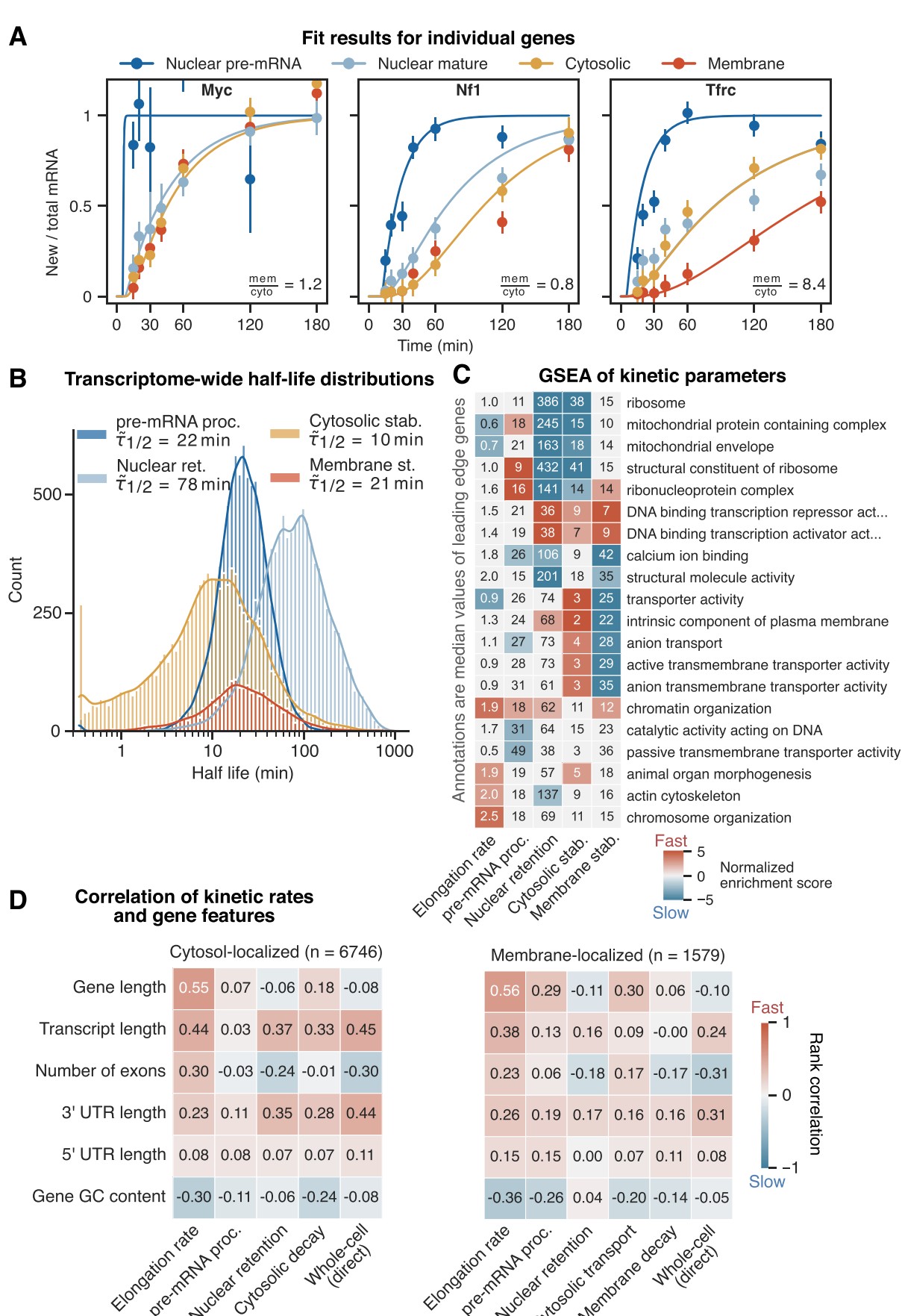

**A  Fit results for individual genes**

Nuclear pre-mRNA Nuclear mature Cytosolic Membrane

**Myc** — $\frac{mem}{cyto} = 1.2$

**Nf1** — $\frac{mem}{cyto} = 0.8$

**Tfrc** — $\frac{mem}{cyto} = 8.4$

New / total mRNA

Time (min)

**B  Transcriptome-wide half-life distributions**

pre-mRNA proc. $\tilde{\tau}_{1/2} = 22\ min$

Nuclear ret. $\tilde{\tau}_{1/2} = 78\ min$

Cytosolic stab. $\tilde{\tau}_{1/2} = 10\ min$

Membrane st. $\tilde{\tau}_{1/2} = 21\ min$

Count

Half life (min)

**C  GSEA of kinetic parameters**

Annotations are median values of leading edge genes

| | Elongation rate | pre-mRNA proc. | Nuclear retention | Cytosolic stab. | Membrane stab. | |
|---|---|---|---|---|---|---|
| 1.0 | 11 | 386 | 38 | 15 | ribosome |
| 0.6 | 18 | 245 | 15 | 10 | mitochondrial protein containing complex |
| 0.7 | 21 | 163 | 18 | 14 | mitochondrial envelope |
| 1.0 | 9 | 432 | 41 | 15 | structural constituent of ribosome |
| 1.6 | 16 | 141 | 14 | 14 | ribonucleoprotein complex |
| 1.5 | 21 | 36 | 9 | 7 | DNA binding transcription repressor act... |
| 1.4 | 19 | 38 | 7 | 9 | DNA binding transcription activator act... |
| 1.8 | 26 | 106 | 9 | 42 | calcium ion binding |
| 2.0 | 15 | 201 | 18 | 35 | structural molecule activity |
| 0.9 | 26 | 74 | 3 | 25 | transporter activity |
| 1.3 | 24 | 68 | 2 | 22 | intrinsic component of plasma membrane |
| 1.1 | 27 | 73 | 4 | 28 | anion transport |
| 0.9 | 28 | 73 | 3 | 29 | active transmembrane transporter activity |
| 0.9 | 31 | 61 | 3 | 35 | anion transmembrane transporter activity |
| 1.9 | 18 | 62 | 11 | 12 | chromatin organization |
| 1.7 | 31 | 64 | 15 | 23 | catalytic activity acting on DNA |
| 0.5 | 49 | 38 | 3 | 36 | passive transmembrane transporter activity |
| 1.9 | 19 | 57 | 5 | 18 | animal organ morphogenesis |
| 2.0 | 18 | 137 | 9 | 16 | actin cytoskeleton |
| 2.5 | 18 | 69 | 11 | 15 | chromosome organization |

Fast — 5
0
Slow — −5

Normalized enrichment score

**D  Correlation of kinetic rates and gene features**

Cytosol-localized (n = 6746)

| | Elongation rate | pre-mRNA proc. | Nuclear retention | Cytosolic decay | Whole-cell (direct) |
|---|---|---|---|---|---|
| Gene length | 0.55 | 0.07 | -0.06 | 0.18 | -0.08 |
| Transcript length | 0.44 | 0.03 | 0.37 | 0.33 | 0.45 |
| Number of exons | 0.30 | -0.03 | -0.24 | -0.01 | -0.30 |
| 3' UTR length | 0.23 | 0.11 | 0.35 | 0.28 | 0.44 |
| 5' UTR length | 0.08 | 0.08 | 0.07 | 0.07 | 0.11 |
| Gene GC content | -0.30 | -0.11 | -0.06 | -0.24 | -0.08 |

Membrane-localized (n = 1579)

| | Elongation rate | pre-mRNA proc. | Nuclear retention | Cytosolic transport | Membrane decay | Whole-cell (direct) |
|---|---|---|---|---|---|---|
| Gene length | 0.56 | 0.29 | -0.11 | 0.30 | 0.06 | -0.10 |
| Transcript length | 0.38 | 0.13 | 0.16 | 0.09 | -0.00 | 0.24 |
| Number of exons | 0.23 | 0.06 | -0.18 | 0.17 | -0.17 | -0.31 |
| 3' UTR length | 0.26 | 0.19 | 0.17 | 0.16 | 0.16 | 0.31 |
| 5' UTR length | 0.15 | 0.15 | 0.00 | 0.07 | 0.11 | 0.08 |
| Gene GC content | -0.36 | -0.26 | 0.04 | -0.20 | -0.14 | -0.05 |

Fast — 1
0
Slow — −1

Rank correlation

◄ **Figure 3.  Across the transcriptome, transcripts show a wide distribution and different combinations of subcellular kinetic parameters.**

(**A**) Fit results of three exemplary genes. Ratio of new to total mRNA is shown over labeling time. Points with error bars depict means with standard errors of new to total ratio across replicates per (*n* = 2–4, all fractions shown). Lines are fit results, delayed as the gene-level curve in Fig. 2B (only fitted compartments shown). Ratio between steady-state expression in membrane and cytosolic compartments is shown in bottom right, indicating if 3-step or 4-step model was used (<1.5 or >1.5, respectively, see Fig. 1E). (**B**) Histogram of nuclear pre-mRNA processing (*n* = 8515), nuclear retention (*n* = 9809), cytosolic stability (*n* = 9809) and membrane stability (*n* = 2131) mRNA half-lives. Median half-life of each parameter is shown in legend. (**C**) Heatmap of results from five GSEAs based on the gene ontology (GO). Genes were ranked by log-scaled, z-scored parameter rates. For each parameter, the two most up- and downregulated terms are shown. In addition, five terms with the lowest adjusted *P* values and adjusted *P* values <0.05 in at least three columns are shown. *P* values were corrected using the Benjamini–Hochberg (BH) method. Gray indicates the term is not significant. Values in heatmap are median half-lives (rate in the first column) of the union of leading edge genes across each row. (**D**) Heatmaps of correlation between kinetic rates and gene features. Transcripts are split into cytosol- (left) and membrane-localized (right). Values in heatmap are spearman rank correlations. Transcript length is the length of the most expressed isoform based on RSEM results. 3′ and 5′ UTR lengths are isoform-specific. Gene GC content is taken from Biomart and includes both intronic and exonic regions. Source data are available online for this figure.

time (see Fig. EV4B), mRNAs encoding transcription regulators have a high positive enrichment score, i.e., short nuclear retention time, while those involved in metabolic processes, specifically translation, have a high negative enrichment score, i.e., a long nuclear retention time. Most transcripts of mitochondrial and ribosomal genes show particularly long nuclear half-lives. This may mechanistically be explained by their enrichment in nuclear speckles, where mRNA transcripts of both gene groups were found to be retained and post-transcriptionally processed (McIntyre et al, 2023). However, part of the long nuclear retention for transcripts encoding for mitochondrial proteins that are co-translationally targeted to the outer mitochondrial membrane (*n* = 102) might be due to a slight mitochondrial contamination in the nuclear fraction (see Fig. EV1). Transcripts associated with transport or ion homeostasis show short cytosolic and long membrane half-lives. When focussing on membrane-localized transcripts, those associated with regulating protein modifications showed the shortest (~14 min), while those associated with calcium ion binding showed the longest (~48 min) membrane half-lives (see Fig. EV4C).

Next, we investigated if kinetic parameters were correlated with gene features, including gene length, number of exons, and length of UTRs. The strongest association was observed with the transcriptional elongation rate, as noted above, and presented a rank correlation of more than 0.5 with gene length (see Fig. 3D). Transcript length shows moderate negative correlation with nuclear retention, cytosolic and whole-cell half-lives for cytosol-localized transcripts, but only weak or no correlation for membrane-localized transcripts. Interestingly, the number of exons seems to play an important role in subcellular mRNA kinetics. Nuclear retention and whole-cell half-lives are both positively correlated with the number of exons in both cytosol- and membrane-localized transcripts (see Fig. 3D). Therefore, mRNAs with many exons take longer to be exported but are then overall more stable. However, for cytosol-localized transcripts, the cytosolic half-life is not correlated with exon number, while the membrane half-life for membrane-localized transcripts is. mRNA derived from longer genes seem to be degraded faster in the cytosol, but not at the ER (Fig. 3D). 3′ UTR length correlates moderately for cytosol- and weakly for membrane-localized mRNAs with all subcellular parameters except pre-mRNA processing, suggesting that the longer the 3′ UTR end, the shorter the half-life as described previously (Spies et al, 2013). Out of the subcellular parameters, nuclear retention is the best predictor of whole-cell half-life (Spearman correlation of *r* = 0.92 and *r* = 0.87 for model-derived and directly-fitted whole-cell half-life, respectively).

## Differentially localized mRNAs exhibit distinct dynamic behavior

To associate mRNA localization with function, we ranked transcripts by membrane enrichment, which was also used to classify mRNA localization, and performed a GSEA. The proteins encoded by transcripts highly enriched in the membrane fraction are located at the plasma membrane, cell surface and endoplasmic reticulum and are involved in transmembrane transport and ion homeostasis (Fig. 4A). Proteins of the more than 7000 cytosol-localized transcripts are distributed across the cell and have a wide range of biological functions, but those encoded by transcripts most highly enriched in the cytosol are located primarily in the nucleus and are involved in organization of chromatin and chromosomes and transcription regulation (Fig. 4A).

Next, we investigated the relationship between subcellular mRNA localization and dynamics. Membrane half-lives are high for membrane-localized transcripts (median = 20 min), while cytosolic half-lives are high for cytosol-localized transcripts (median = 13 min), being quantitatively similar in both compartments with an overall slightly higher membrane stability (see Figs. 4B and EV4A). This seems to indicate that there is a slight, but no significant difference of transcript stability between mRNAs that are translated by ribosomes in the cytosol and mRNAs being targeted to and translated at the ER. Once a membrane-associated transcript is in the cytoplasm, it reaches the membrane-bound compartment rather fast, with timescales ranging from less than a minute to around ten minutes. The degree of membrane enrichment influences dynamics, with higher membrane enrichment leading to shorter cytosolic and longer membrane stability. On the other hand, the degree of cytosolic enrichment seems to have little influence on cytosolic stability.

We next examined if transcripts encoding different targeting sequences show different behavior. We observed that the different targeting signals have a similar median cytosolic half-life of around 2 min (see Fig. 4C). Membrane-localized mRNAs encoding no known targeting signal surprisingly show the shortest median cytosolic half-life, while transcripts encoding only a signal peptide tend to have a slightly longer median cytosolic half-life. Distributions of membrane half-life vary slightly for different targeting signals, with transcripts encoding transmembrane helices having the lowest median (19 min) and transcripts encoding proteins with signal peptides having the highest (32 min). mRNA encoding tail-anchored proteins, with the first transmembrane helix close to the C-terminus, have a much longer median cytosolic half-life than

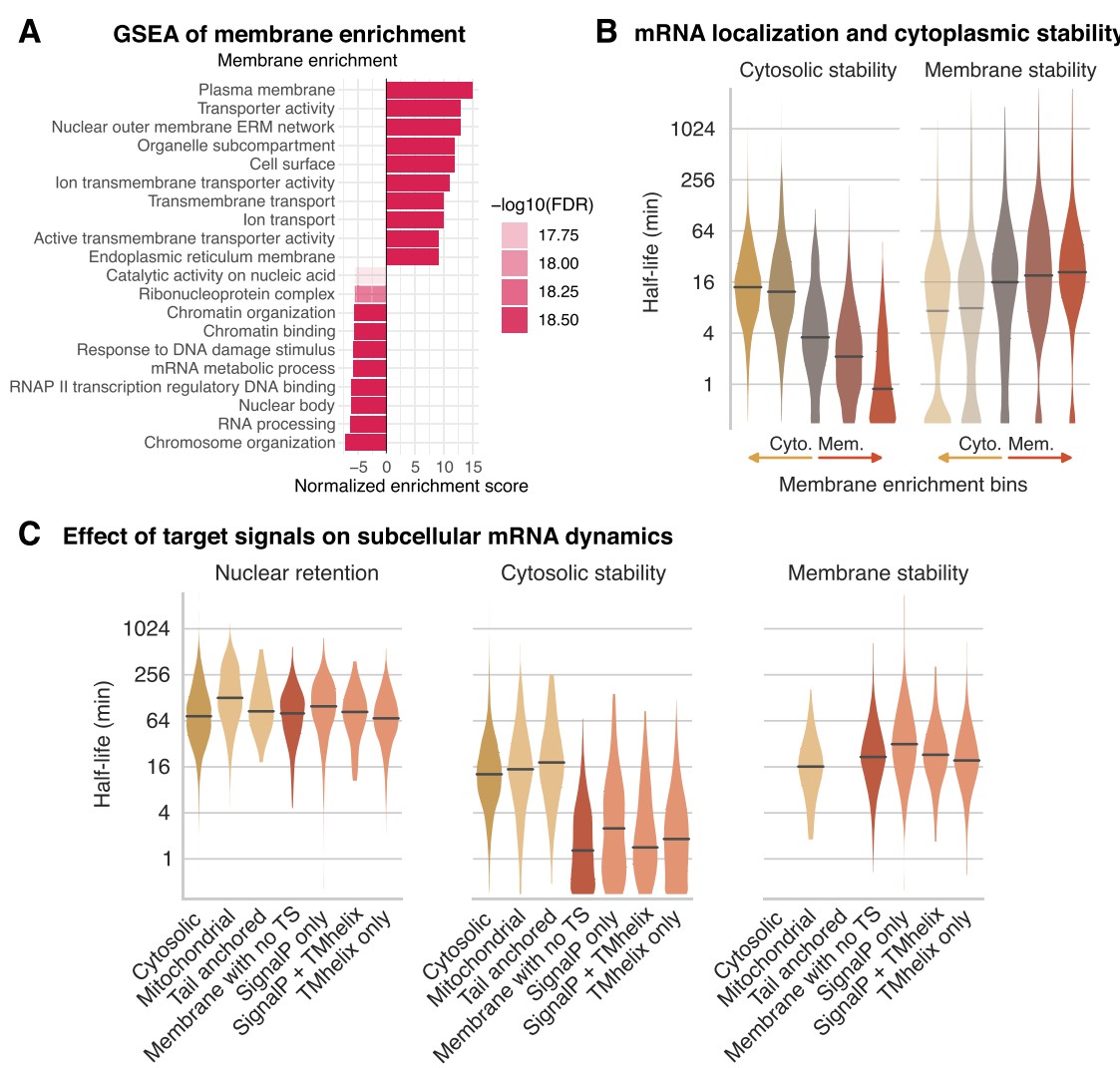

**Figure 4. Differentially localized mRNAs exhibit distinct combinations of kinetic rate constants.**

(A) Result of a GSEA ranking genes by the ratio of membrane over cytosolic RNA expression (membrane enrichment). False-discovery rates (FDR) are BH-corrected. (B) Violin plot of cytosolic and membrane half-lives with transcripts binned by membrane enrichment (see Fig. 1E with cytosol- and membrane-localized further split at 0.8 and 8.7, respectively; from bin 1 to 5: $n = 4409$, $n = 3254$, $n = 437$, $n = 844$, and $n = 844$). Yellow, gray and red colors indicate cytosol, undefined and membrane localization, respectively. Membrane half-lives of cytosol-localized transcripts from additional 4-step model fit are shown transparently. Center lines of violin plots depict the median values. (C) Violin plot of nuclear mature, cytosolic and membrane half-lives with transcripts classified by encoded targeting signals (TS) as in (Zinnall et al, 2022). Classification from left to right: cytosol-localized transcripts with no TS ($n = 6262$), nuclear DNA-encoded mitochondrial proteins ($n = 763$ with 661 being cytosol-localized), transcripts with tail-anchored transmembrane proteins ($n = 78$), membrane-localized transcripts with no known TS ($n = 531$), transcripts encoding signal peptides ($n = 310$) or transmembrane helices ($n = 956$) or both ($n = 47$). Center lines of violin plots depict the median values. Yellow and red colors indicate cytosol and membrane localization, respectively. Source data are available online for this figure.

other co-translationally targeted transcripts. This suggests that co-translational targeting to the ER for transmembrane helix-containing genes happens after the transmembrane helix is translated. Cytosolic or membrane half-lives did not correlate with the number of encoded transmembrane helices, further suggesting that the distance of the first transmembrane helix to the N-terminus influences cytoplasmic kinetics. Nuclear-encoded mitochondrial transcripts are mostly cytosol-localized (661 out of 763) and tend to have similar kinetics to other cytosol-localized transcripts except for nuclear half-lives, which are significantly longer (median of 128 min). Only 74 mitochondrial transcripts are

membrane-localized and have similar cytoplasmic kinetics as ER-localized mRNAs, suggesting that most mitochondrial proteins are post-translationally targeted to mitochondria and only a small fraction are co-translationally targeted to the outer mitochondrial membrane.

## Validation with external datasets and independent approaches

mRNA half-lives have mostly been estimated on the whole-cell level. To be able to compare our subcellular half-lives for mESC to

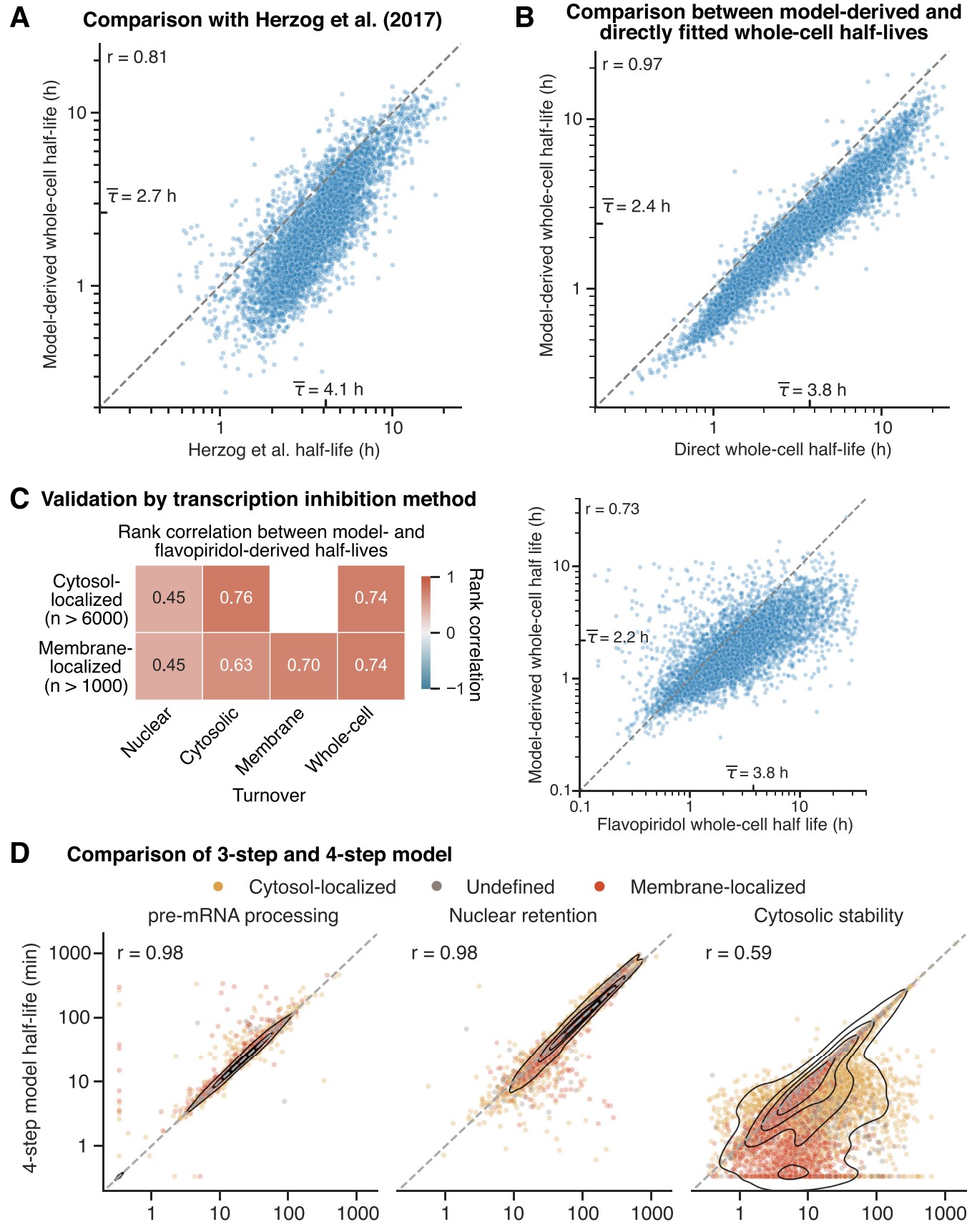

◄        Figure 5.    Validation of subcellular parameter estimates with external datasets and independent approaches.

(A) Comparison with results from first SLAM-seq experiment (Herzog et al, 2017). Scatterplot of model-derived (y axis) and Herzog et al (x axis) whole-cell half-lives (n = 5475). Model-derived whole-cell half-lives are calculated from subcellular rates, see Appendix Fig. S2 and "Methods". Mean half-life is annotated on the corresponding axis. Spearman rank correlation is shown on top left. The dashed, gray line is the identity line. (B) Scatterplot of model-derived and directly-fitted whole-cell half-lives (n = 9809). Model-derived whole-cell half-lives as in (A). Direct whole-cell half-lives obtained by fitting a simple "one minus exponential decay" model to the share of new to total mRNA from whole-cell extract data. Mean half-life is annotated on the corresponding axis. Spearman rank correlation is shown on top left. The dashed, gray line is the identity line. (C) Comparison of model- to flavopiridol-derived half-lives. Left: Correlation heatmap of model- with flavopiridol-derived subcellular half-lives split into cytosol- (upper row) and membrane-localized transcripts (lower row). Values in heatmap are spearman rank correlations, also indicated by color. For details on how half-lives using flavopiridol were derived, see "Methods" and Appendix Fig. S2. Right: Scatterplot of model-derived and flavopiridol whole-cell half-lives (n = 6316). Model-derived whole-cell half-lives as in (A). Mean half-life is annotated on the corresponding axis. Spearman rank correlation is shown on top left. Dashed, gray line is the identity line. (D) Comparison of 3-step and 4-step model, which model mRNA flow until cytosolic and membrane-bound compartments, respectively. Scatterplot of pre-mRNA processing, nuclear retention and cytosolic half-lives with values from 3-step and 4-step model on x and y axis, respectively (n = 9369, all genes were fit with both models). Black lines are 2-dimensional KDE to indicate density of points. For details on the models, see Model sections in "Results" and "Methods". Spearman rank correlation is shown on top left. Dashed gray line is the identity line. Source data are available online for this figure.

those whole-cell half-lives from previous studies, we used our model to predict model-derived whole-cell half-lives from the kinetic parameters and the estimated relative abundance of the three fractions (see Fig. 1D and "Methods" for details). Herzog et al estimated whole-cell RNA half-lives in mESCs as a proof-of-concept when establishing SLAM-seq (Herzog et al, 2017). Our model-derived and Herzog et al whole-cell half-lives show high correlation ($r = 0.81$, see Fig. 5A). For transcripts with half-lives longer than 4 h, values agree by ranking (90th percentile 4.6 h and 6.8 h for model-derived and Herzog et al, respectively). Interestingly, we find a diverging trend for half-lives that are shorter than 2 h: the half-lives derived by Herzog et al show a high density around 1.5 h, while our model-derived half-lives have a longer tail with values ranging down to 0.5 h (10th percentile 0.9 h and 2.0 h for model-derived and Herzog et al, respectively). This may indicate a lower sensitivity of detecting shorter half-lives for the whole-cell pulse-chase design.

In order to augment the robustness of our investigation into the estimated half-lives, we further compared our model-derived half-lives with a simple "one minus exponential decay" model that we fit on SLAM-seq data of our whole-cell samples, referred to as "direct" whole-cell half-lives in the following. Comparing the direct to the model-derived whole-cell half-lives, we find a near-perfect rank correlation ($r = 0.97$), but the values were systematically lower in our compartment model (see Fig. 5B; mean half-life 2.4 h and 3.8 h for model-derived and direct, respectively). This bias was most likely a result of the length-dependent labeling bias as well as the non-exponential nature of the process. Interestingly, the 90th percentile value of the direct half-lives is higher compared to both our model-derived half-lives and those estimated by Herzog et al (7.6 h, 4.6 h, and 6.8 h, respectively).

To experimentally validate our metabolic labeling-derived subcellular flow rates with an independent (but more interfering) method, we inhibited transcription using flavopiridol (Chao and Price, 2001) followed by subcellular fractionation and RNA sequencing, resulting in a second set of subcellular rates (see "Methods" for details). Reassuringly, we find high correlation between the two sets of cytosolic, membrane and whole-cell half-lives for both cytosol- and membrane-localized mRNAs. Interestingly, the correlation was only moderate for nuclear mature half-lives (see Fig. 5C, left) which may be due to stress-induced changes in nuclear RNA turnover. When comparing the flavopiridol- and metabolic labeling-derived whole-cell half-lives (Fig. 5C, right), we

find that for half-lives longer than 4 h, flavopiridol-derived half-lives were generally longer than metabolic labeling-derived ones, suggesting reduction in mRNA decay after transcription blockage.

As we used different models for cytosol and membrane-localized mRNAs (3-step and 4-step model, respectively), we asked how model choice influences the kinetic parameters. We therefore fitted both mathematical models to our subcellular SLAM-seq data, and compared the half-lives between the two models. We found a near-perfect correlation for pre-mRNA processing ($r = 0.98$) and nuclear retention ($r = 0.98$), which are modeled the same way in both models (see Fig. 5D). However, for the cytosolic stability we find lower agreement ($r = 0.59$), as here the 4-step model used additional information to constrain the ratio between cytosolic and membrane half-lives. This results in all membrane-localized, but also many cytosol-localized transcripts exhibiting lower cytosolic half-lives in the 4-step than in the 3-step model. For the former, this was desired, as T2C mutations are distinct between cytosol and membrane compartments, whereas for the latter, this is undesirable, as the T2C mutations are similarly frequent in both membrane and cytosol compartments and the fit then systematically underestimates the cytosolic half-life to accurately fit the membrane half-life.

Taken together, our validation efforts using our and published datasets suggest that the compartment-derived rates provide a high-quality quantitative assessment of mRNA turnover.

## Discussion

Our study provides a comprehensive analysis of and resource for nucleocytoplasmic mRNA kinetics in mESCs. Applying mathematical modeling to time-resolved subcellular SLAM-seq data we were able to quantify intracellular mRNA flow and offer valuable insight into the dynamics of the mRNA metabolism. Previously, mRNA kinetics were studied mostly on the whole-cell level, while our approach yields information on subcellular kinetics, even dissecting cytoplasmic into cytosolic and membrane dynamics, distinguishing it from recent studies on subcellular mRNA kinetics (Ietswaart et al, 2024; Müller et al, 2024).

Here, we modeled subcellular mRNA kinetics using a linear, inhomogeneous system of ordinary differential equations, accounting for different steps in the life cycle of mRNA molecules. Incorporating steady-state ratios in the fitting procedure allowed us to distinguish between export and degradation in the nuclear compartment and

ensured that the fitted parameters align with the measured subcellular mRNA expression levels. The quantitative accuracy of the steady-state ratios was ensured by accounting for the varying amounts of mRNA present in the different cellular compartments in a manner similar to a recently published method (Dai et al, 2022).

To account for an observed labeling bias toward the 3' end of transcripts when using poly(A)-selection and full-length transcript sequencing, we included a transcript elongation rate in our model. This modeling approach allowed for the estimation of kinetic parameters of pre-mRNA processing, nuclear retention, cytosolic, and membrane stability for mRNAs on a global level, including an approximation of transcriptional elongation rates. Our estimated mean elongation rates of 1.0 to 2.2 kb/min for shorter and longer genes, respectively, are highly similar to median elongation rate estimations of 1 kb/min measured before in mESCs (Shao et al, 2022) and 1.25–1.75 kb/min measured before in five different human cell lines (Veloso et al, 2014). Furthermore, our analysis revealed that transcriptional elongation was higher for longer genes, as observed previously (Veloso et al, 2014; Jonkers et al, 2014). In both of these two studies, differences in gene length-specific elongation rates were correlated with distinct histone modifications, suggesting that gene structure and epigenetic modification influence RNA polymerase II elongation rates. Although we have good agreement with previous findings, our method to estimate elongation rates has limits. It cannot consider the increase of labeling efficiency early in the time series, for which a dataset with long-read sequencing of transcripts would be better suited. Furthermore, a more densely sampled time series would be required to improve the transcript elongation estimate for shorter genes.

Across transcripts, the timescales of the subcellular kinetic parameters span multiple orders of magnitude. One of the key findings of our study was the observation that nuclear retention is the rate-limiting step for most transcripts, as evidenced by the significantly longer median half-life of 78 min for mature mRNA in the nucleus compared to the cytosolic half-life of 10 min. This finding is consistent with prior works applying imaging-based approaches, demonstrating that nuclear retention of transcripts serves as an effective mechanism for buffering noise (Battich et al, 2015; Bahar Halpern et al, 2015). Battich and coworkers derived nuclear retention times between ~5–90 min for close to 300 newly synthesized transcripts, with a median of ~20 min (Battich et al, 2015). However, the authors argued that these nuclear retention times are likely an underestimation for most other genes, since these genes were fast-responding genes during stress signaling. Moreover, Halpern and colleagues showed that in mouse tissues spliced and polyadenylated mRNAs are retained in the nucleus for many protein-coding genes to reduce cytoplasmic gene expression noise (Bahar Halpern et al, 2015). Ren and coworkers found that a substantial fraction of newly synthesized transcripts was retained in the nucleus even after 6 h, corroborating the previous observation of nuclear mRNA retention (Ren et al, 2023). Choi and colleagues conducted mRNA interactome-capture experiments combined with pulse-chase labeling in HeLa cells and found that export-competent and cytoplasmic translating mRNPs are assembled mainly at 60–70 min and 90–120 min chase time, respectively (Choi et al, 2024). Similarly, by applying sequencing of metabolically labeled RNA in the nuclear and cytosolic compartments followed by mathematical modeling, Müller et al recently reported that mRNA molecules generally spend most of their life in the nucleus, highlighting nuclear retention as a critical determinant of subcellular mRNA dynamics (Müller et al, 2024). The relative amount of poly(A) mRNA in the nucleus varies across cell types (Bahar Halpern et al, 2015;

Dai et al, 2022; Gondran et al, 1999). As the nucleus makes up a large part of the cell volume in mESCs and we observe a high relative nuclear mRNA amount, the nuclear retention half-lives estimated here are likely to be higher than in other, differentiated cell types.

By comparing models excluding and including nuclear degradation, we found evidence for nuclear degradation of mature mRNA for 6% of expressed genes. Most of the transcripts from those genes are degraded in the nucleus, but those that are exported to the cytoplasm show a high cytoplasmic stability.

To identify functional categories associated with genes exhibiting extreme subcellular rates, we undertook gene set enrichment analyses. mRNAs transcribed from genes involved in transcription regulation were enriched among those with short nuclear half-lives. Conversely, mRNAs from genes associated with translation and metabolic processes, in particular mitochondrial and ribosomal genes showed enrichment among those with longer nuclear half-lives. Interestingly, mRNAs of nuclear-encoded mitochondrial genes presented mostly similar cytoplasmic kinetics when compared to cytosol-localized transcripts, suggesting mostly post-translational targeting to the mitochondria with only around 10% of transcripts being co-translationally targeted.

Using cytosolic and membrane mRNA expression levels, we were able to identify cytosol– and membrane-localized transcripts with the latter mostly being associated with the ER membrane. These membrane transcripts were localized within two minutes to the ER, mostly independent of the targeting signal, and then were stable for around 20 min at the ER, with transcripts targeted by the signal recognition particle having slightly longer half-lives. Cytosolic half-lives of cytosol-localized transcripts were centered slightly below 20 min, revealing that cytosol- and membrane-localized mRNAs have similar cytoplasmic stability, suggesting that differences in protein expression between cytosol- and membrane-localized transcripts with similar mRNA levels derive mainly from differences in translational efficiency (Voigt et al, 2017; Lashkevich and Dmitriev, 2021; Zinnall et al, 2022). An intriguing prospect for future inquiry revolves around the impact of translation on the subcellular localization of mRNA. Recent findings have shown that, notably, when translation initiation is inhibited, the ER becomes the predominant site for the localization of newly exported mRNAs, prompting questions about the intricate relationship between translation and subcellular mRNA localization (Child et al, 2023).

Since nuclear retention is the rate-limiting factor, it is harder to correctly estimate all processes taking place afterward, especially if they happen on a comparatively short time scale. Using the well-measured steady-state ratios helps to essentially determine the short and difficult to estimate cytosolic stability from the longer and more easily determined membrane stability in the case of membrane-localized transcripts, where slight contaminations between nuclear and membrane compartment would further hinder correct estimation.

Based on subcellular parameters, we calculated model-derived whole-cell half-lives that correlate highly with previous experimentally determined whole-cell half-life estimates. This analysis provided confidence in the accuracy of the subcellular rates and their ability to represent whole-cell mRNA kinetics. Both model-derived and direct whole-cell half-lives were correlated highly with nuclear retention half-lives. To validate the subcellular flow rates further, we inhibited transcription and performed subcellular fractionation, generating a second set of subcellular rates. These rates showed good agreement with the original rates, further supporting the reliability of the findings.

In conclusion, our findings offer deeper insights into the dynamics of mRNA metabolism and uncover compartment-specific features of post-transcriptional regulation, showing that mature mRNAs spend most of their lifetime in the nucleus in mESCs. The relationship between mRNA kinetics and gene functions give directions for further investigation to analyze the RNA-binding proteins and molecular processes, like translation, influencing subcellular mRNA dynamics. These findings provide a foundation for future research into the mechanisms of mRNA processing and localization within mammalian cells, ultimately contributing to our broader understanding of gene expression regulation in a subcellular context.

# Methods

**Reagents and tools table**

| Reagent/resource | Reference or source | Identifier or catalog number |
| --- | --- | --- |
| **Experimental models** | | |
| E14TG2a mESCs (*M. musculus*) | Iacovino et al, 2014 | |
| **Recombinant DNA** | | |
| **Antibodies** | | |
| Goat anti-rabbit HRP | Agilent | P044801-2 |
| Goat anti-mouse HRP | Agilent | P044701-2 |
| Mouse anti-beta-tubulin | Sigma-Aldrich | T8328-200UL |
| Mouse anti-GAPDH | Sigma-Aldrich | G8795 |
| Mouse anti-TBP | Abcam | ab818 |
| Mouse anti-Lamin A/C | Invitrogen | 14-9688-80 |
| Rabbit anti-H3 | Abcam | ab1791 |
| Rabbit anti-BCAP31 | Proteintech | 11200-1-AP |
| Rabbit anti-S6 | Cell Signaling Technology | 2217 |
| **Oligonucleotides and other sequence-based reagents** | | |
| RT-PCR primer p21 gene | This study | Methods |
| Random hexamers | ThermoFisher Scientific | 48-190-011 |
| **Chemicals, enzymes, and other reagents** | | |
| Advanced DMEM/F12 | ThermoFisher Scientific | 12634028 |
| Neurobasal | ThermoFisher Scientific | 21103049 |
| Knockout™ DMEM | ThermoFisher Scientific | 10829018 |
| Fetal Bovine Serum qualified for ES cells | Life Technologies | 16141079 |
| N2 Supplement | ThermoFisher Scientific | 17502048 |
| B27 Supplement | ThermoFisher Scientific | 17504001 |
| GlutaMax | ThermoFisher Scientific | 35050061 |
| MEM non-essential amino acid | ThermoFisher Scientific | 11140050 |
| Nucleosides | Merck Millipore | ES-008-D |
| β-mercaptoethanol | ThermoFisher Scientific | 21985023 |
| CHIR99021 inhibitor | Invitrogen | SML1046-5MG |
| PD0325901 inhibitor | Invitrogen | PZ0162-5MG |
| Leukemia inhibitory factor | Merck Millipore | ESG1107 |

| Reagent/resource | Reference or source | Identifier or catalog number |
| --- | --- | --- |
| 4-thiouridine (4sU) | ChemGenes | RP-2304 |
| DPBS, no calcium, no magnesium | ThermoFisher Scientific | 14190169 |
| Cycloheximide | Biochemika | A0879,0001 |
| Complete Protease Inhibitor Cocktail, EDTA-free | Roche | COEDTAF-RO |
| RNaseOUT™ Recombinant Ribonuclease Inhibitor | ThermoFisher Scientific | 10777019 |
| Digitonin | Sigma-Aldrich | D141-100MG |
| Amersham ECL Western Blotting Detection Reagent | GE Healthcare | RPN2209 |
| Flavopiridol | Biotrend | HY-10005-10mg |
| Trizol LS | ThermoFisher Scientific | 10296028 |
| Nitrocellulose blotting membrane | GE Healthcare | 10600004 |
| TruSeq Stranded mRNA Library Prep Kit | Illumina | 20020594 |
| SuperScript III Reverse Transcriptase | ThermoFisher Scientific | 18080085 |
| Random hexamers | ThermoFisher Scientific | 48-190-011 |
| SYBR Green PCR Master Mix | Applied Biosystems | 43-643-46 |
| IGEPAL | Sigma-Aldrich | 7365-45-9 |
| 1,4-Dithiothreitol (DTT) | Sigma-Aldrich | 10197777001 |
| Dimethyl sulfoxide (DMSO) | Sigma-Aldrich | D8418-250ML |
| Iodoacetamide (IAA) | Sigma-Aldrich | I6125-5G |
| NuPAGETM NOVEXTM 4-12% Bis-Tris | ThermoFisher Scientific | NP0323BOX |
| **Software** | | |
| https://github.com/steinbrecht/subcellular-SLAM | This study | |
| Adobe Illustrator | https://www.adobe.com/products/illustrator.html | |
| STAR (v2.7.6) | https://github.com/alexdobin/STAR | |
| HTSeq-count (v0.11.1) | https://htseq.readthedocs.io/en/latest/ | |
| RSEM (v.1.3.3) | https://deweylab.github.io/RSEM/ | |
| DESeq2 (v1.40.2) | https://github.com/genepattern/DESeq2 | |
| **Other** | | |
| Centrifuge 5417R | Eppendorf | |
| Centrifuge Multifuge 3SR+ | Thermo Scientific | |
| Amersham Imager 680 | GE Healthcare | |
| HiSeq 4000 | Illumina | |
| NanoDrop One/One^c | ThermoFisher Scientific | |
| Qubit 2.0 Fluorometer | ThermoFisher Scientific | |

## Mouse embryonic stem cell (mESC) culture

E14TG2a mESCs (Iacovino et al, 2014) were cultured in 0.1% gelatin [w/v]-coated plates in "2i + LIF" ES medium [Advanced DMEM/F12 (12634028, ThermoFisher Scientific)—Neurobasal (21103049, ThermoFisher Scientific)—Knockout™ DMEM (10829018, ThermoFisher Scientific) (1:1:0.5), 14% Fetal Bovine Serum qualified for ES cells (16141079, Life Technologies), 1× N2 (17502048, ThermoFisher Scientific), 1× B27 (17504001, Thermo-Fisher Scientific), 1× GlutaMax (35050061, ThermoFisher Scientific), 1× MEM Non-Essential Amino Acid (11140050, ThermoFisher Scientific), 1× Nucleosides (ES-008-D, Merck Millipore), 100 µM β-mercaptoethanol (21985023, ThermoFisher Scientific), 3 µM CHIR99021 (SML1046-5MG, Invitrogen) and 1 µM PD0325901 (PZ0162-5MG, Invitrogen), 1000 U/ml Leukemia inhibitory factor (ESG1107, Merck Millipore)], under a controlled atmosphere at 5% $CO_2$ and 37 °C. mESCs were seeded the day before the experiments at a density of $3 \times 10^5$ cells/ml.

## Metabolic labeling and cell fractionation

Independently passaged biological replicates of mESCs (~$3.5 \times 10^7$ cells per replicate) were separately labeled in "2i + LIF" ES medium supplemented with 500 µM (4 replicates for 20 and 60 min; 2 replicates for 15, 30, and 40 min; 3 replicates for 0 min) or 100 µM (2 replicates for 60, 120, and 180 min) 4-thiouridine (4sU, RP-2304, ChemGenes) and fractionated by sequential detergent extraction, as described previously (Jagannathan et al, 2011) with minor modifications. Briefly, medium was aspirated, 5 ml of ice-cold PBS supplemented with 100 µM cycloheximide (A0879,0001, Biochemika) was added to the plates, cells were scraped from the plates, transferred to 15ml falcon and spun down. Pellet was resuspended in 500 µl of ice-cold permeabilization buffer (110 mM KOAc, 25 mM K-HEPES pH 7.2, 2.5 mM Mg(OAc)2, 1 mM EGTA with freshly added 0.015% digitonin, 1 mM DTT, 100 µg/ml cycloheximide, 1× Complete Protease Inhibitor Cocktail and 40 U/mL RNaseOUT™). 100 µl of the sample was taken aside as Total extract and rest was incubated for 10 min at 4 °C with rotation, followed by centrifugation at 3000 × g 5 min at 4 °C. Supernatant (corresponding to the Cytosolic fraction) was transferred to new tube while the pellet was resuspended in 5 ml of wash buffer (110 mM KOAc, 25 mM K-HEPES pH 7.2, 2.5 mM Mg(OAc)2, 1 mM EGTA with freshly added 0.004% digitonin, 1mM DTT, 100 µg/ml cycloheximide) and spun down again at 3000 × g 5 min at 4 °C. After centrifugation washed pellet was mixed with 500 µl of ice-cold lysis buffer (400 mM KOAc, 25 mM K-HEPES pH 7.2, 15 mM Mg(OAc)2, 0.5% (v/v) NP-40 and freshly added 1 mM DTT, 100 µg/ml cycloheximide, 1× Complete Protease Inhibitor Cocktail, 40 U/mL RNase Out) and incubated for 5 min on ice followed by centrifugation at 3000 × g 5 min at 4 °C to collect the supernatant (corresponding to the Membrane fraction) and the pellet (insoluble and the nuclear fraction). For additional purity nuclei were loaded on 10% sucrose cushion in lysis buffer and centrifuged at 200 × g 5 min at 4 °C. The cytosolic and membrane fractions were clarified at 7500 × g 10 min at 4 °C to remove cell debris. In total, 20 µl of all fractions were taken for western analysis while the rest were mixed with Trizol LS (10296028, ThermoFisher Scientific) for subsequent RNA isolation.

## RNA isolation, alkylation of 4sU-labeled RNA, and SLAM-seq

RNA extraction was carried out by the manufacturer's protocol but including 0.1 mM DTT (final concentration) during isopropanol precipitation and dissolving RNA in 1mM DTT to prevent oxidation of thiol groups (Herzog et al, 2017). For a typical SLAM-seq experiment, 5 µg of DNase-treated total RNA were incubated in reaction mix (50 mM sodium phosphate pH 8.0, 50% DMSO, 10 mM iodoacetamide) at 50 °C for 15 min. The reaction was stopped by adding 1 µl of 1M DTT and RNA was ethanol precipitated. One microgram of total RNA was used as an input for TruSeq Stranded mRNA Library Prep Kit (20020594, Illumina) following the manufacturer's instructions. Briefly, poly-A containing mRNA molecules were captured from the total RNA sample using magnetic beads. Following this selection, the mRNA was fragmented, and complementary DNA (cDNA) was synthesized. Illumina adapters with unique barcode sequences for each sample were then attached, and the library is amplified through PCR. The multiplexed, full-length transcript libraries were sequenced using HiSeq 4000 (Illumina) for pair-end 75 cycles by the BIH Genomics platform at the Max Delbrück Center for Molecular Medicine.

## Western blotting

Protein lysates from cell fractionation experiments were separated on 10% SDS PAGE, transferred to nitrocellulose blotting membrane (10600004, GE Healthcare), blocked in 5% dry milk and probed for cytosolic (beta-tubulin, GAPDH), membrane (BCAP31), nuclear (Lamin A/C, TBP, H3) markers and S6 ribosomal protein. Antibodies were used at a dilution of 1:5000 for anti-beta-tubulin (T8328-200UL, Sigma-Aldrich, mouse), 1:25,000 for anti-GAPDH (G8795, Sigma-Aldrich, mouse), 1:2000 for anti-TBP (ab818, Abcam, mouse), 1:500 for anti-Lamin A/C (14-9688-80, Invitrogen, mouse), 1:1000 for anti-H3 (ab1791, Abcam, rabbit), anti-BCAP31 (11200-1-AP, Proteintech, rabbit) and anti-S6 (2217, Cell Signaling Technology, rabbit) and detected by 1:2000 dilution of respective secondary HRP-antibody-conjugates (anti-rabbit, P044801-2, Agilent; anti-mouse, P044701-2, Dako). Primary antibodies were incubated at room temperature for one hour followed by three washing steps for 5 min and incubation with secondary antibodies for one hour. Images were acquired using Amersham ECL Western Blotting Detection Reagent (RPN2209, GE Healthcare) on an Amersham Imager 680 (GE Healthcare).

## Transcriptional inhibition by flavopiridol

Three independently passaged biological replicates of mESCs (~$3.5 \times 10^7$ cells per replicate) were cultured in "2i + LIF" ES medium supplemented with 1 µM flavopiridol (HY-10005-10mg, Biotrend) to block the transcription. Cells were harvested at 0, 30, 60, 120, and 180 min after addition of flavopiridol followed by cell fractionation as described above. Obtained fractions were mixed with Trizol LS and RNA was isolated following the manufacturer's instructions. One microgram of total RNA was used as an input for TruSeq Stranded mRNA Library Prep Kit according to the instructions of the manufacturer. The multiplexed, full-length

transcript libraries were sequenced using HiSeq 4000 for pair-end 75 cycles by the BIH Genomics platform at the Max Delbrück Center for Molecular Medicine.

## RT-qPCR

For assessment of stress response caused by 4sU toxicity mESCs were incubated with 100, 250, 500 μM 4sU for 0, 60, 120, and 240 min with subsequent RNA isolation. Reverse transcription was performed by SuperScript III Reverse Transcriptase (18080085, ThermoFisher Scientific) and random hexamers (48-190-011, ThermoFisher Scientific) following the manufacturer's instruction. Around 100 ng of the synthesized cDNA was used as an input for 20 μl qPCR reaction using SYBR Green PCR Master Mix (Applied Biosystems) with the gene-specific primer pairs targeting stress-responsive gene p21 (forward: 5'-TCGCTGTCTTGCACTCTGGTGT-3', reverse: 5'-CCAATCTGCGCTTGGAGTGATAG-3'). The mean CT value was calculated for three biological replicates.

## Custom genome annotation for intronic regions

In order to sum up mutations separately for intronic gene regions, we created a custom genome annotation containing intronic regions based on the Gencode Release M14 (GRCm38.p5). Intronic regions were determined from the gaps between exons and further filtered by exonic regions from other genes. For alignment of exons, the unaltered genome annotation from Gencode was used.

## Alignment and read counting

After demultiplexing and adapter trimming, raw sequencing files were aligned using STAR (v2.7.6), once for exons and once for introns with the custom genome annotation. The following STAR options were used: "outFilterMultimapNmax=10, alignSJDBoverhangMin=3, outFilterMismatchNmax=35, alignEndsType=EndToEnd and seedSearchStartLmax=10". To increase the mappability of highly labeled fragments and avoid potential bias, we lowered the parameter "seedSearchStartLmax" to 10 (from default 50), which limits the maximal fragmented length to 10 bases. Reads in exons were counted with HTSeq-count (v0.11.1) and TPM values were obtained by RSEM (v.1.3.3) using default parameters.

## Exclusion of SNPs from T2C mutation counting

VarScan's (v2.3.9) mpileup2snp command was used with the options minVarFreq="0.4" and minCov="10" to obtain SNPs from the BAM files from all unlabeled samples (uniquely mapped reads only). We created one pileup file containing only T2C and A2G mutations, which was used later on to exclude those SNPs from being erroneously counted as 4sU labeling-induced T2C conversion.

## T2C mutation counting and normalization

Mutations with respect to the reference sequence in each read were determined using a custom C program. Since the primary read maps to the reference sequence and the secondary read to the reverse complement, we searched for T2C and A2G mutations, respectively. To quantify the sequencing error rate $p_e$, we searched for A2G

mutations in the primary read and T2C mutations in the secondary read. Mutations were separately counted in for exons and the intronic region. For each region, the ratio between all T2C mutations in reads mapping to that region and all Ts in reads mapping to that region was computed. To confirm the validity of our metabolic labeling sequencing data, we repeated the alignment and T2C counting with SLAM-DUNK, which is an established pipeline that aligns only on the 3' UTR. When comparing labeling data from our custom C program to SLAM-DUNK, we find very high agreement (Spearman $r$ >0.9) from 30 min labeling time onwards (see Appendix Fig. S1). We conclude that our data analysis pipeline matches the standard of published metabolic labeling analysis pipelines.

The conversion rate corresponding to the random 4sU incorporation rate into transcripts was estimated using a two step process. For each sample, a frequency distribution was calculated indicating the frequency of $a_{kn}$ of $k$ mutations in reads with $n$ Ts. We then generated an initial estimate for $p_{conv}$ using the expectation-maximization (EM) algorithm described in (Jürges et al, 2018). Subsequently, we fitted a binomial mixture model to the observed mutation frequencies using the output of the EM algorithm and the sequencing error rate as inputs for $p_{conv}$ and $p_e$, respectively. An overview of the conversion rates can be seen in Fig. EV1E. Finally, the ratio of new to total mRNA was obtained from the T2C/T ratio by dividing with the estimated conversion rate. The ratio of new to total mRNA (per exon or for all intronic regions of a gene) was used later on as input for the kinetic model (see Fig. 1A, bottom, F).

## Additional data analysis

Genes were defined as expressed if they had an average TPM value ≥1.5 in at least one subcellular fraction. Membrane-to-cytosol enrichment was equal to the fold change between the average TPM values of membrane and cytosolic samples. To define membrane-bound and cytosolic mRNAs we used membrane enrichment cutoffs ≥3.0 and ≤1.5, respectively, for all expressed mRNAs. Fitting was performed throughout with the python package lmfit (v1.0.3) using the Levenberg-Marquardt "least-squares" algorithm to minimize the chi-squared unless otherwise specified. Gene set enrichment analyses were performed with fgsea (v1.28.0) with gene set sizes limited from 100 to 600. All rates used to rank the genes in the GSEAs were taken from the 3-step model to ensure intrinsic consistency, except the membrane decay rate, which was taken from the 4-step model. For general data analysis purposes, Python 3.9 and R v4.3.2 were used. Read mapping, counting, T2C data processing and model fitting was implemented in Snakemake (Mölder et al, 2021).

## Minimal effect of metabolic labeling on the transcriptome

To assess the effect of the metabolic labeling on transcriptional output, we performed a differential gene expression analysis on RNA count data from whole-cell extracts using DESeq2 (v1.40.2). Two different 4sU concentrations were used: 500 μM (from 0 min to 180 min) and 100 μM (from 60 min to 180 min). Comparing all subsequent time points to the $t = 0$ time point for the high 4sU dose, we found that up until 40 min no genes, at 60 min one gene (*Kantr*) and from 120 min onwards, more than 250 genes were

differentially expressed (see Fig. EV1A, log2FC >0.5 and adjusted $P$ value <0.05). We decided to exclude all samples with 500 µM 4sU labeling for more than 60 min and keep samples using 500 µM labeling up until and including 60 min. Accordingly, *Kantr* was removed from subsequent analyses. For samples labeled with 100 µM 4sU, the 120 min and 180 min time points were compared to the 60 min time point (as no control measurements were done in this batch). No genes were found to be differentially expressed (see Fig. EV1A, log2FC >0.5 and adjusted $P$ value <0.05). As the higher dosage already has a minimal effect at 60 min, it can be assumed that the lower dosage leaves the cells unperturbed at 60 min and hence the 60 min 100 µM samples are a valid control. As the lowest adjusted $p$ value in the comparisons for the low dosage was 0.6, it can be assumed that the transcriptional state of the cells does not change during the 180 min labeling duration with 100 µM 4sU. All in all, this shows a negligible effect of 4sU labeling on the transcriptome in our samples.

## Quantifying relative mRNA abundance of subcellular compartments

To get a biologically accurate quantification of the RNA expression ratio between different subcellular compartments, we need to quantify how much RNA is present in those compartments. This allows us to constrain the parameter space by steady-state expression ratios and to reconstruct whole-cell dynamics from subcellular kinetic parameters. Under the assumption that, in the gene space containing all expressed genes, the RNA expression vector of the whole-cell extract can be approximated by summing the RNA expression vectors of the three subcellular fractions with a corresponding factor for each, we fitted the model $a \cdot \text{TPM}_{\text{nuc}} + b \cdot \text{TPM}_{\text{cyto}} + c \cdot \text{TPM}_{\text{mem}} = 1 \cdot \text{TPM}_{\text{whole-cell}}$ to the whole-cell RNA expression with the constraint $a + b + c = 1$. Only highly expressed genes were included ($\text{TPM}_{\text{whole-cell}} > 50$, $n = 2725$). RNA expression vectors (and their errors) were the mean (and the standard error) across all replicates and time points of the corresponding fraction in gene space. The inverses of the squared standard errors were used as weights in the fitting procedure for both the independent and the response variables. The "ODR" function from function scipy (v1.11.2) was used to perform the total least-squares fit.

This resulted in the following values for the relative abundances: nuclear factor = 0.39, cytosolic factor = 0.15 and membrane factor = 0.45 (see Fig. 1D). Relative errors on the fitted parameters were equal to or smaller than 2%. It is of note that the cytosolic abundance is the lowest. The averaged subcellular TPM counts are multiplied by their corresponding factor before calculating the steady-state ratios that are used to constrain the parameter space in the fits (Box 1).

## Model fitting of subcellular mRNA dynamics

Summarizing different models and genes, the following kinetic rates were estimated: transcript elongation rate, pre-mRNA processing rate, nuclear export rate, nuclear decay rate, cytosolic decay rate, cytosolic transport rate and membrane decay rate (see Box 1). The half-lives, calculated from those rates, were primarily shown in the results section. Per gene, the analytical solutions of the rescaled system, shown in Box 2, were used to fit all corresponding kinetic rates simultaneously to the subcellular,

time-resolved labeling data. More specifically, the mean share of new to total mRNA for exons and introns across replicates was used as input data for the fit (see example gene with data from cytosolic compartment in Fig. 2B). Each data point was weighted by the inverse of a mixed standard error across replicates, with the mixed standard error consisting half of the standard error per time and compartment and half of the standard error averaged over time per compartment. Acting as a variance-stabilization mechanism, the minimum of the standard error for each exon was set to 0.06 (the mean standard error across all exons at $t = 15$ min). Per gene, there is one set of subcellular parameters shared by all exons, with each exonic fit curve being time-delayed by the elongation rate proportionately to the distance to 3' end. As parameter boundaries a minimum of 0.001 min$^{-1}$ and a maximum of 2 min$^{-1}$ was set for all kinetic rates (for the nuclear decay rate the minimum was set to 1e-06 min$^{-1}$). For the elongation rate, minimum and maximum boundaries were set to 0.1 kb/min and 10 kb/min. To further constrain the parameter space, we used the measured mRNA expression ratios via the equations shown in Box 1 and the "expr" keyword from lmfit. The upper and lower limits between parameter ratios were given by the steady-state ratios plus and minus five times its standard error. Taking *Tfrc* as an example, its membrane/cytosol expression ratio (plus/minus standard error) is given by $\frac{\text{rel}_{\text{mem}} \cdot \text{TPM}_{\text{mem}}}{\text{rel}_{\text{cyto}} \cdot \text{TPM}_{\text{cyto}}} = \frac{0.45 \cdot 200.6}{0.15 \cdot 24.0} = 25.1 \ (\pm 1.1)$, where the TPM values were multiplied with their corresponding relative mRNA abundance to obtain an absolute steady-state expression ratio. Hence, the lower and upper limit of the ratio $\frac{k_3}{\gamma_4}$ were set to $25.1 - 5 \cdot 1.1 = 19.6$ and $25.1 + 5 \cdot 1.1 = 30.6$, respectively (in other words: $k_3$ has to be at least 19.6 times larger than $\gamma_4$, but not more than 30.6 times). Per gene, the kinetic parameters were fit in logarithmic space 200 times with random initial conditions using a computationally inexpensive local fitting method. Out of the ten lowest chi-squared fits, the result with parameter values farthest from the parameter boundaries was chosen as best and final fit. The quality control involved three steps: genes were excluded if the best fit showed (a) reduced chi-squared >4, (b) relative standard deviation of the ten least chi-squared fits of any parameter >0.05, or (c) nuclear or cytosolic (cytosol-localized) or nuclear and membrane (membrane-localized) parameter values at the maximum allowed boundary (see Figs. EV2A–C). Standard errors on the parameters were calculated as the square root of the diagonal elements of the inverse Hessian matrix (1$\sigma$ uncertainty output by lmfit). Relative standard errors were the standard errors divided by their corresponding parameter values. 75% of parameter estimates have a relative error smaller than 19% (see Fig. EV2D). Rank correlations between parameters are shown in Fig. EV2E. The cell cycle rate of roughly 14 h for mESCs (Herzog et al, 2017; Waisman et al, 2019) confounds our rate estimation only minimally, as the estimated rates are typically an order of magnitude smaller than the average cell cycle.

## Discriminating models with and without nuclear mRNA degradation

The role and contribution of nuclear decay of poly(A) mRNA has been under discussion (Schmid and Jensen, 2018). In our analysis, we fitted four models to the poly(A) selection-based metabolic labeling sequencing data: the 3-step and 4-step models, as described earlier, both once including and once excluding a nuclear decay parameter. To test if the measured mRNA dynamics

**Box 2.   Analytical solutions of the rescaled system and their usage in the fitting procedure**

The analytical solutions to the rescaled ODE system (see Box 1, top right) are shown. For intron-containing genes ($n = 8548$), solutions of the first, second, third and fourth step are used to fit nuclear pre-, nuclear mature, cytosolic and membrane-bound mRNA, respectively, with the fourth step being used only for membrane-localized and undefined transcripts ($n = 1726$). As indicated on the top right, we incorporate the transcript elongation rate $v$ to time-delay the solutions: for mature mRNA we fit on exon level and $d$ is the distance of the exon to the 3' end, and for pre-mRNA we fit all intronic regions together and $d$ is the gene length times

0.1, with this heuristic factor being the mean relative exon distance to 3' end across genes weighted by expression. $\delta$ is the overall delay of 5 min until any labeling is experimentally observed. The modified time is set to 0 for all negative values. For intron-less genes or genes with reliable T2C data present only for one exon ($n = 1314$), the first, second and third solutions are used to fit nuclear mature, cytosolic and membrane-bound mRNA, respectively, with the third step being used only for membrane-localized or undefined transcripts ($n = 318$). In this case, transcript elongation and pre-mRNA processing rates are not fit and $\tilde{t} = t - \delta$.

Analytical solutions of the rescaled system    BOX 2    Modified time: $\tilde{t} = t - d/v - \delta$

$$\tilde{x}_1(\tilde{t}) = 1 - e^{-r_1 \tilde{t}}$$

$$\tilde{x}_2(\tilde{t}) = \frac{r_1 \left(1 - e^{-r_2 \tilde{t}}\right) + r_2 \left(e^{-r_1 \tilde{t}} - 1\right)}{r_1 - r_2}$$

$$\tilde{x}_3(\tilde{t}) = 1 - e^{-r_3 \tilde{t}} + \frac{r_3 \left(e^{-r_2 \tilde{t}} - e^{-r_3 \tilde{t}}\right)}{r_2 - r_3} - \frac{r_2 r_3 \left((r_1 - r_2) e^{-r_3 \tilde{t}} + (r_2 - r_3) e^{-r_1 \tilde{t}} + (r_3 - r_1) e^{-r_2 \tilde{t}}\right)}{(r_1 - r_2)(r_1 - r_3)(r_2 - r_3)}$$

$$\tilde{x}_4(\tilde{t}) = 1 - e^{-r_4 \tilde{t}} - r_2 r_3 r_4 \left(\frac{e^{-r_2 \tilde{t}}}{(r_1 - r_2)(r_2 - r_3)(r_2 - r_4)} + \frac{e^{-r_3 \tilde{t}}}{(r_1 - r_3)(r_3 - r_2)(r_3 - r_4)} + \frac{e^{-r_4 \tilde{t}}}{(r_1 - r_4)(r_4 - r_2)(r_4 - r_3)} - \frac{e^{-r_1 \tilde{t}}}{(r_1 - r_2)(r_1 - r_3)(r_1 - r_4)}\right) + \frac{r_4 \left(e^{-r_3 \tilde{t}} - e^{-r_4 \tilde{t}}\right)}{r_3 - r_4} - \frac{r_3 r_4 \left((r_2 - r_3) e^{-r_4 \tilde{t}} + (r_3 - r_4) e^{-r_2 \tilde{t}} + (r_4 - r_2) e^{-r_3 \tilde{t}}\right)}{(r_2 - r_3)(r_2 - r_4)(r_3 - r_4)}$$

are better explained by a model including nuclear decay, we compared the Bayesian Information Criterion (BIC) from fit results of models excluding ($\gamma_2 = 0$) and including ($\gamma_2 \neq 0$) a nuclear decay parameter. We calculated the BIC difference between the null model excluding and the extended model including nuclear decay for all 9809 transcripts (see Fig. EV5A). For cytosol-localized transcripts, the BIC values from the 3-step models were compared. For undefined and membrane-localized transcripts, the BIC values from the 4-step models were compared. A BIC difference >10 corresponds to a Bayes factor >150 and shows very strong evidence for the extended model, including nuclear decay (Raftery, 1995). If further the standard error on the nuclear decay parameter is smaller than the parameter value itself, i.e., the estimation uncertainty excludes the possibility of the nuclear decay being 0, we chose the model including nuclear decay for that particular transcript. Based on these criteria, 579 (6% of successfully fitted) transcripts were modeled with nuclear degradation, with the ratio of cytosol- and membrane localization being similar to the ratio seen in all other transcripts. To test if certain biological processes are enriched in the nuclear decaying transcripts, a GO analysis including Wikipathways of the 579 transcripts was performed with GProfiler2 (v.0.2.3) taking all successfully fitted transcripts as background. The results from the over-representation analysis are shown in Fig. EV5C.

## Transcription inhibition-derived subcellular half-lives

To validate our metabolic labeling-derived subcellular flow rates, we inhibited transcription in mESCs using flavopiridol for 0, 30, 60, 120, and 240 min followed by subcellular fractionation and RNA sequencing (as specified above). Reads were aligned with STAR (v2.7.6) and counted with RSEM (v1.3.3). Library sizes were normalized using genes with known half-lives greater than 14 h (Herzog et al, 2017). Samples of each time series were further normalized to the corresponding $t = 0$ time point (Appendix Fig. S2C). We fitted a simple exponential decay model per compartment for roughly 10,000 genes, giving the aggregated subcellular turnover rates (see example in Appendix Fig. S2D), in contrast to the rates of transition from one compartment to the next. Transcription inhibition-derived half-lives are shown in Fig. 5C and Appendix Fig. S2E.

## Model-derived half-lives

To compare the transitional, metabolic labeling-derived half-lives to the aggregated, flavopiridol-derived half-lives, we generated model-derived pulse-chase data from the transitional rates using the equations in Box 2 and the fact that the share of old mRNA is given by one minus the share of new mRNA. To the model-derived pulse-chase data ($n = 9862$) we fitted a simple exponential decay model (see example fit for a single gene in Appendix Fig. S2A). For the subcellular compartments, the equations in

Box 2 can be used as is, while for the whole-cell compartment the subcellular solutions are summed with their corresponding relative RNA abundance factors. Model-derived aggregated half-lives (subcellular and whole-cell) are used in Appendix Fig. S2B and Fig. 5E. Model-derived whole-cell half-lives are also used in Fig. 5A.

## Targeting signal and gene group annotations

Signal peptide and transmembrane helix annotations were downloaded from Ensembl Biomart. We defined tail-anchored proteins as those transmembrane domain-containing proteins lacking signal peptide, for which the first transmembrane helix started 50 or less amino acids from the C-terminus. A list of mitochondrial proteins encoded in the nuclear DNA was obtained from Mitocarta 2. Transcription factors were defined by a list of mESC transcription factors from the Gifford lab.

# Data availability

Metabolic labeling RNA-seq data are accessible from the NCBI Gene Expression Omnibus (GEO) under the accession number GSE252199. Flavopiridol transcription inhibition RNA-seq data are accessible from the NCBI GEO under the accession number GSE256335. Data processing and model fitting scripts are publicly available on GitHub at https://github.com/steinbrecht/subcellular-SLAM.

The source data of this paper are collected in the following database record: biostudies:S-SCDT-10_1038-S44320-024-00073-2.

# Peer review information

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

## Acknowledgements

The authors thank Stefan L. Ameres for help with the experimental design. The authors acknowledge the Genomics technology platform of the Max Delbrück Center for Molecular Medicine for high-throughput sequencing. Computation has been performed on the HPC for the Research cluster of the Berlin Institute of Health. Figure 1A was created with BioRender. We thank members of the Blüthgen and Landthaler labs for fruitful discussions. Funding was provided by Deutsche Forschungsgemeinschaft (grants RTG2424 CompCancer and SFB/TRR 186 to DS).

## Author contributions

**David Steinbrecht**: Software; Formal analysis; Validation; Investigation; Visualization; Methodology; Writing—original draft; Writing—review and editing. **Igor Minia**: Formal analysis; Investigation; Methodology; Writing—review and editing. **Miha Milek**: Software; Formal analysis; Investigation; Methodology. **Johannes Meisig**: Data curation; Software; Formal analysis. **Nils Blüthgen**: Conceptualization; Data curation; Software; Formal analysis; Supervision; Funding acquisition; Visualization; Project administration; Writing—review and editing. **Markus Landthaler**: Conceptualization; Supervision; Funding acquisition; Methodology; Project administration; Writing—review and editing.

In addition to the CRediT author contributions listed above, the contributions in detail are:

ML and NB conceived the study. ML and IM designed the experiments. IM carried out all of the experimental work. MM and JM performed bioinformatic data analysis. DS and JM performed modeling. DS prepared the manuscript and the majority of figures. IM, NB, ML, and MM critically revised and contributed to the manuscript.

Source data underlying figure panels in this paper may have individual authorship assigned. Where available, figure panel/source data authorship is listed in the following database record: biostudies:S-SCDT-10_1038-S44320-024-00073-2.

## Funding

## Disclosure and competing interests statement

The authors declare no competing interests.

# Expanded View Figures

**Figure EV1.   Overview of mRNA expression and T2C labeling data.**

(A) Differential expression analysis of whole-cell extract RNA sequencing data. Volcano plots showing BH-adjusted *p* values against fold changes logarithmized to base 2. Facets show subsequent 4sU labeling times tested against either t=0 min labeling for the high 4sU dosage (500 μM) or t = 60 min labeling for the low 4sU dosage (100 μM). Differentially expressed genes (padj < 0.05 & log2fc > 0.5, indicated by horizontal and vertical dashed lines) are shown as red dots and stable genes are shown as black dots. Brackets indicate that both subcellular and whole-cell samples of the time point and 4sU concentration tested in the facet were excluded from further analysis. Considering only data included in our main analysis, only one gene, *Kantr*, is differentially expressed and was removed from subsequent analyses. (B) Gene expression in subcellular fractions. Histograms show TPM values averaged over all time points and replicates for nuclear, cytosolic and membrane fractions. Vertical dashed lines indicate an average TPM of 1.5. If a gene has an average TPM >1.5 in at least one subcellular fraction, it is considered expressed. (C) Mitochondrial contamination of the nuclear fraction. The ratio of the summed counts from all mt-DNA-encoded over all nuclear-encoded transcripts is shown for all samples (black dots) split by fraction. Orange lines depict the median across samples. (D) Box plot of ratio between membrane and cytosolic expression (membrane enrichment) with transcripts classified by encoded TS as in Fig. 4C. Classification from left to right: cytosol-localized transcripts with no TS (n = 6262), nuclear DNA-encoded mitochondrial proteins (n = 763 with 661 being cytosol-localized), transcripts with tail-anchored transmembrane proteins (n = 78), membrane-localized transcripts with no known TS (n = 531), transcripts encoding signal peptides (n = 310) or transmembrane helices (n = 956) or both (n = 47). Center lines of box plots depict the median values. Notches show the 95% confidence interval of median values acquired through boot-strapping (n = 1000). Lower and upper hinges of box plots correspond to the 25th and 75th percentiles, respectively. (E) Estimates of T2C conversion rate. Conversion rates of individual samples are plotted over 4sU labeling time. Conversion rates were estimated fitting a Binomial mixture model to T and T2C count data of each sample. Color indicates compartment. Shape indicates 4sU concentration. Conversion rate increases for high dosage (roughly two-fold increase from 15 to 60 min), but stays constant for low dosage. (F) Comparison between samples labeled with 500 μM or 100μM 4sU for 60 min. Boxes show the share of new to total mRNA in subcellular compartments with colors indicating specific replicates. Center lines of box plots depict the median values. Lower and upper hinges of box plots correspond to the 25th and 75th percentiles, respectively. Although 4sU conversion rates are different between samples, see (D), the share of new mRNA is similar. (G) Principal component analysis of the T2C labeling data using 2,000 most variable genes. Samples cluster by exposure time to 4sU rather than replicate, with nuclear pre-mRNA being separated from other compartments. Source data are available online for this figure.

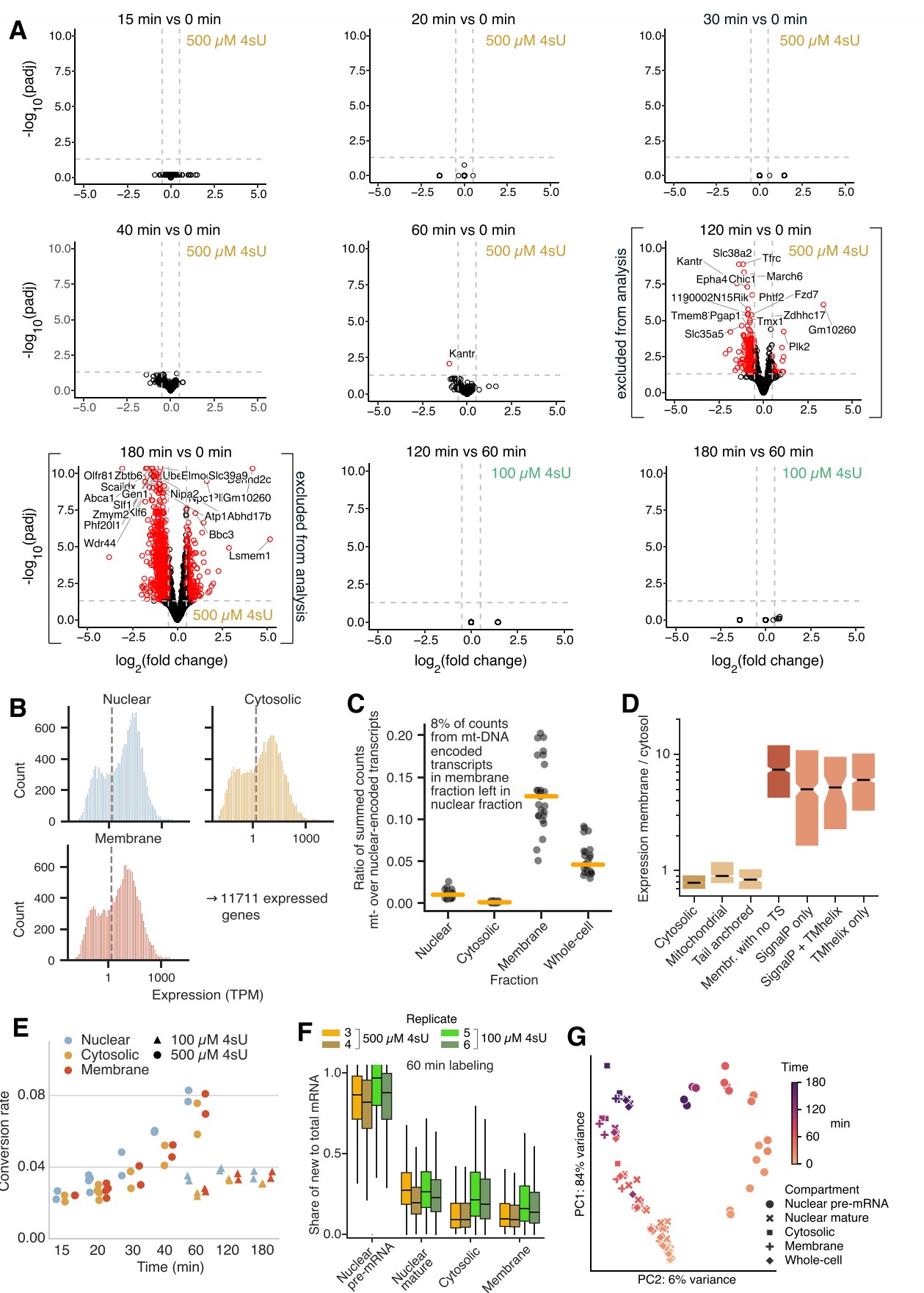

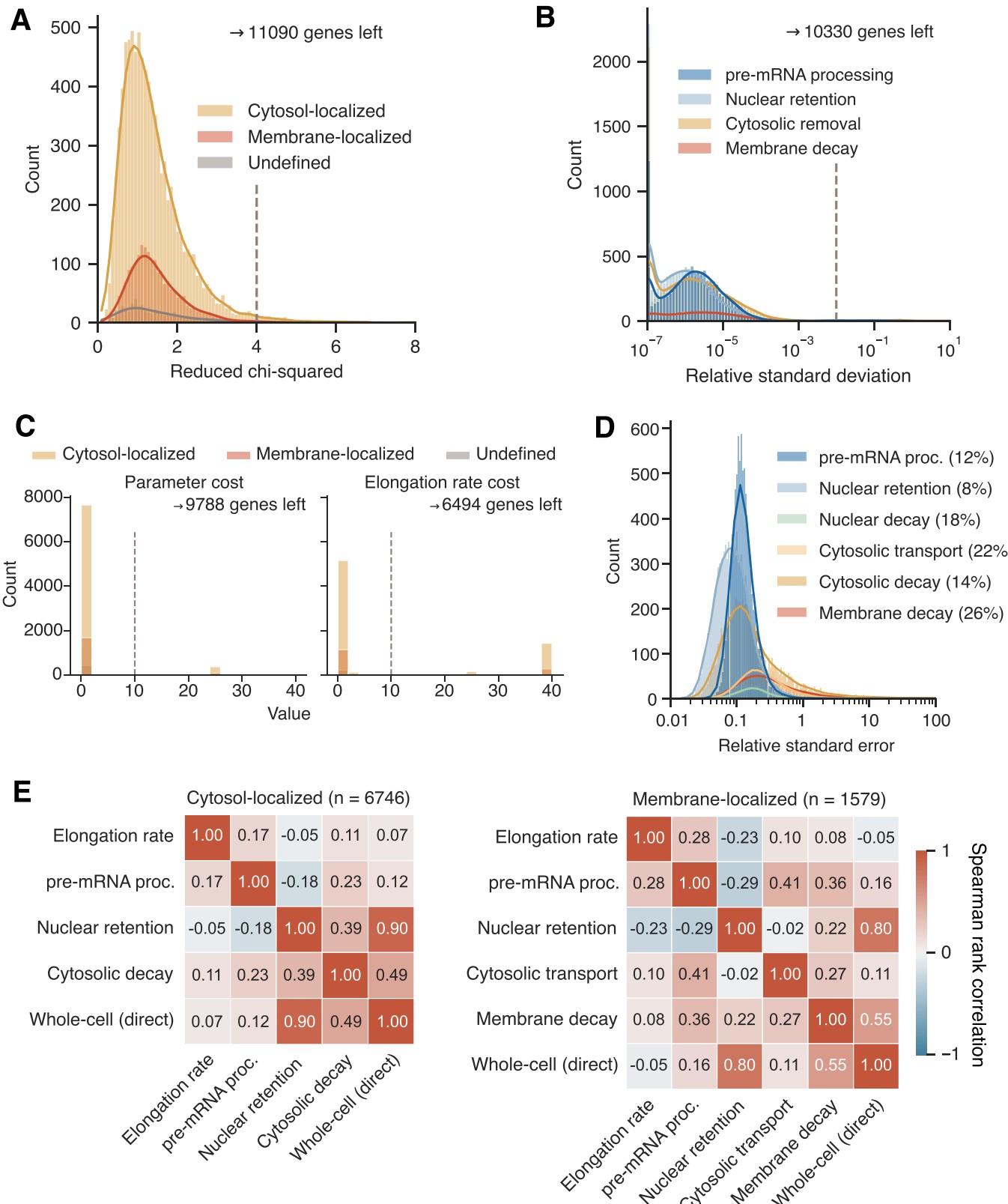

**Figure EV2.  Quality control of fit results.**

(**A**) First quality control step. Histograms of reduced chi-squared, which is minimized during fitting (optimal value is 1). Vertical dashed line shows cutoff of 4. If the best fit for a transcript has a value higher than the cutoff, the transcript is excluded. (**B**) Second quality control step. Histograms of relative standard deviation, calculated by dividing the standard deviation of the ten best-fit results by the value of the best fit. All values < 1e-07 were set to 1e-07. Vertical dashed line shows cutoff of 0.05. If the relative standard deviation for a transcript has a value higher than the cutoff, the gene is excluded. (**C**) Third quality control step. Histograms of the boundary cost for all kinetic parameters (left) and elongation rate (right). Boundary cost is near 0 if fit value is far away from upper and lower limits and increases drastically as the value approaches the allowed limits. A high boundary cost therefore indicates that the fit was stuck at maximum or minimum allowed value. Vertical dashed line shows cutoff of 10. If the best fit for a transcript has a parameter cost higher than the cutoff, the transcript is excluded. Transcripts with a elongation rate cost higher than the cutoff are only excluded for analyses specifically regarding the elongation rate. Colors indicate mRNA localization. (**D**) Accuracy of parameter estimations. Histograms of the relative standard error colored by parameter. Relative standard errors are the square root of the diagonal elements of the inverse Hessian matrix ($1\sigma$ uncertainty output by lmfit) divided by the corresponding parameter value. Median relative standard errors of each parameter are shown in brackets as percentages in the figure legend. 75% of parameter estimates have a relative error smaller than 19%. (**E**) Heatmaps showing the Spearman rank correlation between kinetic parameters for cytosol- (left) and membrane-localized (right) transcripts. Source data are available online for this figure.

                                                    

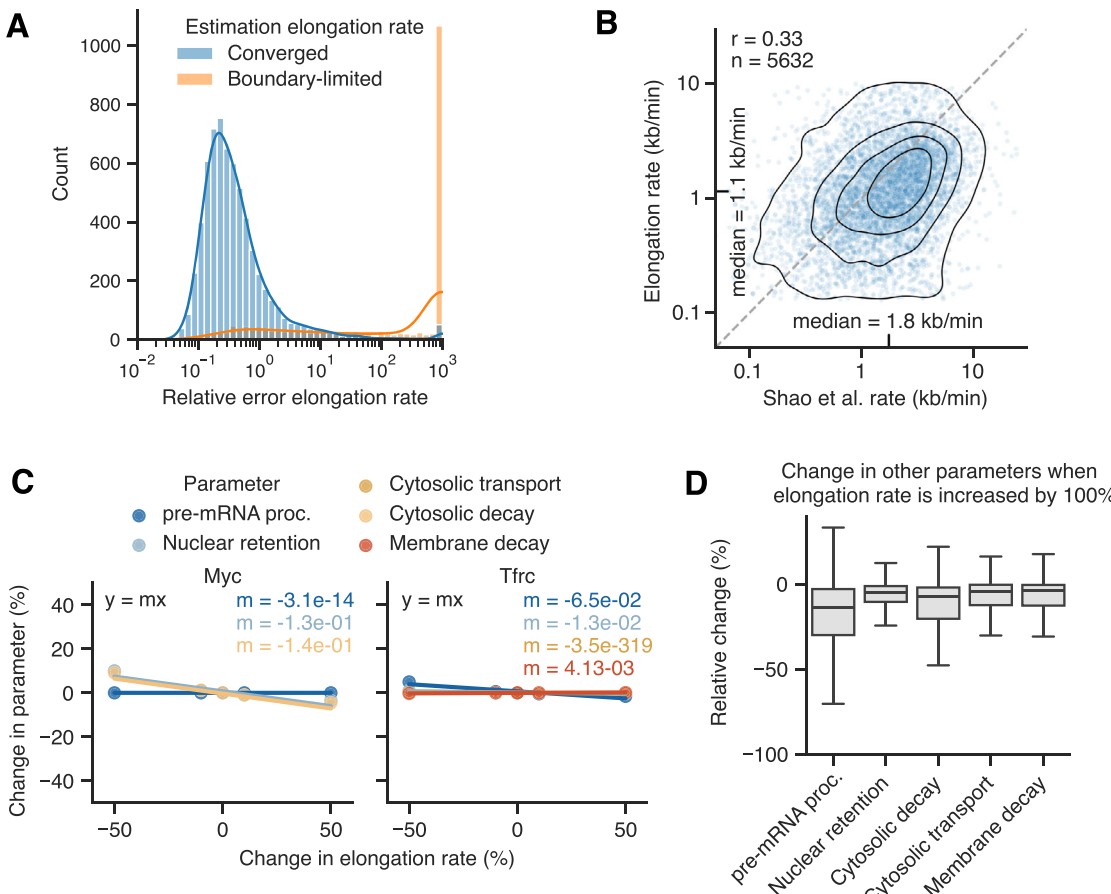

**Figure EV3. Quality control of the transcript elongation rate.**

(A) Accuracy of elongation rate estimations. Histogram of the relative standard error of the elongation rate. Relative standard errors are the square root of the diagonal elements of the inverse Hessian matrix (1σ uncertainty output by lmfit) divided by the corresponding parameter value. 75% of the converged fits (n = 6496) have a relative error smaller than 63% with an overall median of 30%. For boundary-limited estimates (n = 2007) errors were high or could not be determined. For illustration purposes, here, relative standard errors containing infinities or NAs were set to 1000. (B) Comparison of estimated transcription elongation rate to published data (Shao et al, 2022). Scatterplot shows converged elongation rates and from Shao et al measured in serum-naive state mESCs. Spearman rank correlation is shown on top left. Black lines are 2-dimensional KDE to indicate density of points. Dashed, gray line is the identity line. (C) Sensitivity analysis for the elongation rate parameter. Regression plots for transcripts of two exemplary genes, *Myc* and *Tfrc*, are shown. For all multiple-exons transcripts that do not show nuclear decay, we repeated the fitting procedure fixing the elongation rate at -50%, -10%, +10% and +50% of its best-fit value and initializing with the best-fit values of the remaining parameters. Then, for each transcript and parameter, we fitted a linear regression $y = mx$, where y is the change in the fitted parameter and x is the fixed relative change of the elongation rate. The slope m gives a measure of how much a parameter is influenced by changes in the elongation rate. (D) Overview of the sensitivity analysis for the elongation rate parameter. Box plots show the distribution of the slopes from linear regression (see D) for all fitted parameters and across all transcripts with multiple exons (n = 7935). Slopes were multiplied by 100 to represent relative change in percent. When the elongation rate is increased by 100%, the median change in the other parameters ranges from 4% for the membrane decay to 14% for the pre-mRNA processing rate. Transcripts that show nuclear degradation (n = 566) were excluded here due to simplicity reasons. Source data are available online for this figure.

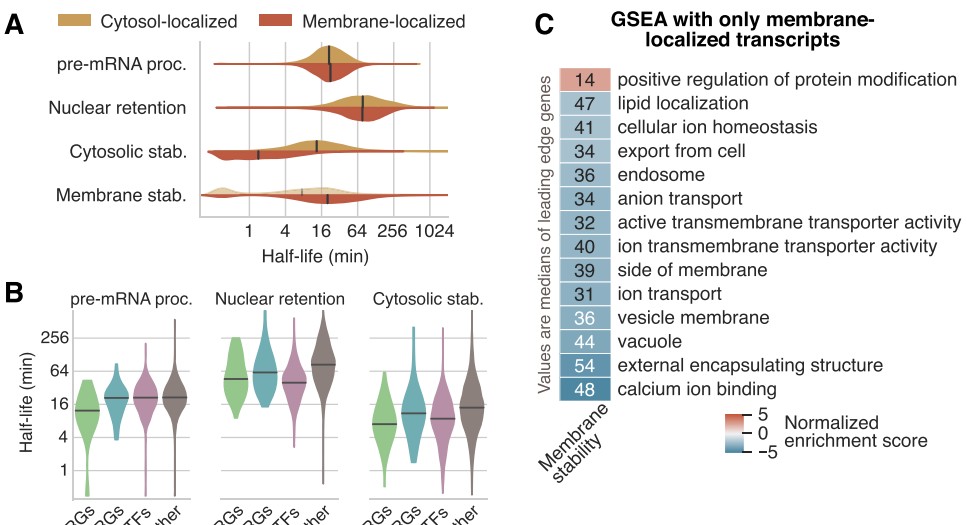

**Figure EV4.   Kinetic differences between different gene groups.**

(**A**) Violin plots of pre-mRNA processing, nuclear retention, cytosolic and membrane half-lives for cytosol- (n = 7677) and membrane-localized (n = 1693) transcripts. (**B**) Violin plots of pre-mRNA processing, nuclear retention and cytosolic half-lives for primary response genes (PRGs, n = 56), secondary response genes (SRGs, n = 37), transcription factors (TFs, n = 709) and all other (n = 6875) transcripts. Only cytosol-localized transcripts were included. Definitions of PRGs and SRGs are taken from (Uhlitz et al, 2017). For definition of TFs, see Methods. (**C**) GSEA based on the GO using only membrane-localized transcripts and ranking by membrane decay. All significant terms (BH-adjusted *p* values <0.05) are shown. Color indicates normalized enrichment score. Annotated values are the median half-lives of leading edge genes of each term. Source data are available online for this figure.

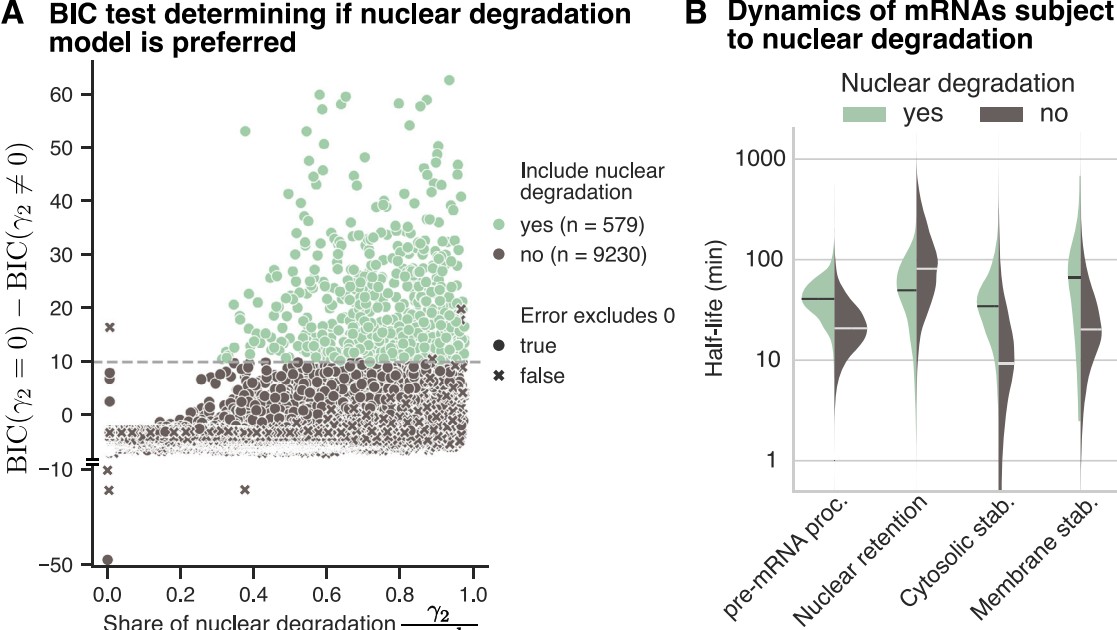

**A** BIC test determining if nuclear degradation model is preferred

**B** Dynamics of mRNAs subject to nuclear degradation

**C** GO analysis of nuclear decaying transcripts

| Term name (intersection size) | BH-adjusted p-value |
| --- | --- |
| Base-exision repair (9) | 0.05 |
| DNA replication maintenance of fidelity (11) | 0.05 |
| Negative regulation of DNA metabolic process (16) | 0.05 |
| DNA damage response (60) | 0.05 |
| Toll like receptor signaling (7) | 0.04 |

**Figure EV5. Nuclear degradation of polyadenylated mRNAs.**

(A) Comparison between models excluding and including the nuclear degradation parameter $\gamma_2$. Scatterplot shows the value difference between the Bayesian Information Criterion (BIC) from fit results of models excluding ($\gamma_2 = 0$) and including ($\gamma_2 \neq 0$) a nuclear decay parameter over the share of nuclear degradation at overall nuclear dynamics from the $\gamma_2 \neq 0$ model (n = 9809). For cytosol-localized transcripts, the BIC values from the 3-step models are compared. For undefined and membrane-localized transcripts, the BIC values from the 4-step models are compared. Based on (Raftery, 1995), the model including nuclear degradation is chosen if the BIC difference exceeds 10 (colored green), otherwise the model without nuclear degradation is chosen (colored black). Further selection is based on $\gamma_2 - s_{\gamma_2} > 0$ with $s_{\gamma_2}$ being the standard error on the nuclear decay parameter. This test (true or false indicated by shape) checks that the 0 value is not included within the standard error and only if true the model with nuclear degradation is chosen. (B) Violin plots showing pre-mRNA processing, nuclear retention, cytosolic stability and membrane stability half-lives for transcripts modeled with (n = 579) and without (n = 9230) nuclear decay. Nuclear retention half-life is calculated as $\tau_2 = \frac{\ln 2}{k_2 + \gamma_2}$. Center lines of violin plots depict the median values. (C) Results from a GO analysis of all transcripts modeled with nuclear decay. All significant results (BH-corrected *p* value <0.05) are shown. Source data are available online for this figure.

