## [Peer Review File · Molecular Systems Biology]

Subcellular mRNA kinetic modeling reveals nuclear retention as rate-limiting

David Steinbrecht, Igor Minia, Miha Milek, Johannes Meisig, Nils Blüthgen, and Markus Landthaler

Corresponding author(s): Markus Landthaler (markus.landthaler@mdc-berlin.de) , Nils Blüthgen (nils.bluthgen@charite.de)

Review Timeline:

Submission Date:	23rd Feb 24
Editorial Decision:	3rd Apr 24
Revision Received:	3rd Jul 24
Editorial Decision:	29th Jul 24
Revision Received:	17th Oct 24
Accepted:	22nd Oct 24

Editors: Maria Polychronidou and Jingyi Hou

Transaction Report:

3rd Apr 2024

Manuscript Number: MSB-2024-12284

Title: Subcellular mRNA kinetic modeling reveals nuclear retention as rate-limiting

Dear Markus,

Thank you again for submitting your work to Molecular Systems Biology. We have now heard back from the three reviewers who agreed to evaluate your study. As you will see below, the reviewers think that the study seems like a relevant contribution to the field. However, they raise several concerns, which we would ask you to address in a revision.

I think that the reviewers' recommendations are rather clear and I therefore see no need to repeat all the comments listed below. Of note, reviewer #1 points out that additional controls and orthogonal validations need to be performed to better support the main conclusions. Moreover, reviewer #2 raises several technical concerns related to the data analysis and the model that need to be carefully addressed.

All issues raised by the reviewers would need to be satisfactorily addressed. As you may already know, our editorial policy allows in principle a single round of major revision. It is therefore essential to provide responses to the reviewers' comments that are as complete as possible. If you have any questions or if you would like to discuss your revision plan with me, please feel free to get in touch.

On a more editorial level, we would ask you to address the following points:

- Please provide a .doc version of the manuscript text (including legends for main Figures and EV Figures) and individual production quality figure files for the main Figures and EV Figures (one file per figure).
- Please include 5 keywords.
- We have replaced Supplementary Information by the Expanded View (EV format). In this case (unless the number of EV figures becomes > 6 during revision), all additional figures can be provided as EV Figures. Please provide one file per EV Figure. Their legends should be included in the manuscript text. For detailed instructions regarding expanded view please refer to our Author Guidelines: .
- In case you need to include > 6 additional figures, they should all be provided in a PDF called Appendix. Please include a Table of Contents (with page numbers) in the beginning of the Appendix. Appendix figures should be labeled and called out as: "Appendix Figure S1, Appendix Figure S2..." etc. Each legend should be below the corresponding Figure in the Appendix. For detailed instructions regarding expanded view please refer to our Author Guidelines: .
- Supplementary Data File should be provided as Dataset EV1. Please include a description of the table/dataset in a separate sheet within the file.
- Please provide a "standfirst text" summarizing the study in one or two sentences (approximately 250 characters), three to four "bullet points" highlighting the main findings and a "synopsis image" (exactly 550px width and max 400px height, jpeg or png format) to highlight the paper on our homepage.
- All Materials and Methods need to be described in the main text. We would encourage you to use 'Structured Methods', our new Materials and Methods format. According to this format, the Material and Methods section should include a Reagents and Tools Table (listing key reagents, experimental models, software and relevant equipment and including their sources and relevant identifiers) followed by a Methods and Protocols section in which we encourage the authors to describe their methods using a step-by-step protocol format with bullet points, to facilitate the adoption of the methodologies across labs. More information on how to adhere to this format as well as downloadable templates (.doc or .xls) for the Reagents and Tools Table can be found in our author guidelines: . An example of a Method paper with Structured Methods can be found here:
- Please include a Data availability section describing how the data and code have been made available. This section needs to be formatted according to the example below:
The datasets and computer code produced in this study are available in the following databases:
 - Chip-Seq data: Gene Expression Omnibus GSE46748 (<https://www.ncbi.nlm.nih.gov/geo/query/acc.cgi?acc=GSE46748>)
 - Modeling computer scripts: GitHub (<https://github.com/SysBioChalmers/GECKO/releases/tag/v1.0>)
 - [data type]: [full name of the resource] [accession number/identifier] ([doi or URL or identifiers.org/DATABASE:ACCESSION])
- For data quantification: please specify the name of the statistical test used to generate error bars and P values, the number (n)

of independent experiments (specify technical or biological replicates) underlying each data point and the test used to calculate p-values in each figure legend. The figure legends should contain a basic description of n, P and the test applied. Graphs must include a description of the bars and the error bars (s.d., s.e.m.).

- Please include a "Disclosure & Competing Interests Statement" in the main text.

- The References should be formatted according to the Molecular Systems Biology reference style (i.e., ordered alphabetically and listing the first 10 authors followed by et al).

- When you resubmit your manuscript, please download our CHECKLIST (<https://bit.ly/EMBOPressAuthorChecklist>) and include the completed form in your submission.

Please note that the Author Checklist will be published alongside the paper as part of the transparent process (<https://www.embopress.org/page/journal/17444292/authorguide#transparentprocess>).

If you feel you can satisfactorily deal with these points and those listed by the referees, you may wish to submit a revised version of your manuscript. Please attach a covering letter giving details of the way in which you have handled each of the points raised by the referees. A revised manuscript will be once again subject to review and you probably understand that we can give you no guarantee at this stage that the eventual outcome will be favorable.

Kind regards,

Maria

Maria Polychronidou, PhD
Senior Editor
Molecular Systems Biology

We realize that it is difficult to revise to a specific deadline. In the interest of protecting the conceptual advance provided by the work, we recommend a revision within 3 months (2nd Jul 2024). Please discuss the revision progress ahead of this time with the editor if you require more time to complete the revisions. Use the link below to submit your revision:

IMPORTANT: When you send your revision, we will require the following items:

1. the manuscript text in LaTeX, RTF or MS Word format
2. a letter with a detailed description of the changes made in response to the referees. Please specify clearly the exact places in the text (pages and paragraphs) where each change has been made in response to each specific comment given
3. three to four 'bullet points' highlighting the main findings of your study
4. a short 'blurb' text summarizing in two sentences the study (max. 250 characters)
5. a 'thumbnail image' (550px width and max 400px height, Illustrator, PowerPoint or jpeg format), which can be used as 'visual title' for the synopsis section of your paper.

6. Please include an author contributions statement after the Acknowledgements section (see <https://www.embopress.org/page/journal/17444292/authorguide>)

7. Please complete the CHECKLIST available at (<https://bit.ly/EMBOPressAuthorChecklist>).

Please note that the Author Checklist will be published alongside the paper as part of the transparent process (<https://www.embopress.org/page/journal/17444292/authorguide#transparentprocess>).

See also figure legend guidelines: <https://www.embopress.org/page/journal/17444292/authorguide#figureformat>

9. Please note that corresponding authors are required to supply an ORCID ID for their name upon submission of a revised manuscript (EMBO Press signed a joint statement to encourage ORCID adoption).

(<https://www.embopress.org/page/journal/17444292/authorguide#editorialprocess>)

Currently, our records indicate that the ORCID for your account is 0000-0002-1075-8734.

Link Not Available

*** PLEASE NOTE *** As part of the EMBO Press transparent editorial process initiative (see our Editorial at <https://dx.doi.org/10.1038/msb.2010.72>), Molecular Systems Biology publishes online a Review Process File with each accepted manuscripts. This file will be published in conjunction with your paper and will include the anonymous referee reports, your point-by-point response and all pertinent correspondence relating to the manuscript. If you do NOT want this File to be published, please inform the editorial office at msb@embo.org within 14 days upon receipt of the present letter.

Reviewer #1:

Steinbrecht et al have used 4sU metabolic labeling, SLAM-seq, time series analysis and subcellular fractionation to measure RNA flow in mouse ESCs. As the authors note in the introduction, their work builds upon many other studies using a similar set of approaches. The major distinguishing factor between this work and others is the addition of the membrane fraction and an analysis of transcription elongation. The authors developed a model that relies on the assumption of RNA conservation throughout their analysis. A lot of assumptions are made without experimental validation and many of the key findings should be bolstered by orthogonal validation. Overall, this paper would be a good fit for MSB if the following concerns can be addressed.

Major concerns:

- 1) The estimation of transcription elongation rates is novel and interesting, but it needs more work. An elongation model provides a delay that aligns with the data, but the authors do not demonstrate that their extracted rates actually reflect transcription elongation. The authors should compare their rates to those measured by other approaches, either published or measured by the authors. They could also perturb transcription elongation through Pol II trigger loop mutations to see if their rates change as expected (as in PMID 33631106). Specific questions are:
 - a. Does the elongation rate estimation impact the measurements of downstream RNA flow rates?
 - b. Is transcription elongation the main reason for the time-delay? Could premature termination or something else explain some of the effect?
 - c. Do the estimated rates reflect real elongation rates? Jonkers et al estimate elongation rates in the same cell type as used in this study. How do the two datasets compare?
 - d. If Jonkers et al is unsuitable for a comparison, are there other published datasets? Could the authors measure the rates themselves? Perhaps their flavopiridol experiment or an extension of the experiment could even be used to generate rough estimates of transcription elongation rates.
- 2) A key assumption of the authors' model is that "the RNA expression in the whole cell can be reconstructed by summing the subcellular RNA expression with a corresponding factor for each". This assumption at first seems reasonable, however, it is not likely to hold up. Subcellular fractionation is imperfect, and much RNA is likely lost during the many wash steps, which could vary across the transcriptome. There's also the chance for RNA degradation to occur post-lysis that would also lead to RNA loss. Can the authors quantify how much RNA might be lost by sequencing wash steps and work out how robust the modeling and rates estimates are due to possible loss?
- 3) Another model assumption is that RNA processing occurs post-transcriptionally, but >60% of all RNA maturation occurs co-transcriptionally. How does co-transcriptional processing impact the authors' models?
- 4) The section on nuclear-encoded mitochondrial transcripts is confusing and has a lot of caveats. It is unclear how a transcript can be largely cytoplasmic but with similar kinetics as other transcripts aside from long nuclear half-lives. This would seem to make these transcripts largely nuclear residing. Furthermore, mitochondrial co-purification with the nucleus is well established for subcellular purification protocols. If the authors have mitochondrial contamination, the long nuclear half-lives for mitochondrial protein transcripts might be explained by this alone. Mitochondrial contamination was not monitored by the authors in their subcellular fractionation experiments, so the authors need to address this possibility.
- 5) The authors mention that a Bayesian approach would be required to distinguish between the 3 step and 4 step models. However, the suitability of various models with different numbers of parameter can be determined via other statistical approaches, such as use of the Akaike information criterion. Could the authors estimate which model is a better fit to each gene's data?
- 6) A major finding of the manuscript is that nuclear retention time is the rate-limiting step for RNAs as they spend most of their life spans in the nucleus. This is surprising. A microscopy-based approach, such as FISH, would be necessary to confirm this main finding, either at the single gene level or at the entire polyA transcriptome level.

Minor comments

1) Could the authors speculate on how species/cell type specific the authors' findings are?

Reviewer #2:

Review of the manuscript "Subcellular mRNA kinetic modeling reveals nuclear retention as rate-limiting" by Steinbrecht, Landthaler et al.

The present manuscript is a remarkable attempt to quantitatively dissect the mRNA life cycle into the processes of transcription, pre-mRNA maturation, nuclear export, cytosolic degradation, and transport/attachment to intracellular membranes (endoplasmic reticulum). To achieve this goal, the authors combine sophisticated experimental techniques such as subcellular fractionation and metabolic RNA labelling with 4-thiouridine in an RNA-seq time series experiment. While this is difficult enough, the biggest challenge is the data analysis, which requires advanced modelling techniques, ranging from the estimation of the labelling efficiency and the proportion of newly synthesized mRNA in the total mRNA, to the modelling of RNA metabolism by a multi-compartment ordinary differential equation system, to the parameter fitting of this model. If successful, this will be a breakthrough in the analysis of mRNA regulation.

Unfortunately, I have to state that this analysis contains several serious shortcomings, listed below, which invalidate the results. I believe that the data generated are of good quality and that the model as such is suitable for answering the questions posed by the authors. Perhaps some of my objections are not justified because I did not understand or misunderstood the details of the bioinformatic analysis due to a number of missing descriptions. However, it seems to me that the whole analysis needs to be repeated with major changes.

Major points

I will list the points mainly in the chronological order of their appearance in the manuscript. The points that are absolutely critical for the correctness of the results are marked with an asterisk (*).

1. The manuscript lacks relevant supporting information that is necessary to reproduce the analysis and prove the validity of the analysis. For example, the cDNA library preparation and short read sequencing protocol is completely missing. The only information given in the text is that they used poly(A) selection prior to reverse transcription. However, it is unclear how the sequences were fragmented, size-selected, and which sequencing protocol was used. Importantly, one needs to mention that a whole transcript sequencing protocol was used, which allows to quantify the rate of new mRNA in several exons of a transcript separately.
2. Using the STAR aligner without adjusting the mapping parameters results in a mapping bias from the unwanted removal of labeled reads. The parameters by which STAR was called are not given in the manuscript. The analysis must be repeated using SLAM-DUNK (Neumann, Herzog et al., BMC Bioinformatics 20, 258 (2019). <https://doi.org/10.1186/s12859-019-2849-7>), which is the standard mapping tool specifically developed to map T>C converted reads from SLAM-seq experiments in an unbiased manner. In addition, the authors did not mention whether they corrected the data for T>C SNPs or also for sites subject to post-transcriptional editing. At least the former is necessary.
3. (*) The fitting procedure for the relative abundances of the three RNA fractions (nuc, cyto and mem) is inadequate, for several reasons: 1. The regression approach, as it has been applied, assumes that the whole-cell ratio is the only data carrying an error, while the three fractions are assumed to be covariates whose measurement error is neglected. This can lead to detrimental effects. A total least squares regression would have been a better choice. 2. Even worse, any regression approach that relies on the assumption of Gaussian errors seems inappropriate here, because the data is discrete by nature (count data), and the errors, particularly for the low abundance transcripts, are very far from Gaussian. Further, there are many outliers to be expected. I did not find any pre-filtering for medium/high abundance genes to ameliorate this effect. As the estimation of the relative mRNA fraction abundances is the basis of all further claims, I require the authors to perform a more robust estimation. I suggest a non-parametric regression method.
4. Assessment of the effect of 4sU labeling on the transcriptome: The reason why the authors find only 3 differentially expressed genes at 180min is that they impose an additional fold change cutoff of 2, in addition to an (adjusted) p-values cutoff of 0.1. If one had used a p-value cutoff only, many genes (hundreds to thousands) would have been found (Supp. Fig. 1A). This procedure is not acceptable in view of the assumptions based on which the model parameters are fitted: The model only applies to transcripts in dynamic steady-state. Therefore, I suggest filtering out genes showing a temporal trend in expression (as defined, e.g., by a linear regression with time as a covariate) from further analysis.
5. (*) Estimating labeling efficiency, which is pivotal to estimating all other parameters, is flawed. The EM algorithm described in (Juerges et al.) is not open source, and the authors have re-implemented it. I had a look at the code on the authors' Github repository (in particular "mixture_optimization.R"), and I found the code sound. The algorithm as such is applicable in the present situation, given the labeling concentrations and durations of the time series experiment.
The problem is that the present study uses a sequencing protocol that maps mRNA fragments from the whole transcript, whereas previous studies (Juerges et al., Herzog et al., Mueller et al., Smalec et al.) use 3'-end sequencing. This avoids the problem that the authors also recognized and even quantified in Figure 2: exons distal to the 3'-end of a transcript experience a shorter effective labeling period than exons at the 3'-end, and this labeling period lies in the past, viewed from the time of sampling. Including all mapped reads from a sample in the estimation leads to a systematic underestimation of the labeling efficiency. A potential remedy might be inclusion only of reads that are within a reasonably short distance of the 3'end (say, 5kb corresponding a few minutes the Polymerase takes to synthesize this stretch). Of course, this comes at the cost of fewer reads

that can be considered for estimation.

6. The classification of transcripts as membrane-localized or cytosol-localized is inconsistent. First, the cutoff for "cytosol-localized in Fig. 1E should at least be below 1 (meaning more mRNA in the cytosol than at the membrane), but it was chosen to be 1.5. Second, the ratio shown in Fig. 1E depends crucially on the global estimate of the membrane-bound vs. the cytosolic mRNA amount. The latter has been obtained by a crude fitting procedure (see above) which is notoriously unstable.

7. It is enigmatic to me that - according to the authors' estimates, about 50% of all mRNA are membrane-localized, while about 80% of the transcripts are estimated to be enriched in the cytosol relative to the membrane. This implies that there must be a few highly abundant mRNAs that are preferentially localized at the membrane. I would like to see some further investigation of this strange effect. Moreover, the estimation of relative abundances is based on the total mRNA only. A second, complementary approach can be based on the mRNA-specific model parameters that they estimated: Using those, one can derive the relative abundances of steady-state nuclear, cytosolic and membrane-localized mRNA levels of each transcript. Summing over all transcripts that can be analyzed yields another estimate of the relative abundances of total mRNA. Comparing the result with the relative abundances they obtained previously is an essential step to verify the consistency of the parameter estimation.

8. (*) I appreciate the elegant re-formulation of the multi-compartment models in terms of the relative mRNA abundances (relative to their steady-state abundance). This facilitates the formulation of the ODE system and its parameterization. The fitting of the parameters however could be improved. The fitting procedure (Levenberg-Marquardt weighted least squares fit, which is ok) results in a point estimator. Yet there has been no attempt to estimate its variance, either using Fisher information or Monte Carlo sampling. It is indispensable to assign confidence regions to the parameter estimates, because many of the transcripts simply are not expressed highly enough (and hence sequenced deeply enough) to yield reliable estimates. In particular, I doubt that up to 5 parameters can be reliably fitted to these time series, especially the r3 and r4 parameters, when nuclear export is slow.

The authors performed two independent replicates of their experiment. There has to be a comparison of the two independent parameter estimates obtained from applying the fitting procedure to each of the two replicate time series experiments separately.

9. The models (3-step vs. 4-step model) that are fitted to the respective transcripts are chosen depending on the cytosolic vs membrane-bound mRNA ratio. As both models have their own biases, it is not meaningful to compare rate estimates from two different models (despite Fig. 5D, showing a consistency of the pre-processing and nuclear retention rate estimates between the 3-step and the 4-step model). All downstream analyses, most seriously the gene set enrichment analyses, will be compromised by this systematic difference between the transcripts using the 3-step respectively the 4-step model.

10. (*) It remains unexplained how the pre-mRNA by nuclear mature RNA ratio has been estimated. The only hint I found in the manuscript was this: "More specifically, the mean share of new to total RNA for exons and introns across replicates was used as input data for the fit (see example gene with data from cytosolic compartment in Fig. 2B)". I interpret it such as the reads overlapping with introns are considered pre-mRNA. It is unlikely that there are enough intron-overlapping reads per transcript to produce a reliable estimate of the pre-mRNA processing rates. Even if this were given, the reads that map fully onto exons will be a mixture of pre-mRNA and mature nuclear mRNA, and thus are not suitable as input for the fitting of the share of new, mature nuclear mRNA.

Furthermore it is unclear which biological process the authors aim to quantify by what they call "pre-mRNA processing rate". It cannot be the rate at which pre-mRNA is spliced, because this also happens co-transcriptionally. What they actually try to measure is the rate by which introns present in polyadenylated RNA are removed - a rather elusive quantity.

11. (*) The estimation of the transcription elongation rates is fundamentally flawed, which is related to point 5 above. The authors implicitly assume that the upstream exons of new RNA are labeled with the same efficiency as the exons proximal to the 3' end of the transcript. This however is not true, as the labeled upstream exons are synthesized much longer before they become part of a mature transcript. Consequently, the time at which they were labeled dates back to an earlier time point at which the labeling efficiency was potentially much lower. Ignoring this effect will lead to a downward bias of the estimated share of new RNA of this exon compared to the other exons. In the left plot of Figure 2B, the curves of the upstream exons will appear flatter as they really are. This in turn will be interpreted as a greater time delay, resulting in a downward bias of the elongation rate. Again, the authors do not provide a detailed description of how the polymerase II elongation rates were estimated. Since the authors did not even mention the many pitfalls I identified, I have to assume that they did not consider them.

Minor points

12. p4: The authors state that (Müller et al., 2023) provide evidence for mRNA degradation in the nucleus. This is incorrect. On the contrary, Müller et al. do not find evidence for substantial mRNA degradation in the nucleus. I assume that the authors meant Smalec et al., 2022), who do find some nuclear mRNA degradation.

13. Figure 1C needs clarification. It is stated in the Results and Methods that two replicate time series were measured, yet Figure 1C shows 6 replicates. Are these technical replicates? Was each sample sequenced on three lanes?

14. Figure 1D does not prove that the fitting procedure is valid. As the three RNA fractions (nuc, cyto and mem) have a similar expression profile as whole-cell RNA, so will have any convex combination of the three fractions. The plot also does not allow an assessment of the goodness-of-fit. Figure 1D should be removed.

15. The sentence "In addition, the distribution of share of labeled mRNAs over all quantified genes (n=9862) reflected the cellular mRNA maturation and showed expected trends for all subcellular compartments (Fig. 1E)." is difficult to grasp and could be rephrased. Further, it should refer to Fig. 1F instead of 1E.

16. p8: The references in the sentence "To integrate the subcellular transcriptome and metabolic RNA labeling data and to estimate half-lives as a description of subcellular mRNA dynamics, we developed transcript-wise mathematical models similar to previous work (28771467)." are missing.

17. p8: "(membrane decay g_4)" - it should probably be read " γ_4 " instead.
18. p9: Box 1 should explain the meaning of the quantities x_1, \dots, x_4 (pre-, nuclear mature, cytosolic, membrane-associated RNA).
19. p10: In Figure 2B it should be explained that by "time", the time of the labeling period until cell harvesting is meant. It also must be explained how the corresponding data was obtained
20. In Box 1, it is unclear to me where the steady-state ratios mentioned there come into play (to rescale, merely the individual steady-state levels need to be calculated, which is straightforward), or where these steady-state ratios are used later in the results. Please either explain or remove these ratios from the box.
21. Box 2 is not really helpful, as the formulas are not interpretable. It should be moved to the Supplementary methods. I would not even use the closed form solutions for prediction, as they contain singularities, which can make the predictions numerically unstable when p_i is approximately equal to p_{i+1} for some $i=1,2,3$. One might use a standard ODE solver instead, which is fast and numerically stable (but I expect this will leave the results essentially unchanged).
22. It would have been nice if the authors had measured a time point in the labelling time series (e.g. 60 min) simultaneously for a labelling efficiency of 500uM and 100uM 4sU. This would have given two independent estimates of the proportion of new mRNA in each transcript population and would have provided a strong argument for why switching labelling concentrations does not introduce a significant bias.

Reviewer #3:

In the present manuscript, Steinbrecht, Minia and colleagues combine SLAM-seq with biochemical fractionation to derive subcellular mRNA kinetics in mouse embryonic stem cells. The approach is elegant, sound and results are well presented. Overall the manuscript is very well written, follows a clear logic and, while descriptive in nature, yields valuable insights into the subcellular residence time of RNAs. I believe the manuscript is of high quality and well suited for MSB. I have a few minor comments that should be addressable without additional experiments:

I wonder if the authors need to account for the fast cell cycle of mESC. Dependent on cell density and nutrient supply, mESCs have a cell cycle of 8-12 hours, hence dilution by cell cycle contributes to the observed decay, especially for those genes with a long half life.

Page 8, second paragraph - one citation not properly inserted (28771467)

Fig 2B: The fitting of labeling delay appears to consistently underestimate the actual values (discrepancy fit curve and data points). This may have a trivial reason but it should be discussed.

Fig 3D: it is surprising to me that cytosolic transport correlated with gene length but not number of exons. There is a good albeit not perfect correlation between exon count and gene length (0.42 according to <https://doi.org/10.3389/fgene.2021.559998>), so one would expect both to yield similar correlation with a third variable like cytosolic transport.

Fig 4C, can you indicate how many transcripts are in each category? The authors state that "Membrane-localized mRNAs encoding no known targeting signal surprisingly show the shortest median cytosolic half-life" but the difference is really small and maybe not significant. Could the authors speculate (maybe by investigating what is known about individual transcripts), what the mRNAs without a TS do at the membrane and how they get enriched there?

Fig 5D, rightmost graph, the explanation is insufficient or unclear: it seems that a subset of transcripts has very good correlation (on the diagonal). Then a large fraction is quite uncorrelated, and also a considerable number falls on the x axis (is that because there is a pseudocount added to transcripts undetected in the membrane fraction?)

Just out of curiosity, could the authors correlate the subcellular transcript kinetics with presence of miRNA binding sites, presence of annotated upstream sORF, length of polyA-tail (maybe known from another study)

Response to reviewer comments (MSB-2024-12284)

We thank the reviewers for their constructive feedback and suggestions on our manuscript. Their comments have been invaluable in helping us improve it. We have addressed all the issues raised. Additionally, we greatly appreciate the reviewers' recognition that "the manuscript is of high quality and well suited for MSB" and that "this will be a breakthrough in the analysis of mRNA regulation".

Below, we provide our responses (in black) to each reviewer's comments (in blue). At the end of each response, we indicate in square brackets where the corresponding changes appear in the revised manuscript.

During the incorporation of the feedback, some parameters changed slightly, leading us to re-fit our model. All plots and descriptions using the best fit results have been updated. The main findings remain unchanged and are consistent with the initial version of the manuscript.

In our first manuscript version, we included nuclear degradation of polyadenylated transcripts by default. In our revised manuscript, we performed fits including and excluding nuclear degradation and then selected the better performing model based on the Bayesian Information Criterion (BIC). Now, 579 transcripts are modeled with nuclear degradation. The selection procedure is explained in the Methods section "Discriminating models with and without nuclear mRNA degradation" and illustrated in Fig. EV4A.

We hope our additions and changes to the manuscript are to the reviewers' satisfaction and could clarify the major questions and concerns that were raised. We believe the revised manuscript fulfills the high standards of MSB.

Reviewer #1

Steinbrecht et al have used 4sU metabolic labeling, SLAM-seq, time series analysis and subcellular fractionation to measure RNA flow in mouse ESCs. As the authors note in the introduction, their work builds upon many other studies using a similar set of approaches. The major distinguishing factor between this work and others is the addition of the membrane fraction and an analysis of transcription elongation. The authors developed a model that relies on the assumption of RNA conservation throughout their analysis. A lot of assumptions are made without experimental validation and many of the key findings should be bolstered by orthogonal validation. Overall, this paper would be a good fit for MSB if the following concerns can be addressed.

Major concerns:

1) The estimation of transcription elongation rates is novel and interesting, but it needs more work. An elongation model provides a delay that aligns with the data, but the authors do not demonstrate that their extracted rates actually reflect transcription elongation. The authors should compare their rates to those measured by other approaches, either published or measured by the authors. They could also perturb transcription elongation through Pol II trigger loop mutations to see if their rates change as expected (as in PMID 33631106). Specific questions are:

- a. Does the elongation rate estimation impact the measurements of downstream RNA flow rates?
- b. Is transcription elongation the main reason for the time-delay? Could premature termination or something else explain some of the effect?
- c. Do the estimated rates reflect real elongation rates? Jonkers et al estimate elongation rates in the same cell type as used in this study. How do the two datasets compare?
- d. If Jonkers et al is unsuitable for a comparison, are there other published datasets? Could the authors measure the rates themselves? Perhaps their flavopiridol experiment or an extension of the experiment could even be used to generate rough estimates of transcription elongation rates.

We thank the reviewer for these comments and in the following are the answers to the questions from above:

- We fit the transcription elongation rate to account for the labeling delay observed in exons distant to the 3' end. It does affect the estimation of the subcellular flow rates and, hence, it prevents the underestimation of subcellular flow rates that would otherwise occur.
- The combination of PolyA selection with 4sU incorporation is the main reason. Premature termination is very unlikely to have an effect.
- We compared our estimates of the transcription elongation rate to very recent data from Shao et al. 2022, where the authors use transient transcriptome sequencing combined with polymerase II occupancy profiling to estimate transcription elongation in mESCs. There is an overall moderate Spearman correlation of 0.33, with our median of 1.1 kb/min being slightly lower than their median of 1.8 kb/min (see supplied Fig). Our elongation rates and those of Shao et al. are distributed very similarly. Further, there is agreement with median elongation rate estimations of 1.25 to 1.75 kb/min measured before in five different human cell lines (Veloso et al, 2014). Correlation between our and Jonkers et al. data is low (Spearman $r=0.06$). As we have reasonable overlap with the more recent data, we assume that our estimation of transcription elongation is sound. [Fig. EV2F, p10 "Model parameterization unveils length-dependent elongation rate", p.19 "Discussion"]
- See response to c.

2) A key assumption of the authors' model is that "the RNA expression in the whole cell can be reconstructed by summing the subcellular RNA expression with a corresponding factor for each". This assumption at first seems reasonable, however, it is not likely to hold up. Subcellular fractionation is imperfect, and much RNA is likely lost during the many wash steps, which could vary across the transcriptome. There's also the chance for RNA degradation to occur post-lysis that would also lead to RNA loss. Can the authors quantify how much RNA might be lost by sequencing wash steps and work out how robust the modeling and rates estimates are due to possible loss?

As the data is normalized to transcripts per million (TPM), any proportional loss affecting all transcripts would be irrelevant for the analysis. Any transcript-specific loss that varies between compartments, which may e.g. occur due to degradation post lysis would lead to off-diagonal estimates for the total transcript. However, we don't observe those strongly off-diagonal points. Contaminations of the fractions can be estimated from proteins with known subcellular distribution. A Western blot analysis of the fractions is shown in Figure 1 shows that contaminations between fractions are negligible. Furthermore, TPM values in replicates are very similar to each other and show no significant inconsistencies.

3) Another model assumption is that RNA processing occurs post-transcriptionally, but >60% of all RNA maturation occurs co-transcriptionally. How does co-transcriptional processing impact the authors' models?

This is a very good point. As we polyA-select, the pre-mRNA processing rate describes the rate at which introns present in polyadenylated RNA are removed. As such, it is not the splicing rate that is quantified here, but rather a post-polyadenylation mRNA maturation rate. We expanded the description of the pre-mRNA processing rate in the Results subsection describing the model. [p8 "Model of subcellular mRNA dynamics"]

4) The section on nuclear-encoded mitochondrial transcripts is confusing and has a lot of caveats. It is unclear how a transcript can be largely cytoplasmic but with similar kinetics as other transcripts

aside from long nuclear half-lives. This would seem to make these transcripts largely nuclear residing. Furthermore, mitochondrial co-purification with the nucleus is well established for subcellular purification protocols. If the authors have mitochondrial contamination, the long nuclear half-lives for mitochondrial protein transcripts might be explained by this alone. Mitochondrial contamination was not monitored by the authors in their subcellular fractionation experiments, so the authors need to address this possibility.

We believe there is a misunderstanding here. We specifically refer to transcripts of nuclear DNA-encoded mitochondrial proteins, not mtDNA-encoded proteins. Mitochondrial contamination would not influence RNA levels of nuclear-encoded transcripts.

5) The authors mention that a Bayesian approach would be required to distinguish between the 3 step and 4 step models. However, the suitability of various models with different numbers of parameter can be determined via other statistical approaches, such as use of the Akaike information criterion. Could the authors estimate which model is a better fit to each gene's data?

This is an interesting point. We compared the BIC and the Akaike information criterion (AIC) between 3- and 4-step models. Both criteria gave very similar information. As the number of data points changes a lot when using the 4-step model (membrane fraction data), the comparison is difficult to interpret, especially as the number of new data points added in the 4-step model varies between genes as it increases with the number of exons. For well-fitted transcripts (reduced chi-sqr in the interval [0.8, 1.2]), the trend is that undefined and membrane-localized transcripts are fit better by 4-step model (with only 16%

of transcripts being preferentially fit by the 3-step model), while cytosol-localized transcripts show a trend of being better fit by the 3-step model (although for 32% of transcripts the 4-step model is still preferred according to BIC). For transcripts with very low reduced chi-sqr ($\text{red_chi} \ll 1$), the 4-step model is preferred, as the first term in the BIC/AIC (loglikelihood or in our case $\log(\text{chi}^2/N)$) becomes negative and scales with number of data points, i.e. more data points lower the BIC/AIC. This is particularly visible for genes with many exons, as the number of data points increases a lot when adding membrane data. In the plot on the right, the difference between BIC values of the 3-step minus the 4-step model are plotted over the number of exons for every gene. The threshold of when one model is preferred over the other are set to ± 10 and are based on Raftery (*Sociol. Methodol.*, 1995). One can see that for few exons, it is hard to choose one model over the other, while the spread of BIC difference increases with the number of exons. The trend that cytosol-localized transcripts are fitted better by the 3-step model and membrane-localized transcripts are fitted better by the 4-step model can be seen. Hence, we think that choosing the model based on membrane/cytosol expression ratio is justified.

6) A major finding of the manuscript is that nuclear retention time is the rate-limiting step for RNAs as they spend most of their life spans in the nucleus. This is surprising. A microscopy-based approach, such as FISH, would be necessary to confirm this main finding, either at the single gene level or at the entire polyA transcriptome level.

We agree that an independent measurement using a different technique, such as FISH, would help to undermine our main finding. As our labs do not have much expertise using FISH, we were not able to perform FISH experiments. We note that Ren et al. (*Nature Methods*, 2023), using metabolic labeling with smFISH, observed that “a substantial fraction of reads (around 40%) was retained in the nucleus even after 6 h”. Choi et al. (*Mol. Cell*, 2024) conducted mRNA interactome capture experiments combined with pulse-chase labeling in HeLa cells and found that export competent mRNPs are assembled mainly at 60–70 min chase time while cytoplasmic translation factors peak around 100 min chase time (Figure 5 in Choi et al. *Mol Cell*, 2024). Battich et al. (*Cell*, 2015) derived nuclear retention times between ~5–90 min for close to 300 newly synthesized transcripts using smFISH, with a median of ~20 min. However, the authors argued that these nuclear retention times are likely an underestimation for most other genes, since these genes were fast-responding genes during stress signaling. Battich et al. (*Cell*, 2015) and Halpern et al. (*Cell Reports*, 2015) proposed that nuclear retention acts as a noise buffer to stabilize cytoplasmic mRNA levels. These independent, orthogonal studies corroborate our finding of long nuclear retention. [p19 “Discussion”]

Minor comments

1) Could the authors speculate on how species/cell type specific the authors' findings are?

Certainly. As the nucleus makes up a large part of the cell volume in mESCs and we observe a high relative nuclear mRNA amount, the nuclear retention half-lives estimated here are likely to be higher than in other, differentiated mammalian cell types. Ren et al. (*Nature Methods*, 2023) measured subcellular mRNA kinetics in different primary human cell lines and found that kinetic clusters showed consistent dynamic patterns within each gene cluster across different cell types, while marker genes showed different dynamics between cell types (see Fig. 5l,m,n). Chen et al. (*PLOS Genetics*, 2017) measured nuclear retention and cytoplasmic decay in *Drosophila* and found an average half-life slightly above one hour for both parameters, suggesting that nuclear retention is not rate-limiting in *Drosophila*. Miller et al. (*Mol. Sys. Bio.*, 2011) showed that, in *Saccharomyces cerevisiae*, mRNAs have a median half-life of around 11 minutes. Given the findings of recent publications, we speculate that overall mRNA dynamics are similar in mammalian cells, but are different (with overall shorter half-lives) in non-mammalian cells.

[p20 “Discussion”]

Reviewer #2

The present manuscript is a remarkable attempt to quantitatively dissect the mRNA life cycle into the processes of transcription, pre-mRNA maturation, nuclear export, cytosolic degradation, and transport/attachment to intracellular membranes (endoplasmic reticulum). To achieve this goal, the authors combine sophisticated experimental techniques such as subcellular fractionation and metabolic RNA labelling with 4-thiouridine in an RNA-seq time series experiment. While this is difficult enough, the biggest challenge is the data analysis, which requires advanced modelling techniques, ranging from the estimation of the labelling efficiency and the proportion of newly synthesized mRNA in the total mRNA, to the modelling of RNA metabolism by a multi-compartment ordinary differential equation system, to the parameter fitting of this model. If successful, this will be a breakthrough in the analysis of mRNA regulation.

Unfortunately, I have to state that this analysis contains several serious shortcomings, listed below, which invalidate the results. I believe that the data generated are of good quality and that the model as such is suitable for answering the questions posed by the authors. Perhaps some of my objections are not justified because I did not understand or misunderstood the details of the bioinformatic analysis due to a number of missing descriptions. However, it seems to me that the whole analysis needs to be repeated with major changes.

Major points

I will list the points mainly in the chronological order of their appearance in the manuscript. The points that are absolutely critical for the correctness of the results are marked with an asterisk (*).

1. The manuscript lacks relevant supporting information that is necessary to reproduce the analysis and prove the validity of the analysis. For example, the cDNA library preparation and short read sequencing protocol is completely missing. The only information given in the text is that they used poly(A) selection prior to reverse transcription. However, it is unclear how the sequences were fragmented, size-selected, and which sequencing protocol was used. Importantly, one needs to mention that a whole transcript sequencing protocol was used, which allows to quantify the rate of new mRNA in several exons of a transcript separately.

Thank you for highlighting this omission. We provided a subsection in the Methods on RNA isolation, alkylation of 4sU-labeled RNA and SLAM-seq that describes the sequencing protocol in detail.

[p. 23 “RNA isolation, alkylation of 4sU-labeled RNA and SLAM-seq”]

2. Using the STAR aligner without adjusting the mapping parameters results in a mapping bias from the unwanted removal of labeled reads. The parameters by which STAR was called are not given in the manuscript. The analysis must be repeated using SLAM-DUNK (Neumann, Herzog et al., BMC Bioinformatics 20, 258 (2019). <https://doi.org/10.1186/s12859-019-2849-7>), which is the standard mapping tool specifically developed to map T>C converted reads from SLAM-seq experiments in an unbiased manner. In addition, the authors did not mention whether they corrected the data for T>C SNPs or also for sites subject to post-transcriptional editing. At least the former is necessary.

Thank you again for highlighting this omission. We already considered these issues, but failed to include this in the manuscript. Mapping with STAR, we lowered the value of the “seedSearchStartLmax” option to 10 (from default 50), which means each read is fragmented into pieces with a maximum of 10 bases. This increases the mappability of highly labeled reads with multiple T2C/A2G conversions (in a similar fashion to nextgenmap used in the SLAMD-DUNK pipeline). Furthermore, we used VarScan’s mpileup2snp command to obtain SNPs from the BAM files from all unlabeled samples. We created one pileup file containing only T2C and A2G mutations, which was used later on to exclude those SNPs from being erroneously counted as 4sU-labeling induced T2C conversion. We added two subsections to the Methods that describe these steps.

[p24 “Alignment and read counting”, p24 “Exclusion of SNPs from T2C counting”]

Still, to confirm the validity of our metabolic labeling sequencing data, we repeated the alignment and T2C counting with SLAM-DUNK. When comparing the SLAM-DUNK data (where alignment is done only on the 3' UTR) with data from our most 3' exons, we find very high agreement (Spearman $r > 0.9$) from 30 minutes onwards. The attached plot shows scatterplots of T2C/T count data of all expressed transcripts from all subcellular samples, faceted by timepoints. There is no systematic over- or underestimation visible (the first 3 KDE levels follow the diagonal very closely). Even the $t=15$ and $t=20$ min time points show high agreement ($r = 0.73$ and $r = 0.87$, resp.). For the $t=0$ min samples the agreement is lower, since the T2C mutations are largely sequencing errors. We conclude that our data analysis pipeline matches the standard of published metabolic labeling analysis pipelines. Since SLAM-DUNK does not calculate the conversion rate of each sample and therefore the ratio of new to total RNA cannot be determined in this pipeline, we couldn't use the SLAM-DUNK data to re-run our model.

3. (*) The fitting procedure for the relative abundances of the three RNA fractions (nuc, cyto and mem) is inadequate, for several reasons: 1. The regression approach, as it has been applied, assumes that the whole-cell ratio is the only data carrying an error, while the three fractions are assumed to be covariates whose measurement error is neglected. This can lead to detrimental effects. A total least squares regression would have been a better choice. 2. Even worse, any regression approach that relies on the assumption of Gaussian errors seems inappropriate here, because the data is discrete by nature (count data), and the errors, particularly for the low abundance transcripts, are very far from Gaussian. Further, there are many outliers to be expected. I did not find any pre-filtering for medium/high abundance genes to ameliorate this effect. As the estimation of the relative mRNA fraction abundances is the basis of all further claims, I require the authors to perform a more robust estimation. I suggest a non-parametric regression method.

This is a valid criticism. We did consider pre-filtering for highly abundant genes, but the fit results did not change. Hence, we left the filtering out. To address your point, we now re-fitted the relative abundances of the subcellular RNA fractions using a total-least-squares fitting method (ODR module from `scipy`) that includes errors on whole-cell and all subcellular TPM values. Furthermore, we filtered for highly-expressed genes (TPM > 50, corresponding to roughly 1000 counts), where the assumption of Gaussian errors should hold. The relative abundances changed slightly, with nuclear, cytosolic and membrane coefficients of 0.39, 0.15 and 0.45 (before: 0.33, 0.2 and 0.47). The estimated errors on the parameter values are at least two orders of magnitude smaller than the values. The fit has reduced chi-squared value of 1.9. We believe this method reliably estimates the relative mRNA abundances. Consequently, we re-ran our fit of the subcellular flow rates using the expression ratios including the new relative mRNA abundances.
[p26 "Quantifying relative mRNA abundance of subcellular compartments", Fig. 1D]

4. Assessment of the effect of 4sU labeling on the transcriptome: The reason why the authors find only 3 differentially expressed genes at 180min is that they impose an additional fold change cutoff of 2, in addition to an (adjusted) p-values cutoff of 0.1. If one had used a p-value cutoff only, many genes (hundreds to thousands) would have been found (Supp. Fig. 1A). This procedure is not acceptable in view of the assumptions based on which the model parameters are fitted: The model only applies to transcripts in dynamic steady-state. Therefore, I suggest filtering out genes showing a temporal trend in expression (as defined, e.g., by a linear regression with time as a covariate) from further analysis.

This is a valid point. We repeated the differential expression analysis of RNA count data from whole-cell extracts and now distinguished between samples subjected to high (500 μ M) or low (100 μ M) 4sU labeling. The results can be seen in Fig. EV1A. Comparing all subsequent time points to the t=0 time point for the high 4sU dose, we found that up until 40 min no genes, at 60 min one gene (Kantr) and from 120 min onwards more than 250 genes were differentially expressed ($\log_2\text{FC} > 0.5$ and adjusted p-value < 0.05). We decided to exclude all samples with 500 μ M 4sU labeling for more than 60 min and keep samples using 500 μ M labeling up until and including 60 min. Accordingly, Kantr was removed from subsequent analyses. For samples labeled with 100 μ M 4sU, the 120 min and 180 min time points were compared to the 60 min time point (as no control measurements were done in this batch). No genes were found to be differentially expressed. As the higher dosage already has a minimal effect at 60 min, it can be assumed that the lower dosage leaves the cells unperturbed at 60 min and hence the 60 min 100 μ M samples are a valid control. As the lowest adjusted p-value in the comparisons for the low dosage was 0.6, it can be assumed that the transcriptional state of the cells does not change during the 180 min labeling duration with 100 μ M 4sU. All in all, this shows a negligible effect of 4sU labeling on the transcriptome in our samples.
[p25 "Minimal effect of metabolic labeling on the transcriptome", Fig. EV1A]

5. (*) Estimating labeling efficiency, which is pivotal to estimating all other parameters, is flawed. The EM algorithm described in (Juerges et al.) is not open source, and the authors have re-implemented it. I had a look at the code on the authors' Github repository (in particular "mixture

optimization.R"), and I found the code sound. The algorithm as such is applicable in the present situation, given the labeling concentrations and durations of the time series experiment. The problem is that the present study uses a sequencing protocol that maps mRNA fragments from the whole transcript, whereas previous studies (Juerges et al., Herzog et al., Mueller et al., Smalec et al.) use 3'-end sequencing. This avoids the problem that the authors also recognized and even quantified in Figure 2: exons distal to the 3'-end of a transcript experience a shorter effective labeling period than exons at the 3'-end, and this labeling period lies in the past, viewed from the time of sampling. Including all mapped reads from a sample in the estimation leads to a systematic underestimation of the labeling efficiency. A potential remedy might be inclusion only of reads that are within a reasonably short distance of the 3' end (say, 5kb corresponding a few minutes the Polymerase takes to synthesize this stretch). Of course, this comes at the cost of fewer reads that can be considered for estimation.

The reviewer raises an interesting point here. However, we don't think our analysis is affected strongly by the point raised here. When looking at the change in estimated conversion rates over time (see attached plot), the conversion rates change slowly from 15 to 60 min for the 500 μM 4sU concentration with the 60 min samples having roughly twice the values as the 15 min samples. For the long labeling times with 100 μM 4sU however, the conversion rates stay constant. This indicates that the "currently available"

4sU concentration to reads distal compared to close to the 3' end is not different for the longer time points and should not be drastically different for shortest time points. As the reviewer noted, our binomial mixture model correctly rescales the labeling data. Furthermore, restricting the input data to only the last 5 kb to the 3' end would mean a great loss of information. As SLAM-seq data is subject to statistical noise, our subcellular flow rates might be influenced a lot more by statistical noise if we used only the last 5 kb of any gene. We hope the reviewer is satisfied if we leave our approach to include all reads. [Fig. EV1D]

6. The classification of transcripts as membrane-localized or cytosol-localized is inconsistent. First, the cutoff for "cytosol-localized in Fig.1E should at least be below 1 (meaning more mRNA in the cytosol than at the membrane), but it was chosen to be 1.5. Second, the ratio shown in Fig. 1E depends crucially on the global estimate of the membrane-bound vs. the cytosolic mRNA amount. The latter has been obtained by a crude fitting procedure (see above) which is notoriously unstable.

Our classification of cytosol- and membrane-localized transcripts is based on the bimodality of the distribution seen in Fig. 1E. Choosing a cutoff < 1 for cytosol-localized transcripts has no biological foundation, as localization to the ER is not exclusive for transcripts encoding target signals (Chen et al. *MBoC* 2011, Reid et al. *JBC* 2011). Transcripts from many genes encoding no TS are partially localized to the ER. As such, there is no "correct" ratio for cytosol-localized transcripts. We chose the cutoff visually, dividing the groups into transcripts that are highly enriched in the membrane fraction (which comprise mostly transcripts encoding TS, but not exclusively) and transcripts that are mainly cytosol-localized or have roughly equal share of cytosol and membrane localization. Hence, this classification also does not depend on the coefficients from the relative mRNA amounts, as those simply shift the distribution, but do not change its shape.

7. It is enigmatic to me that - according to the authors' estimates, about 50% of all mRNA are membrane-localized, while about 80% of the transcripts are estimated to be enriched in the cytosol relative to the membrane. This implies that there must be a few highly abundant mRNAs that are preferentially localized at the membrane. I would like to see some further investigation of this strange effect. Moreover, the estimation of relative abundances is based on the total mRNA only. A second, complementary approach can be based on the mRNA-specific model parameters that they estimated: Using those, one can derive the relative abundances of steady-state nuclear, cytosolic and membrane-localized mRNA levels of each transcript. Summing over all transcripts that can be analyzed yields another estimate of the relative abundances of total mRNA. Comparing the result with the relative abundances they obtained previously is an essential step to verify the consistency of the parameter estimation.

We hope that our response to point 6 already answers the question raised here. The attached plot shows the membrane/cytosol expression ratio of different groups of transcripts (before Supp. Fig. 1E, now Fig. EV1C). Mainly transcripts encoding targeting signals have a high membrane enrichment. See Chen et al. *MBoC* 2011 and Reid et al. *JBC* 2011 for more details on the topic of mRNA localization.

[Fig. EV1C]

8. (*) I appreciate the elegant re-formulation of the multi-compartment models in terms of the relative mRNA abundances (relative to their steady-state abundance). This facilitates the formulation of the ODE system and its parameterization. The fitting of the parameters however could be improved. The fitting procedure (Levenberg-Marquardt weighted least squares fit, which is ok) results in a point estimator. Yet there has been no attempt to estimate its variance, either using Fisher information or Monte Carlo sampling. It is indispensable to assign confidence regions to the parameter estimates, because many of the transcripts simply are not expressed highly enough (and hence sequenced deeply enough) to yield reliable estimates. In particular, I doubt that up to 5 parameters can be reliably fitted to these time series, especially the r3 and r4 parameters, when nuclear export is slow. The authors performed two independent replicates of their experiment. There has to be a comparison of the two independent parameter estimates obtained from applying the fitting procedure to each of the two replicate time series experiments separately.

We thank the reviewer for highlighting this shortcoming. When re-fitting, we included Imfit's standard errors in our data tables. Imfit calculates the standard error by inverting the Hessian matrix (which is equivalent to the Fisher information). For a small subset of genes, we calculated confidence intervals using Imfit's F-Test. Standard errors and confidence intervals agreed largely. Hence, we continued with using the standard errors, as they are computationally inexpensive to

calculate. We included a plot showing the distribution of the errors on all parameters for all genes. The median relative errors are around 10% to 25% for all parameters. Only a minority of transcripts show high relative errors. Together with the fact that there are no infinities in the standard errors on any of the parameters, we conclude that the estimation accuracy on the parameters is overall sufficient.

As we have varying numbers of replicates per time point, we cannot apply the fitting procedure to each replicate individually. The data for each sample comes from a different well with different cells, so there is no biological connection between e.g. the sample of the membrane fraction replicate 1 of 15 min or 20 min labeling. Only if we were continuously measuring the same cell or cells over time (as in some microscopy approaches, would comparing individual replicates make sense. Furthermore, our modeling approach uses the measurement uncertainties between different replicates as weights in the fitting and thus maximizes estimation accuracy by extracting the most information out of the available data. If we were to use only one replicate and ran the fit twice, we would also not be able to merge the results later on, as for any non-linear process, the parameter estimates cannot be simply averaged between replicates. [p27 "Model fitting of subcellular mRNA dynamics", Fig. EV2D]

9. The models (3-step vs. 4-step model) that are fitted to the respective transcripts are chosen depending on the cytosolic vs membrane-bound mRNA ratio. As both models have their own biases, it is not meaningful to compare rate estimates from two different models (despite Fig. 5D, showing a consistency of the pre-processing and nuclear retention rate estimates between the 3-step and the 4-step model). All downstream analyses, most seriously the gene set enrichment analyses, will be compromised by this systematic difference between the transcripts using the 3-step respectively the 4-step model.

This is a valid point. When doing the GSEAs, we used all rates from the 3-step model and only the membrane decay rate from the 4-step model (for all genes equally). Hence, the rates were intrinsically consistent. We clarified this at the appropriate text passage. [p25 "Additional data analysis", Fig. 1C]

10. (*) It remains unexplained how the pre-mRNA by nuclear mature RNA ratio has been estimated. The only hint I found in the manuscript was this: "More specifically, the mean share of new to total RNA for exons and introns across replicates was used as input data for the fit (see example gene with data from cytosolic compartment in Fig. 2B)". I interpret it such as the reads overlapping with introns are considered pre-mRNA. It is unlikely that there are enough intron-overlapping reads per transcript to produce a reliable estimate of the pre-mRNA processing rates. Even if this were given, the reads that map fully onto exons will be a mixture of pre-mRNA and mature nuclear mRNA, and thus are not suitable as input for the fitting of the share of new, mature nuclear mRNA.

Furthermore it is unclear which biological process the authors aim to quantify by what they call "pre-mRNA processing rate". It cannot be the rate at which pre-mRNA is spliced, because this also happens co-transcriptionally. What they actually try to measure is the rate by which introns present in polyadenylated RNA are removed - a rather elusive quantity.

We thank the reviewer for pointing out this omission. We have included a section in the Methods that describes how we obtain the ratio of new to total RNA for intronic regions. We hope this

clarifies it. Also, we added sentences further describing which biological processes the pre-mRNA processing rate measures in the corresponding Results subsection. [p24 “Custom genome annotation for intronic regions”, p8 “Model of subcellular mRNA dynamics”]

11. (*) The estimation of the transcription elongation rates is fundamentally flawed, which is related to point 5 above. The authors implicitly assume that the upstream exons of new RNA are labeled with the same efficiency as the exons proximal to the 3' end of the transcript. This however is not true, as the labeled upstream exons are synthesized much longer before they become part of a mature transcript. Consequently, the time at which they were labeled dates back to an earlier time point at which the labeling efficiency was potentially much lower.

Ignoring this effect will lead to a downward bias of the estimated share of new RNA of this exon compared to the other exons. In the left plot of Figure 2B, the curves of the upstream exons will appear flatter as they really are. This in turn will be interpreted as a greater time delay, resulting in a downward bias of the elongation rate. Again, the authors do not provide a detailed description of how the polymerase II elongation rates were estimated. Since the authors did not even mention the many pitfalls I identified, I have to assume that they did not consider them.

Here, an interesting point is raised. There is agreement between our mean elongation rate of 1.6 kb/min with median elongation rate estimations of 1.25 to 1.75 kb/min measured before in five different human cell lines (Veloso et al *Genome Res.* 2014). Increases in elongation rate of RNA polymerase II for longer genes, as well as our quantitative range of values, coincide with results from previous studies focusing specifically on transcript elongation (Jonkers et al. *eLife* 2014; Fuchs et al. *Genome Biology* 2014). The elongation rate could have a minimal downward bias, as we see in comparison with Shao et al. data with our median of 1.1 kb/min being slightly lower than their median of 1.8 kb/min. However, even if the elongation rate had a slight downward bias, this bias should be consistent across genes based on the argument made by the reviewer, so that observations as in Fig. 2D or results from the GSEA should still give meaningful insights. Descriptions of how the elongation rate was fitted can be found in the results section “Modeling of transcriptional elongation rates” and the caption text of Figure 2.

Minor points

12. p4: The authors state that (Müller et al., 2023) provide evidence for mRNA degradation in the nucleus. This is incorrect. On the contrary, Müller et al. do not find evidence for substantial mRNA degradation in the nucleus. I assume that the authors meant Smalec et al., (2022), who do find some nuclear mRNA degradation.

This was a typo and is fixed in the revised manuscript. We thank the reviewer for pointing this out. [p3 “Introduction”]

13. Figure 1C needs clarification. It is stated in the Results and Methods that two replicate time series were measured, yet Figure 1C shows 6 replicates. Are these technical replicates? Was each sample sequenced on three lanes?

We apologize for this error. We performed three experimental runs (with two replicates for each time point except t=0) with mostly different, but sometimes overlapping time points, leading to replicate numbers from 1 to 6. This is now correctly explained in the Methods. We thank the reviewer for pointing this out.

[p22 “Metabolic labeling and cell fractionation”]

14. Figure 1D does not prove that the fitting procedure is valid. As the three RNA fractions (nuc, cyto and mem) have a similar expression profile as whole-cell RNA, so will have any convex combination of the three fractions. The plot also does not allow an assessment of the goodness-of-fit. Figure 1D should be removed.

As already mentioned in our response to point 3, we re-fitted the relative mRNA amounts using orthogonal distance regression. We included now in the plot the reduced chi-sqr value as a goodness-of-fit statistic. If the reviewer insists, we can move the figure to the Supplements. However, we think it illustrates how we estimate the relative mRNA amounts and in this way helps the reader understand how we can use the expression ratios to constrain the fit parameters in our model.

[p26 "Quantifying relative mRNA abundance of subcellular compartments", Fig. 1D]

15. The sentence "In addition, the distribution of share of labeled mRNAs over all quantified genes (n=9862) reflected the cellular mRNA maturation and showed expected trends for all subcellular compartments (Fig.1E)." is difficult to grasp and could be rephrased. Further, it should refer to Fig. 1F instead of 1E.

The Figure reference is fixed. We thank the reviewer for pointing out this mistake.

[p6 "Spatiotemporal measurement of newly-transcribed mRNA"]

16. p8: The references in the sentence "To integrate the subcellular transcriptome and metabolic RNA labeling data and to estimate half-lives as a description of subcellular mRNA dynamics, we developed transcript-wise mathematical models similar to previous work (28771467)." are missing.

The reference is fixed in the revised manuscript. We thank the reviewer for pointing out this mistake.

[p7 "Model of subcellular mRNA dynamics"]

17. p8: "(membrane decay γ_4)" - it should probably by read " γ_4 " instead.

It now shows the greek letter. We thank the reviewer for pointing out this mistake.

[p7 "Model of subcellular mRNA dynamics"]

18. p9: Box1 should explain the meaning of the quantities x_1, \dots, x_4 (pre-, nuclear mature, cytosolic, membrane-associated RNA).

The variables x_1, \dots, x_4 were already described, but are now explicitly written at the corresponding place in the Box 1 description. The Box 1 description should now be more easily understandable.

[p8 Box1 caption]

19. p10: In Figure 2B it should be explained that by "time", the time of the labeling period until cell harvesting is meant. It also must be explained how the corresponding data was obtained

The figure caption now includes a sentence specifying that the x-axis corresponds to 4sU labeling time.

[Fig. 2B]

20. In Box 1, it is unclear to me where the steady-state ratios mentioned there come into play (to rescale, merely the individual steady-state levels need to be calculated, which is straightforward), or where these steady-state ratios are used later in the results. Please either explain or remove these ratios from the box.

Thank you for pointing out this unclarity. We have revised the caption of Box 1. Additionally, we added a concrete example of how the expression ratios were used to constrain the fit parameters in the Methods.

[Box 1, p27 "Model fitting of subcellular mRNA dynamics"]

21. Box 2 is not really helpful, as the formulas are not interpretable. It should be moved to the Supplementary methods. I would not even use the closed form solutions for prediction, as they contain singularities, which can make the predictions numerically unstable when p_i is approximately equal to p_{i+1} for some $i=1,2,3$. One might use a standard ODE solver instead, which is fast and numerically stable (but I expect this will leave the results essentially unchanged).

This is an interesting fact. We have not come across difficulties with singularities. The analytical solutions are used to e.g. calculate the model-derived whole-cell trajectories and we therefore think it is of use to leave them in the Methods section. Furthermore, they can be verified by the reader.

22. It would have been nice if the authors had measured a time point in the labelling time series (e.g. 60 min) simultaneously for a labelling efficiency of 500uM and 100uM 4sU. This would have given two independent estimates of the proportion of new mRNA in each transcript population and would have provided a strong argument for why switching labelling concentrations does not introduce a significant bias.

We labeled the 60 min time point with 100 μ M and 500 μ M 4sU. A comparison of the new-to-total ratio can be seen in the attached figure (also in Fig EV1E). There is no large bias visible between replicates 3,4 (high dosage, 500 μ M) and 5,6 (low dosage, 100 μ M). For the low dosage replicates, the 25th- and 75th percentile are further apart, as statistics are worse with less labeling. For the cytosolic fraction, replicates 5 and 6 (low dosage) have slightly higher median values. [Fig. EV1E]

Reviewer #3

In the present manuscript, Steinbrecht, Minia and colleagues combine SLAM-seq with biochemical fractionation to derive subcellular mRNA kinetics in mouse embryonic stem cells. The approach is elegant, sound and results are well presented. Overall the manuscript is very well written, follows a clear logic and, while descriptive in nature, yields valuable insights into the subcellular residence time of RNAs. I believe the manuscript is of high quality and well suited for MSB. I have a few minor comments that should be addressable without additional experiments:

I wonder if the authors need to account for the fast cell cycle of mESC. Dependent on cell density and nutrient supply, mESCs have a cell cycle of 8-12 hours, hence dilution by cell cycle contributes to the observed decay, especially for those genes with a long half-life.

This is a very good point. Unfortunately, it would be extremely difficult for us to account for the cell cycle, as we used bulk RNA sequencing. Ren et al. (*Nature Methods*, 2023) quantified subcellular flow rates using metabolic labeling combined with fluorescent imaging at single-cell resolution. They accounted for cell cycle. For most genes, subcellular rates were unaffected by cell cycle.

Only two clusters of genes showed differences in the G1 phase. Hence, we assume that our data should not be hugely confounded by cell cycle.

Page 8, second paragraph - one citation not properly inserted (28771467)

The citation is fixed in the revised manuscript. We thank the reviewer for pointing this mistake out. [p7 "Model of subcellular mRNA dynamics"]

Fig 2B: The fitting of labeling delay appears to consistently underestimate the actual values (discrepancy fit curve and data points). This may have a trivial reason but it should be discussed.

The elongation rate and all subcellular kinetic rates are fit simultaneously in all compartments. Hence, it is possible that fit curves are lower than individual data points in a specific compartment or for a specific exon, as then in another compartment or for other exons the fit curves lie above the data points. We updated Fig. 2B and it now shows *Sff1*, but it is still only an exemplary gene. We hope this clarifies the question.

[Fig. 2B]

Fig 3D: it is surprising to me that cytosolic transport correlated with gene length but not number of exons. There is a good albeit not perfect correlation between exon count and gene length (0.42 according to <https://doi.org/10.3389/fgene.2021.559998>), so one would expect both to yield similar correlation with a third variable like cytosolic transport.

Since it's only a 0.4 correlation, it is not completely transitive and we would not expect the correlation to be the same.

Fig 4C, can you indicate how many transcripts are in each category? The authors state that "Membrane-localized mRNAs encoding no known targeting signal surprisingly show the shortest median cytosolic half-life" but the difference is really small and maybe not significant. Could the authors speculate (maybe by investigating what is known about individual transcripts), what the mRNAs without a TS do at the membrane and how they get enriched there?

We already included in the figure caption how many transcripts are in each category, as there was simply no space in the figure axis. We hope this is satisfactory. The transcripts without a TS use "mRNA based localization", where localization to the ER membrane is independent of the synthesized output and instead facilitated by specific nucleotide sequences in the mRNA itself, see Lashkevich and Dmitriev (*Molecular Biology*, 2021) for a review of the subject.

[Fig. 4C]

Fig 5D, rightmost graph, the explanation is insufficient or unclear: it seems that a subset of transcripts has very good correlation (on the diagonal). Then a large fraction is quite uncorrelated, and also a considerable number falls on the x axis (is that because there is a pseudocount added to transcripts undetected in the membrane fraction?)

We found a near-perfect correlation for pre-mRNA processing ($r = 0.99$) and nuclear retention ($r = 0.98$), which are modeled the same way in both models (see Fig. 5D). However, for the cytosolic stability we find lower agreement ($r = 0.62$), as here the 4-step model used additional information to constrain the ratio between cytosolic and membrane half-lives. This results in all membrane-localized, but also many cytosol-localized transcripts exhibiting lower cytosolic half-lives in the 4-step than in the 3-step model. For the former, this was desired, as T2C mutations are distinct between cytosol and membrane compartments, whereas for the latter, this is undesirable, as the T2C mutations are similarly frequent in both membrane and cytosol compartments and the fit then systematically underestimates the cytosolic half-life to accurately fit the membrane half-life. As we use the 3- and 4-step model for cytosol- and membrane-localized transcripts, respectively, this poses no problem. We described this in the according Results section before.

The upper limit for all subcellular parameters was set to 2 1/min, corresponding to a half-life of roughly 0.5 min. In particular, many membrane-localized transcripts have a cytosolic half-life at this maximum (we assume this is meant by falling on the x-axis).
[Fig. 5D, p16-17 "Validation with external datasets and orthogonal approaches"]

Just out of curiosity, could the authors correlate the subcellular transcript kinetics with presence of miRNA binding sites, presence of annotated upstream sORF, length of polyA-tail (maybe known from another study)

These are very interesting suggestions for further inquiry. In particular, we would also like to correlate the subcellular rates with polyA tail lengths. However, we could not find a suitable dataset for polyA tail lengths in mESCs. Measuring the polyA tail lengths would be out of scope for this investigation. In the updated Fig. 3D we now included correlations with transcript length estimated with RSEM and GC content. We hope this also interests the reviewer.

29th Jul 2024

Manuscript Number: MSB-2024-12284R

Title: Subcellular mRNA kinetic modeling reveals nuclear retention as rate-limiting

Author: David Steinbrecht

Igor Minia

Miha Milek

Johannes Meisig

Nils Blüthgen

Markus Landthaler

Dear Markus,

Thank you for sending us your revised manuscript. We have now received feedback from the three referees who evaluated your study. From the comments below, you will see that Referees #1 and #3 support the publication of the manuscript. Referee #2, while acknowledging that most of their points have been satisfactorily addressed, still has concerns regarding the estimation of transcription elongation rates.

During our pre-decision cross-commenting process (in which the referees are given a chance to make additional comments, including on each other's reports), Referee #2 reiterated their concerns on this issue. Referee #1 agreed with this concern and suggested presenting this section of the manuscript as a promising area for RNA flow measurements, while emphasizing the need for further research. Referee #3 has recommended considering Referee #2's suggestion and, depending on its feasibility, either incorporating the suggestion into the study or providing reasonable discussion of why it is not feasible. I have included the referees' additional comments in full following their reports.

In light of this feedback, we would ask you to explore the analysis suggested by Referees #2 and #3 (specifically related to Referee #2's point #5). If incorporating the suggested analysis is not feasible, please clearly explain the reasons, tone down the language regarding elongation rates, and discuss potential caveats as well as the need for future research in this area. Finally, please address the other remaining issues raised by all three referees.

On a more editorial level, please address the following:

1. Please move the keywords to the main manuscript file.

2. Please remove the "Author contributions" section.

3. Data and code availability

- Rename the section to "Data availability".

- Please provide the specific URLs for GSE256335 and GSE252199 datasets. Remove the reviewer token from the manuscript text and ensure the datasets will be publicly available upon acceptance of the manuscript.

4. Please enter both corresponding authors in the author checklist.

5. Include the legend for Dataset EV1 as a separate tab in the Excel file.

6. Please indicate the statistical test used for data analysis in the legends of figures 3c; EV 1a, c, e; EV 4c.

7. Add missing information related to the sample size n in the legends of figures 1f; 3a; 4b; EV 5c-d.

When you resubmit your manuscript, please download our CHECKLIST (<https://bit.ly/EMBOPressAuthorChecklist>) and include the completed form in your submission.

Please note that the Author Checklist will be published alongside the paper as part of the transparent process (<https://www.embopress.org/page/journal/17444292/authorguide#transparentprocess>).

I look forward to receiving your revised manuscript soon.

Kind regards,

Jingyi

Jingyi Hou, PhD
Scientific Editor
Molecular Systems Biology

We realize that it is difficult to revise to a specific deadline. In the interest of protecting the conceptual advance provided by the work, we recommend a revision within 3 months (27th Oct 2024). Please discuss the revision progress ahead of this time with the editor if you require more time to complete the revisions. Use the link below to submit your revision:

IMPORTANT: When you send your revision, we will require the following items:

1. the manuscript text in LaTeX, RTF or MS Word format
2. a letter with a detailed description of the changes made in response to the referees. Please specify clearly the exact places in the text (pages and paragraphs) where each change has been made in response to each specific comment given
3. three to four 'bullet points' highlighting the main findings of your study
4. a short 'blurb' text summarizing in two sentences the study (max. 250 characters)
5. a 'thumbnail image' (550px width and max 400px height, Illustrator, PowerPoint or jpeg format), which can be used as 'visual title' for the synopsis section of your paper.
6. Please include an author contributions statement after the Acknowledgements section (see <https://www.embopress.org/page/journal/17444292/authorguide>)
7. Please complete the CHECKLIST available at (<https://bit.ly/EMBOPressAuthorChecklist>). Please note that the Author Checklist will be published alongside the paper as part of the transparent process (<https://www.embopress.org/page/journal/17444292/authorguide#transparentprocess>).
8. When assembling figures, please refer to our figure preparation guideline in order to ensure proper formatting and readability in print as well as on screen:

See also figure legend guidelines: <https://www.embopress.org/page/journal/17444292/authorguide#figureformat>

9. Please note that corresponding authors are required to supply an ORCID ID for their name upon submission of a revised manuscript (EMBO Press signed a joint statement to encourage ORCID adoption). (<https://www.embopress.org/page/journal/17444292/authorguide#editorialprocess>)
Currently, our records indicate that the ORCID for your account is 0000-0002-1075-8734.

Link Not Available

11. Include a Reagents and Tools Table as part of the Methods section, which can be downloaded from our author guidelines (<https://www.embopress.org/page/journal/17444292/authorguide#structuredmethods>)

*** PLEASE NOTE *** As part of the EMBO Press transparent editorial process initiative (see our Editorial at <https://dx.doi.org/10.1038/msb.2010.72>), Molecular Systems Biology publishes online a Review Process File with each accepted manuscripts. This file will be published in conjunction with your paper and will include the anonymous referee reports, your point-by-point response and all pertinent correspondence relating to the manuscript. If you do NOT want this File to be published, please inform the editorial office at msb@embo.org within 14 days upon receipt of the present letter.

Reviewer #1:

The authors adequately addressed most of my concerns. One of my original concerns was not understood by the authors. In the first round of review, the concern of mitochondrial co-purification was raised (major concern #4). The authors thought I was

asking about mitochondrial-encoded transcripts arising from mitochondrial co-purification. However, I meant the nuclear-encoded transcripts that localize to the surface of mitochondria through translation or other means. Most of these transcripts encode mitochondrial proteins that are imported into mitochondria after synthesis. Mitochondria are a known contamination to the nuclear fraction during cellular fractionation (see PMID 28766296). So, my concern remains. The authors should investigate whether their nuclear fraction has a substantial mitochondrial contamination and/or name the caveat and tone down their language in this section.

Reviewer #2:

The authors have made considerable efforts to improve the manuscript. Almost all the points I raised, in particular the critical points #3 and #8, #10, have been fully and satisfactorily addressed.

The critical points remaining are #5 and #11, which are related. I am really worried about the validity of the elongation rate estimates, and I ask for further clarification.

Point #5. As shown in their response, the conversion rate does change substantially from ~2.5% at t=15min to ~5% at t=40min (t=60min was excluded from the analysis). A factor of about 2 is a lot for the purpose of new to total ration estimation. I would like to ask the authors to perform a conversion rate estimation, at each time point of the time series, for the reads (read ends to be precise) that are binned into distance to 3'-end classes. Note that binning has already been done in Fig.2A (right), to show new/total ratio estimates, yet assuming identical conversion rates for the 5' and the 3' end of a transcript. For labeling efficiencies above 2%, the estimates for conversion rates should be stable if more than say 100k reads of length 75 are involved. Assuming a Pol II elongation rate of 2kb in murine ESCs (Jonkers I, Lis JT (2015), Nat Rev Mol Cell Biol 16: 167-177), one would expect to observe relevant effects in the 30-50kb length bin, and possibly also in the 15-30kb bin.

Point #11. The comparison with estimates from Shao et al. is not convincing. A Spearman correlation of 0.33 is only marginal. In particular, mRNAs with high elongation rates in Shao et al. do not agree with those in the present manuscript. The introduction of one elongation rate parameter per mRNA leads to highly unstable estimates (by the way, I also assume that the uncertainty of this parameter estimate did not enter the confidence interval estimation for the kinetic rates, which is suboptimal). I am sorry that the transcription elongation rate estimates presented here are not credible. To avoid overfitting, I suggest modeling the elongation rate v not transcript-specifically, but as a function of either the total transcript length, or a function of the distance-to-3'end.

Additionally, I have some remarks / suggestions that the authors might want to consider:

Point 2. The RNA extraction + mapping strategy has been described in much more detail, which is sufficient to reproduce the analysis. Still, I could not find any information on the number of mapped reads per sample, or ideally the distribution of mapped reads per gene/transcript. I recommend including at least a few numbers into Methods - Alignment and read counting. The Figure shown in the authors' response convincingly shows that the results do not critically depend on the choice of mapping strategy. I encourage the authors to include it as a supplementary Figure (into Methods - Alignment and read counting). The only argument that I do not understand is: "Since SLAM-DUNK does not calculate the conversion rate of each sample and therefore the ratio of new to total RNA cannot be determined in this pipeline, we couldn't use the SLAM-DUNK data to re-run our model.". The SLAM-DUNK mapped data merely needs to be threaded through the authors' pipeline (i.e., the Binomial mixture model) to obtain the conversion rate estimates, just like for the STAR-aligned data. But given the agreement of both mapping strategies, this is less relevant.

Point 6. The separation of the RNAs in accordance with the bimodal distribution illustrated in Fig. 1E is deemed appropriate. However, the term "membrane-localized" is misleading because it suggests that the majority of an mRNA's population is localized at the membrane (which is not the case according to the authors' definition). It is essential to explicitly state at the time of defining the term that it merely indicates a tendency for those transcripts to localize at the membrane, which is substantially higher than for the majority of mRNAs. Please include a corresponding statement in the manuscript.

Reviewer #3:

In the revised version, the authors have clarified and addressed nearly all of my comments. Only one remaining point I'd like to raise, since I maybe haven't explained properly in the last report: The author should comment how their model and half-life calculation takes into account (or not) the dilution of existing mRNA by cell division. In the SLAM-seq manuscript, Herzog, et. al. Nat Methods 2017 correct their half-lives according to the fact that their mESC double every 14.7h "To calculate RNA half-lives normalized to cell cycle length, $T > C$ conversions were multiplied by $2(\text{time point}/14.7\text{h})$."

I note there were numerous technical concerns raised by the other reviewers but it appears they have been largely addressed and/or clarified. I recommend publication of the manuscript.

Referee Cross-commenting

Reviewer #2:

I think the paper could be published entirely without the Pol2 elongation story. On the other hand, the elongation rate estimates add value to the manuscript. As I have expressed in my review, I have serious concerns that these estimates are valid. To be precise, I am almost convinced they are unreliable. The literature on Pol2 elongation and measurements of elongation speed (e.g., the relative elongation speed along one transcript) is vast (GRO/PRO-seq, TT-seq, NET-seq, Pol2-ChIP and combinations of these techniques). The authors should provide an analysis of the errors in their gene-specific elongation rate estimates (which is probably high) and they should check association of their elongation rates with chromatin structure.

Reviewer #1:

I agree that the errors are likely to be high and unreliable for the elongation rates. The authors could present this portion of the manuscript as a promising area for RNA flow measurements, but caveat that more work will need to be done.

Reviewer #3:

I think the elongation rate estimates are important for the correct interpretation of the remaining data, hence not possible to simply remove. I am not overly concerned with the low correlation with other elongation estimates (Jonkers et. al., Shao et. al.), since all are mere estimates of an unknown ground truth. Each method has its own error sources, hence it is not surprising to me that there are large discrepancies on individual genes. If the caveats are fairly described in the manuscript, I do not see an issue accepting it at MSB.

I do however agree with Reviewer #2 that if there is a better way of analysis, then it should be explored and the result discussed in the final manuscript. Hence, I would suggest to ask the authors for additional feedback regarding to "perform a conversion rate estimation, at each time point of the time series, for the reads (read ends to be precise) that are binned into distance to 3'-end classes. " as Reviewer 2 suggests. And hopefully the authors can either incorporate this suggestion or provide reasonable discussion why it is not possible. I also note that Reviewer #1 had one remaining concern that wasn't properly, so I would think that one more quick round of revisions could make the manuscript stronger.

Response to reviewer comments (Revision #2)

Dear reviewers,

We thank you for your constructive feedback. We addressed every remaining concern and hope that the revised manuscript is to your satisfaction. In the following, we give our responses (colored red) to each reviewer's comments (colored back).

Reviewer #1:

The authors adequately addressed most of my concerns. One of my original concerns was not understood by the authors. In the first round of review, the concern of mitochondrial co-purification was raised (major concern #4). The authors thought I was asking about mitochondrial-encoded transcripts arising from mitochondrial co-purification. However, I meant the nuclear-encoded transcripts that localize to the surface of mitochondria through translation or other means. Most of these transcripts encode mitochondrial proteins that are imported into mitochondria after synthesis. Mitochondria are a known contamination to the nuclear fraction during cellular fractionation (see PMID 28766296). So, my concern remains. The authors should investigate whether their nuclear fraction has a substantial mitochondrial contamination and/or name the caveat and tone down their language in this section.

We thank the reviewer for clarifying the point. To assess how much the transcriptome of the nuclear compartments is contaminated by mitochondrial contamination, we used the mitochondrially encoded transcripts as a marker. More precisely, we examined the ratio of the summed counts of those transcripts encoded in the mitochondrial genome over those encoded in the nuclear genome per sample. In agreement with the reviewer's intuition, we observed a contamination of the nuclear compartment, although it is minor (median ratio of mt / nuclear transcripts of 1% vs. 13% in the membrane compartment). We saw no noticeable contamination of the cytosolic compartment, and most mitochondrial transcripts were found in the membrane compartments.

Our conclusion is therefore that for those nuclear encoded mRNAs that are translated at the outer mitochondrial membrane (which are ~14% of the 700 transcripts encoding for mitochondrial proteins, classified as shown in Fig. 1E), nuclear retention rates may be slightly underestimated. We included this caveat in the manuscript (page 9 line 5).

Reviewer #2:

The authors have made considerable efforts to improve the manuscript. Almost all the points I raised, in particular the critical points #3 and #8, #10, have been fully and satisfactorily addressed.

The critical points remaining are #5 and #11, which are related. I am really worried about the validity of the elongation rate estimates, and I ask for further clarification.

Point #5. As shown in their response, the conversion rate does change substantially from ~2.5% at t=15min to ~5% at t=40min (t=60min was excluded from the analysis). A factor of about 2 is a lot for the purpose of new to total ratio estimation. I would like to ask the authors to perform a conversion rate estimation, at each time point of the time series, for the reads (read ends to be precise) that are binned into distance to 3'-end classes. Note that binning has already been done in Fig.2A (right), to show new/total ratio estimates, yet assuming identical conversion rates for the 5' and the 3' end of a

transcript. For labeling efficiencies above 2%, the estimates for conversion rates should be stable if more than say 100k reads of length 75 are involved. Assuming a Pol II elongation rate of 2kb in murine ESCs (Jonkers I, Lis JT (2015), Nat Rev Mol Cell Biol 16: 167-177), one would expect to observe relevant effects in the 30-50kb length bin, and possibly also in the 15-30kb bin.

We agree that increasing labeling rates with time will lead to uneven labeling, and our current method would average over these rates. The reviewer suggests binning transcript regions to estimate the time-dependent labeling rate. However, we do not think that this would add precision, as the transcription elongation rate varies nearly 1 order of magnitude between genes, thus these bins would not constitute a homogeneous group in terms of the time where they incorporated the label. We fully agree that other experimental methods are more suitable to provide a direct and also position-dependent transcription elongation rate. Yet, as also reviewer 3 noted, our model requires a transcription elongation rate to make full use of the full-length transcriptomes, as it incorporates the position-dependent delay in labeling that is present in all compartments. To investigate how much the accuracy of the elongation rate estimates influences other parameters of the model, we performed a sensitivity analysis (see below), quantifying how much a change in elongation rate influences the estimate of the other parameters. Interestingly and reassuringly, most parameters are on average only affected by 7% or less if the elongation rate is doubled. The only exception is the pre-mRNA processing rate which would, on average, change by 14%. Hence, our model (or more specifically the estimation of the subcellular kinetic rates) does not critically depend on the elongation rate estimate.

In response to this comment, we added sentences describing shortcomings, caveats and possible improvements in the Results sections “Modeling of transcriptional elongation rates” and “Model parameterization unveils length-dependent elongation rate” (page 7). Further, we now include a new extended view figure (Fig EV2) that focuses on the quality control of the elongation rate estimation and the sensitivity analysis and includes the plots shown below.

The plots below explain the sensitivity analysis for the elongation rate parameter. On the left, regression plots for two exemplary genes, Myc and Tfr, are shown. For all multiple-exons transcripts that do not show nuclear decay, we repeated the fitting procedure fixing the elongation rate at -50%, -10%, +10% and +50% of its best-fit value and initializing with the best-fit values of the remaining parameters. Then, for each transcript and parameter, we fitted a linear regression $y=mx$, where y is the change in the fitted parameter and x is the fixed relative change of the elongation rate. The slope m gives a measure of how much a parameter is influenced by changes in the elongation rate. On the right, we show the distribution of the slopes (scaled to indicate changes in %) for each subcellular parameter for all transcripts with multiple exons ($n=7935$). When the elongation rate is increased by 100%, the median change in the other parameters ranges from 4% for the membrane decay to 14% for the pre-mRNA processing rate.

Point #11. The comparison with estimates from Shao et al. is not convincing. A Spearman correlation of 0.33 is only marginal. In particular, mRNAs with high elongation rates in Shao et al. do not agree with those in the present manuscript. The introduction of one elongation rate parameter per mRNA leads to highly unstable estimates (by the way, I also assume that the uncertainty of this parameter estimate did not enter the confidence interval estimation for the kinetic rates, which is suboptimal). I am sorry that the transcription elongation rate estimates presented here are not credible. To avoid overfitting, I suggest modeling the elongation rate v not transcript-specifically, but as a function of either the total transcript length, or a function of the distance-to-3'end.

The uncertainty of the elongation rate did enter the uncertainty estimation of the other parameters, as all parameters were estimated together using a maximum likelihood approach. We included a histogram of the relative error of the elongation rate estimates in Fig EV3 (and attached here).

75% of the converged fits ($n=6496$, median=30%) have a relative error smaller than 63%. For fits where no point estimate could be made ($n=2007$), mostly because the parameter was stuck at its boundary, errors were high or could not be determined. In the plot, for standard errors containing infinities or NAs, the relative error was set to 1000 for visualization purposes. Boundary-limited estimates were deemed unreliable and hence excluded in analyses regarding the elongation rate.

Modeling the elongation rate based on gene length would not capture gene-to-gene variability in the elongation rate and might hinder estimation of the subcellular rates.

With regards to the moderate correlation to the Shao et al. dataset, we note that the Shao et al. and Jonkers et al. data show a Spearman correlation of 0.49, but only for $n=488$ genes (see Fig. EV4D in Shao et al. 2022). When we compared both datasets, we found $n=1050$ overlapping genes and a Spearman correlation of only 0.19 (which might be due to differences in measured isoforms). This shows that, as Reviewer #3 mentioned in his cross-comment, there is no consensus or ground truth for elongation rates, as each method has its own error sources. As the agreement between our and recently published data is only slightly worse than that from a publication specifically focused on elongation rates, we believe we can still use the modeling procedure that we used before, particularly because we showed that uncertainties in the elongation rate estimates influence the other subcellular parameters, which are the focus of the manuscript, only marginally.

Additionally, I have some remarks / suggestions that the authors might want to consider:

Point 2. The RNA extraction + mapping strategy has been described in much more detail, which is sufficient to reproduce the analysis. Still, I could not find any information on the number of mapped reads per sample, or ideally the distribution of mapped reads per gene/transcript. I recommend including at least a few numbers into Methods - Alignment and read counting.

The Figure shown in the authors' response convincingly shows that the results do not critically depend on the choice of mapping strategy. I encourage the authors to include it as a supplementary Figure (into Methods - Alignment and read counting).

We included the comparison to SLAM-DUNK as a supplementary figure 1 and added a short paragraph in the Methods section "T2C counting and normalization".

The only argument that I do not understand is: "Since SLAM-DUNK does not calculate the conversion rate of each sample and therefore the ratio of new to total RNA cannot be determined in this pipeline,

we couldn't use the SLAM-DUNK data to re-run our model.". The SLAM-DUNK mapped data merely needs to be threaded through the authors' pipeline (i.e., the Binomial mixture model) to obtain the conversion rate estimates, just like for the STAR-aligned data. But given the agreement of both mapping strategies, this is less relevant.

Point 6. The separation of the RNAs in accordance with the bimodal distribution illustrated in Fig. 1E is deemed appropriate. However, the term "membrane-localized" is misleading because it suggests that the majority of an mRNA's population is localized at the membrane (which is not the case according to the authors' definition). It is essential to explicitly state at the time of defining the term that it merely indicates a tendency for those transcripts to localize at the membrane, which is substantially higher than for the majority of mRNAs. Please include a corresponding statement in the manuscript.

We included a corresponding statement that clarifies the definition of membrane-localized (Page 5 last line). We further refer to a quality control plot in Fig. EV1D, where the large majority of membrane-localized transcripts code for proteins with signal peptides or transmembrane helices (roughly 1300 out of 1800).

Reviewer #3:

In the revised version, the authors have clarified and addressed nearly all of my comments. Only one remaining point I'd like to raise, since I maybe haven't explained properly in the last report: The author should comment how their model and half-life calculation considers (or not) the dilution of existing mRNA by cell division. In the SLAM-seq manuscript, Herzog, et. al. Nat Methods 2017 correct their half-lives according to the fact that their mESC double every 14.7h "To calculate RNA half-lives normalized to cell cycle length, T > C conversions were multiplied by 2 (time point/14.7h)."

Indeed, cell growth dilutes the mRNA and therefore influences the apparent rates estimated by our method. However, the vast majority of our subcellular half-lives are much shorter than 14h and therefore do not require a correction. We now specifically address this in the new version of the manuscript on page 20.

I note there were numerous technical concerns raised by the other reviewers but it appears they have been largely addressed and/or clarified. I recommend publication of the manuscript.

Referee Cross-commenting

Reviewer #2:

I think the paper could be published entirely without the Pol2 elongation story. On the other hand, the elongation rate estimates add value to the manuscript. As I have expressed in my review, I have serious concerns that these estimates are valid. To be precise, I am almost convinced they are unreliable. The literature on Pol2 elongation and measurements of elongation speed (e.g., the relative elongation speed along one transcript) is vast (GRO/PRO-seq, TT-seq, NET-seq, Pol2-ChIP and combinations of these techniques). The authors should provide an analysis of the errors in their gene-specific elongation rate estimates (which is probably high) and they should check association of their elongation rates with chromatin structure.

As written above, we agree that other methods may result in better and more precise estimates of elongation rates. As others have shown that histone modifications correlate with elongation rates (which we cite in the manuscript on page 12 line 27, e.g. Veloso et al. 2014 *Genome Res.*), we

believe that a further analysis of such correlation is beyond the scope of the manuscript and does not add value to the manuscript.

Reviewer #1:

I agree that the errors are likely to be high and unreliable for the elongation rates. The authors could present this portion of the manuscript as a promising area for RNA flow measurements, but caveat that more work will need to be done.

We refer to our response to reviewer #2, where we included a plot showing the relative errors of the elongation rate estimations here. For fits with converged elongation rate estimates (n=6496), the median relative error is 30%.

We added a paragraph describing the shortcomings, caveats and possible improvements in the Results sections "Modeling of transcriptional elongation rates" and "Model parameterization unveils length-dependent elongation rate" (page 7).

Reviewer #3:

I think the elongation rate estimates are important for the correct interpretation of the remaining data, hence not possible to simply remove. I am not overly concerned with the low correlation with other elongation estimates (Jonkers et. al., Shao et. al.), since all are mere estimates of an unknown ground truth. Each method has its own error sources, hence it is not surprising to me that there are large discrepancies on individual genes. If the caveats are fairly described in the manuscript, I do not see an issue accepting it at MSB.

I do however agree with Reviewer #2 that if there is a better way of analysis, then it should be explored and the result discussed in the final manuscript. Hence, I would suggest to ask the authors for additional feedback regarding to "perform a conversion rate estimation, at each time point of the time series, for the reads (read ends to be precise) that are binned into distance to 3'-end classes. " as Reviewer 2 suggests. And hopefully the authors can either incorporate this suggestion or provide reasonable discussion why it is not possible. I also note that Reviewer #1 had one remaining concern that wasn't properly, so I would think that one more quick round of revisions could make the manuscript stronger.

As detailed in the response to reviewer #2, we think that binned conversion rate estimates are unreliable due to differences in elongation rates between genes. As shown above, the sensitivity of the other parameters to the elongation rate are rather small, thus those parameters do not critically depend on a precise estimation of the elongation rate.

22nd Oct 2024

Manuscript number: MSB-2024-12284RR

Title: Subcellular mRNA kinetic modeling reveals nuclear retention as rate-limiting

Dear Markus,

Thank you again for sending us your revised manuscript. We are now satisfied with the modifications made and I am pleased to inform you that your paper has been accepted for publication.

Kind regards,
Jingyi

Jingyi Hou, PhD
Scientific Editor
Molecular Systems Biology
